# HOW GRADIENT DESCENT BALANCES FEATURES: A DYNAMICAL ANALYSIS FOR TWO-LAYER NEURAL NETWORKS

**Zhenyu Zhu**[†],      **Fanghui Liu**[‡],      **Volkan Cevher**[†]

† École Polytechnique Fédérale de Lausanne    ‡ University of Warwick

†{zhenyu.zhu, volkan.cevher}@epfl.ch    ‡fanghui.liu@warwick.ac.uk

## ABSTRACT

This paper investigates the fundamental regression task of learning $k$ neurons (*a.k.a.* teachers) from Gaussian input, using two-layer ReLU neural networks with width $m$ (*a.k.a.* students) and $m, k = \mathcal{O}(1)$, trained via gradient descent under proper initialization and a small step-size. Our analysis follows a three-phase structure: *alignment* after weak recovery, *tangential growth*, and *local convergence*, providing deeper insights into the learning dynamics of gradient descent (GD). We prove the global convergence at the rate of $\mathcal{O}(T^{-3})$ for the zero loss of excess risk. Additionally, our results show that GD automatically groups and balances student neurons, revealing an implicit bias toward achieving the minimum "balanced" $\ell_2$-norm in the solution. Our work extends beyond previous studies in exact-parameterization setting ($m = k = 1$, (Yehudai and Ohad, 2020)) and single-neuron setting ($m \geq k = 1$, (Xu and Du, 2023)). The key technical challenge lies in handling the interactions between multiple teachers and students during training, which we address by refining the alignment analysis in Phase 1 and introducing a new dynamic system analysis for tangential components in Phase 2. Our results pave the way for further research on optimizing neural network training dynamics and understanding implicit biases in more complex architectures.

## 1 INTRODUCTION

Learning a target function $f^\star : \mathbb{R}^d \to \mathbb{R}$ via neural networks through gradient descent or flow has received significant attention in machine learning theory. Research in this area primarily focuses on understanding the learnability and dynamics, aiming to theoretically explain the advantage of *feature learning* in neural networks. This problem is often studied under various assumptions about $f^\star$. For instance, $f^\star$ is frequently (implicitly) assumed to be smooth in a kernel regime (Jacot et al., 2018; Allen-Zhu et al., 2019; Arora et al., 2019). Additionally, $f^\star$ might possess further structures, such as being located on a low-dimensional subspace (Mousavi-Hosseini et al., 2023) or a manifold (Arora et al., 2022). A typical example is assuming $f^\star$ is a sparse polynomial (Abbe et al., 2022). In this setting, the separation between kernel methods and neural networks is well studied through metrics like the information exponent (Arous et al., 2021), leap complexity (Abbe et al., 2023), and generative exponent (Damian et al., 2024).

In contrast to smooth functions, another research direction focuses on non-smooth target functions, such as ReLU. This non-smoothness naturally highlights the difference between kernel methods and neural networks in terms of approximation ability (Bach, 2017). As a result, researchers have turned their attention to studying the learning dynamics to gain a deeper understanding of convergence. For instance, they investigate learning with a single ReLU neuron (Wu et al., 2023; Xu and Du, 2023) or multiple ReLU neurons (Zhou et al., 2021; Akiyama and Suzuki, 2023).

We consider the problem of learning one-hidden-layer ReLU networks under the Gaussian measure. The target function $f^\star$ is a sum of multiple ReLU neurons $f^\star(\boldsymbol{x}) := \sum_{l=1}^{k} \phi(\langle \boldsymbol{v}_l, \boldsymbol{x} \rangle)^1$ with the

---

[1]We assume that teacher neurons are positive. However, if both the student and teacher networks contain at least one negative ReLU neuron, our proof techniques can address such cases with minor modifications, requiring no significant changes to the overall analysis.

parameters $\{\boldsymbol{v}_l\}_{l=1}^k$, which can be learned from $n$ i.i.d. samples $\{(\boldsymbol{x}_i, f^\star(\boldsymbol{x}_i))\}_{i=1}^n$ via a two-layer neural network with $m$ (student) neurons with random Gaussian initialization $\{\boldsymbol{w}_i\}_{i=1}^m \sim \mathcal{N}(\boldsymbol{0}, \sigma^2 \boldsymbol{I}_d)$ under the expected squared loss:

$$L(\boldsymbol{W}) = \frac{1}{2}\mathbb{E}_{\boldsymbol{x}\sim\mathcal{N}(\boldsymbol{0},\boldsymbol{I}_d)}\left(\sum_{i=1}^m \phi(\boldsymbol{w}_i^\top \boldsymbol{x}) - f^\star(\boldsymbol{x})\right)^2, \tag{1}$$

which aims to find a good approximation of $f^\star$ from the student network. To ensure learning performance, $m \geq k$ is needed.

This problem is identified as an additive model in statistics (Stone, 1985; Hastie and Tibshirani, 1987), and it receives great attention in theoretical computer science (Chen et al., 2023) and machine learning theory, especially on sample/time complexity as well as training dynamics (Boursier and Flammarion, 2024; Bietti et al., 2023). However, understanding how gradient-based training algorithms recover the teacher network and analyzing the entire training dynamics are still challenging. Therefore, most current analyses are limited to non-gradient-based algorithms (Chen et al., 2023), or *local analysis* for gradient-based algorithms, which assumes that the loss has already been minimized below a very small threshold, or the angles between teacher neurons and their nearest student neurons are already small (called *strong recovery*), e.g., (Zhou et al., 2021). If we go beyond the local analysis, previous result on GD training can only handle specific cases, such as (Yehudai and Ohad, 2020) for $m = k = 1$,(Wu et al., 2018) for $m = k = 2$, and(Xu and Du, 2023; Chistikov et al., 2023) for $m \geq k = 1$. In fact, studying more general cases, such as $m, k = \mathcal{O}(1)$, remains unresolved, even in local analysis. Accordingly, we aim to address the following question:

**How can gradient descent recover teacher neurons and balance features beyond local analysis?**

To better understand the learning dynamics in the above question, we follow the "align then fit" framework (Maennel et al., 2018; Boursier and Flammarion, 2024), which also helps to explain the implicit bias of the learned solution. In this study, we run the gradient descent (GD) over Eq. (1). Since analyzing the entire training dynamics is still challenging and is an open problem, so we assume the *weak recovery*, where for each student neuron, exactly one teacher neuron exists that is not nearly perpendicular to it. Note that the weak recovery condition is still much weaker than the condition with local analysis and strong recovery that will be proved in our analysis. An informal version of our theoretical results is given as below.

**Theorem 1** (Global Convergence after Weak Recovery: Informal). *Under proper assumptions (e.g., teacher neurons are with same length $\|\boldsymbol{v}\|$, and orthogonal to each other), sufficiently small initialization with $\sigma = o(poly(d^{-\frac{1}{2}}))$, and trained via gradient descent with sufficiently small step-size $\eta = o(1)$ to minimize Eq. (1), after time $T^\star = \Omega(\frac{1}{\eta})$, for any $T \in \mathrm{N}$, we have:*

$$L(\boldsymbol{W}(T^\star + T)) \leq \mathcal{O}\left(\frac{\|\boldsymbol{v}\|^2}{\eta^3 T^3}\right), \quad \text{and} \quad \|\boldsymbol{w}_i(T^\star + T)\| = \Theta\left(\frac{k\|\boldsymbol{v}\|}{m}\right) \quad \forall i \in [m], w.h.p.$$

Our result demonstrates that the Eq. (1) can be solved by GD in the polynomial time to find the global minima and achieves the global convergence rate at $\mathcal{O}(1/T^3)$. We admit that the derived sample/time complexity[2] is not optimal, but to our knowledge, this is the first polynomial-time result of GD training beyond the local analysis for Eq. (1) with $m, k = \mathcal{O}(1)$. Besides, our results also indicate that the obtained solution will converge to a minimum "balanced" $\ell_2$ solution, where the "balanced" is determined by student neurons and their respective nearest teacher neurons.

**Technical challenges.** We employ the similar proof framework of Xu and Du (2023) on $m \geq k = 1$. The main challenge of this paper is how to address the coupling of different teacher neurons' influences on the student neurons, even though the teacher neurons are orthogonal to each other. For instance:

- In phase 1, single teacher neuron ($k = 1$) (Xu and Du, 2023) allows for monotonic convergence on the angular difference between the teacher and student neurons. However, this does not hold for $k > 1$. In this case, we use approximations of sine and cosine values for decoupling when the angle is very small or near perpendicular. Hence we can simplify the training dynamics and prove that the sine of the minimum angle converges linearly to a tiny neighborhood.

---

[2]In our setting, the number of training iterations $T$ corresponds to the sample size $n$, and the sample complexity to achieve an $\epsilon$ expected risk is $O(\epsilon^{-1/3}poly(m,k))$.

- In phase 2, during the analysis of neuron growth, the tangential components of the student neurons at each teacher neuron (and for more teacher neurons) are quite complex. Classical recursive relationship in (Xu and Du, 2023) can not handle this. Instead, we develop a new technical tool by building a dynamical system: we formulate the matrix iteration form, estimate the eigenvalues of the transition matrix, and establish the upper and lower bounds of such a dynamical system.

## 2 RELATED WORK

**Dynamics of gradient descent in the teacher-student setting**: Li and Yuan (2017) studied the exact-parameterized setting and proved convergence for SGD with initialization near identity. The separation between kernel methods and two-layer neural networks is further described in Li et al. (2020). To further understand the convergence and generalization of regression tasks using non-linear networks, it is essential to thoroughly analyze the dynamics throughout gradient-based training, commonly described as "align then fit" (Maennel et al., 2018; Boursier and Flammarion, 2024) in a three-phase analysis framework. Xu and Du (2023) provide a global convergence of learning with a single ReLU neuron, where the proof for the local convergence (i.e., the third phase) is given by Zhou et al. (2021). This analysis framework is also used in various settings, e.g., binary classification (Min et al., 2023) and matrix sensing (Xiong et al., 2024). Additionally, Zhou and Ge (2024) propose an algorithm combining gradient descent with convex optimization, achieving strong recovery in a single gradient step through specific initialization. Our work also analyzes the dynamics from weak recovery to strong recovery.

Besides, our problem can be cast as a special case of learning with multi-index model (Bietti et al., 2023) where the link function (i.e., the activation function used in this work) is unknown. However, the techniques are different and our three-phase analysis framework allows for a better understanding of global convergence. Some statistical physics studies work have explored related topics but differ from our work (Goldt et al., 2020; Arnaboldi et al., 2023) by focusing on generalization errors without providing convergence rates or detailed analyses of training dynamics and convergence phases.

**Implicit bias**: Recent studies suggest that gradient descent is implicitly biased towards a low-rank hidden weight matrix or a sparse number of directions represented by the neurons (Safran et al., 2022; Shevchenko et al., 2022; Chizat and Bach, 2020). This implicit bias is often characterized by the minimal norm interpolator, which is closely related to sparsely represented directions (Lyu and Li, 2020). These findings indicate that the early alignment phase enforces the alignment of weights towards a small number of directions, even with omnidirectional initialization, leading to implicit regularization at convergence (Boursier and Flammarion, 2023).

## 3 NOTATIONS, PROBLEM SETTING, AND ASSUMPTIONS

In this section, we give notations that are needed in this paper and then introduce our problem setting as well as the required assumptions in our proof.

### 3.1 NOTATIONS

*Basic notations:* We use the shorthand $[n] := \{1, 2, \ldots, n\}$ for a positive integer $n$. We denote by $a(n) \gtrsim b(n)$: the inequality $a(n) \geq cb(n)$ that hides a positive constant $c$ that is independent of $n$. Vectors (matrices) are denoted by boldface, lower-case (upper-case) letters. The used norm $\|\cdot\|$ in this paper is $\ell_2$ norm if we do not specify. We follow the standard Bachmann–Landau notation in complexity theory e.g., $\mathcal{O}$, $o$, $\Omega$, and $\Theta$ for order notation.

*Notations on angle:* The angle between any two non-zero vectors $\boldsymbol{w}$ and $\boldsymbol{v}$ is denoted as $\angle(\boldsymbol{w}, \boldsymbol{v}) := \cos^{-1} \frac{\langle \boldsymbol{w}, \boldsymbol{v} \rangle}{\|\boldsymbol{w}\|\|\boldsymbol{v}\|}$. Then we use the following notations for any $i, j \in [m], l \in [k]$

- $\theta_{il} \triangleq \angle(\boldsymbol{w}_i, \boldsymbol{v}_l)$: the angle between a student neuron $\boldsymbol{w}_i$ and a teacher neuron $\boldsymbol{v}_l$.

- $\varphi_{ij} \triangleq \angle(\boldsymbol{w}_i, \boldsymbol{w}_j)$: the angle between two neurons $\boldsymbol{w}_i$ and $\boldsymbol{w}_j$ for student model.

- $\tau_i \triangleq \arg\min_j \angle(\boldsymbol{w}_i(0), \boldsymbol{v}_j(0))$: the index of the teacher neuron with **the smallest angle** to the $\boldsymbol{w}_i$ at initialization, in which the smallest angle is denoted as $\theta_{i\star} \triangleq \theta_{i\tau_i} = \angle(\boldsymbol{w}_i, \boldsymbol{v}_{\tau_i})$.

- $\boldsymbol{r}_j \triangleq \sum_{i:\tau_i=l} \boldsymbol{w}_i - \boldsymbol{v}_l$: the difference of the teacher neuron $\boldsymbol{v}_l$ and the sum of the student neurons around $\boldsymbol{v}_l$.

For notational simplicity, by denoting $\bar{\boldsymbol{a}} \triangleq \frac{\boldsymbol{a}}{\|\boldsymbol{a}\|}$, we denote the tangential part $h_{il} \triangleq \langle \boldsymbol{w}_i, \bar{\boldsymbol{v}}_l \rangle$ as the projection of $\boldsymbol{w}_i$ along with the direction of $\boldsymbol{v}_l$; and a similar notation for $h_{i\star} \triangleq \langle \boldsymbol{w}_i, \bar{\boldsymbol{v}}_{\tau_i} \rangle$. Besides, we denote $\boldsymbol{w}_i(t)$ as the vector $\boldsymbol{w}_i$ at time $t$, which also adapts to $\theta_{ij}(t)$, etc.

*Notations on loss:* The standard Gaussian distribution is $\mathcal{N}(0, 1)$ with zero-mean and unit variance. We denote $\mathbb{E}_{\boldsymbol{x} \sim \mathcal{N}(0,1)}$ by $\mathbb{E}_{\boldsymbol{x}}$ for simplicity. By defining the residuals of the neural network as:

$$R(\boldsymbol{x}) := \sum_{i=1}^{m} \phi(\langle \boldsymbol{w}_i, \boldsymbol{x} \rangle) - \sum_{i=1}^{k} \phi(\langle \boldsymbol{v}_i, \boldsymbol{x} \rangle),$$

then the loss can be written as $L(\boldsymbol{W}) = \frac{1}{2} \mathbb{E}_{\boldsymbol{x}} R(\boldsymbol{x})^2$.

## 3.2 CLOSED FORM EXPRESSIONS OF GRADIENT OF LOSS: $\nabla L$

To make our paper self-contained, we present the closed-form expressions for $\nabla L$ when the input data follows a Gaussian distribution, as given by Safran and Shamir (2018), see the details in Appendix D. We denote $\nabla_i \triangleq \frac{\partial L(\boldsymbol{W})}{\partial \boldsymbol{w}_i}$ as the gradient of loss to the $\boldsymbol{w}_i$, when $\boldsymbol{w}_i \neq \boldsymbol{0}$. Then for any $i \in [m]$, the loss function is differentiable with gradient given by:

$$\nabla_i = \frac{1}{2} \sum_{j=1}^{m} \boldsymbol{w}_j - \frac{1}{2} \sum_{l=1}^{k} \boldsymbol{v}_l + \frac{1}{2\pi} \left[ \frac{\boldsymbol{w}_i}{\|\boldsymbol{w}_i\|} \left( \sum_{j=1, j\neq i}^{m} \sin \varphi_{ij} \|\boldsymbol{w}_j\| - \sum_{l=1}^{k} \sin \theta_{il} \|\boldsymbol{v}\| \right) - \sum_{j=1, j\neq i}^{m} \varphi_{ij} \boldsymbol{w}_j + \sum_{l=1}^{k} \theta_{il} \boldsymbol{v}_l \right].$$
(2)

We use random Gaussian initialization for neural network training, i.e., $\forall i \in [m], \boldsymbol{w}_i(0) \sim \mathcal{N}(\boldsymbol{0}, \sigma^2 \boldsymbol{I}_d)$ with the variance $\sigma^2$. Then we can prove that $\|\boldsymbol{w}_i\|$ has bounded norm with high probability if the dimension $d$ is not small, see Lemma 1 in Appendix D.

## 3.3 ASSUMPTIONS

We state the used assumptions in this paper.

**Assumption 1** (Weak recovery). *Regarding the angle $\theta_{ij}(0)$ defined before for any $i \in [m], j \in [k]$, at initialization, denote $\theta_{i\star}(0)$ as the smallest angle between $\boldsymbol{w}_i$ and its closet teacher neuron. The weak recovery assumes $\theta_{i\star}(0) \ll \theta_{ij}(0)$ with $j \in [k]$ and $j \neq \tau_i$. We mathematically formulate this as below.*

- *$\theta_{i\star}(0)$ is acute: $0 < \frac{\pi}{2} - \theta_{i\star}(0) \triangleq \zeta_i = \Theta(1)$, and $\zeta_i \in (0, \frac{\pi}{2}]$.*

- *$\theta_{ij}(0)$ is close to orthogonal: $\left| \frac{\pi}{2} - \theta_{ij}(0) \right| \leq \zeta = o(1)$ with $j \in [k]$ and $j \neq \tau_i$.*

**Remark:** The weak recovery assumption requires that a student neuron is not orthogonal to its closet teacher neuron but is nearly orthogonal to the remaining teacher neurons (Dandi et al., 2024). If we focus on the single ReLU case like Xu and Du (2023), this assumption can be directly removed because there is only one teacher neuron. With only one teacher neuron, there are no competing neurons for alignment, and thus the angle between the student and teacher neuron is naturally the smallest.

**Assumption 2** (Orthogonal and same norm for teacher neurons). *The teacher neurons are given by $\{\boldsymbol{v}_i\}_{i=1}^{k}$, and are assumed to be orthogonal to each other with the same norm, i.e., $\langle \boldsymbol{v}_i, \boldsymbol{v}_j \rangle = 0$ and $\|\boldsymbol{v}_i\| = \|\boldsymbol{v}_j\| = \|\boldsymbol{v}\|$, $\forall i \neq j$, $i, j \in [m]$. Clearly, we have $k \leq d$ due to the orthogonality of $k$ teacher neurons.*

**Remark:** This assumption requires all teacher neurons pointing to different (orthogonal) directions, which is important for identifiability or recovery. It aligns with practical considerations by allowing diverse tasks such that the target feature directions do not significantly overlap. This assumption as well as its variant (e.g., separation among teacher neurons) has been widely used in previous theoretical results, e.g., (Zhou et al., 2021; Oko et al., 2024; Simsek et al., 2023). We can relax this assumption where the teacher neurons are nearly orthogonal and have similar norms. However, such relaxation would require additional computations in our analysis. On the other hand, extending the analysis to the fully non-orthogonal case (arbitrary angles between teacher neurons) remains an open problem under our setting and would likely require new techniques. To avoid unnecessary complexity and focus on the core analysis, we concentrate on the basic assumptions.

**Assumption 3** (Balance condition at initialization). *At initialization, we record the number of student neurons $\boldsymbol{w}_i$ with $\tau_i = l$ as $m_l$. Then we assume $\frac{m}{3k} \leq m_l \leq \frac{3m}{k}, \forall l \in [k]$.*

**Remark:** This is a balance condition such that the number of merged student neurons among each teacher neuron is not extremely small or large. It is motivated by Boursier et al. (2022, Assumption 3) and (Wojtowytsch, 2020) that requires the student neurons to cover all directions of the teacher neurons. Our assumption requires student neurons coincide with teacher neurons in a *balanced* way.

## 4 MAIN RESULTS

In this section, we will provide the main theoretical results. First, in Section 4.1, we provide the primary result on global convergence. Then, in Section 4.2, we summarize the training dynamics across the three phases, highlighting their durations, key processes, and contributions to global convergence in a unified framework. Then, in the following subsections, we discuss the training dynamics of the three phases and provide proof sketches. In Section 4.3.1, we provide the main dynamics and final state results of the alignment process in the first phase. In Section 4.3.2, we provide the main dynamics and final state results of the tangential growth process in the second phase. In Section 4.3.3, we provide the results of the local convergence in the third phase and then achieve the final global convergence result.

### 4.1 MAIN THEOREM

**Theorem 2.** *Assume $d = \Omega(\log(m/\delta))$ with $\delta \in (0, 1)$, under Assumptions 1 2 and 3, let $\sigma = o(poly(m^{-k^2}, d^{-\frac{1}{2}})) = o(poly(d^{-\frac{1}{2}}))$, and trained via gradient descent with step-size, $\eta = o(poly(m^{-k^2})) = o(poly(1))$ to minimize Eq. (1), then there exists a $T^\star = \Omega(\frac{k \log k \log m}{m\eta}) = \Omega(\frac{1}{\eta})$ such that with probability at least $1 - \delta$ over the initialization, for any $T \in \mathbb{N}$, we have:*

$$L(\boldsymbol{W}(T^\star + T)) \leq \mathcal{O}\left(\frac{k^{12} \|\boldsymbol{v}\|^2}{\eta^3 T^3}\right), \quad \text{and} \quad \frac{\|\boldsymbol{v}\|}{4m_{\tau_i}} \leq \|\boldsymbol{w}_i(T^\star + T)\| \leq \frac{4 \|\boldsymbol{v}\|}{m_{\tau_i}} \quad \forall i \in [m].$$

**Remark**: Theorem 2 provides a convergence rate of $T^{-3}$, which is consistent with previous results (Xu and Du, 2023). Moreover, it indicates that the more teacher neurons and the larger their norms, the slower the convergence rate. This aligns with our intuition that when the initialization is very small, a larger norm and more teacher neurons require student neurons to take more time to align and converge to the teacher neurons. Unlike (Xu and Du, 2023), our results appear to be independent of the number of student neurons $m$, but rely on the choice of the initialization that implicitly depends on $m$. Furthermore, our results indicate that the student neurons will implicitly converge to a specific teacher neuron and maintain a balance among themselves.

### 4.2 SUMMARY OF MAIN RESULTS WITH PHASE ANALYSIS

We summarize the three training phases and associated dynamics in a unified framework to provide a clear overview of our findings. This structure highlights the processes that occur in each phase, their respective contributions to the global convergence result, and key observations. For a concise summary of the duration and key results, see Table 1 at the end of this section:

- **Phase 1 - Alignment:** In this phase, each student neuron aligns with a specific teacher neuron by minimizing the angle between their vectors. This process depends on the initialization scale ($\sigma$) and the learning rate ($\eta$). By the end of this phase, the angle between student and teacher neurons is reduced to $\mathcal{O}(\epsilon_1)$, ensuring proper alignment while maintaining bounded norms.

- **Phase 2 - Tangential Growth:** Following alignment, the student neurons grow tangentially along the direction of their aligned teacher neurons. During this phase, the projection strength between student neurons and teacher neurons balances, leading to a linear decrease in the loss. This phase ensures that the norms of student neurons become comparable to those of teacher neurons.

- **Phase 3 - Local Convergence:** In the final phase, the training achieves global convergence as the loss becomes sufficiently small. The loss decreases at a rate of $\mathcal{O}(T^{-3})$, and the

student neurons converge implicitly to the directions of their corresponding teacher neurons. This phase solidifies the final convergence result.

Table 1: Summary of Training Phases and Dynamics

| Phase | Duration | Key Results |
|---|---|---|
| Phase 1: Alignment | $T_1 = \Theta\left(\frac{\epsilon_1^2}{\eta}\right)$ | Angle reduced to $\mathcal{O}(\epsilon_1)$; norms remain bounded. |
| Phase 2: Tangential Growth | $T_2 = \Theta\left(\frac{1}{\eta} \ln\left(\frac{1}{\epsilon_2}\right)\right)$ | Projection strength balanced; loss decreases linearly. |
| Phase 3: Local Convergence | $T_3 = \mathcal{O}(T^{-3})$ | Loss decreases at $\mathcal{O}(T^{-3})$; neurons converge to teacher directions. |

## 4.3 PROOF OVERVIEW

In this section, we provide a sketch of the Theorem 2. The complete proof can be found in the appendix. Our proof is primarily divided into three phases: alignment (Section 4.3.1), tangential growth (Section 4.3.2), and global convergence (Section 4.3.3). Finally we can summarize the results in these three phases for the main result.

### 4.3.1 PHASE 1 - ALIGNMENT

During this phase, each student neuron individually aligns with a specific teacher neuron. The outcomes of this section are divided into two main parts: *i)* the upper and lower bounds on the lengths of the student neurons, as well as the angle between student and teacher neurons during the time Theorem 3. *ii)* the upper bound on the angle between student and teacher neurons at the end of phase 1, as well as the balance of projection strength from different student neurons onto the teacher neuron Corollary 1. The detailed derivation can be found in Appendix F.

**Theorem 3** (Phase 1: Alignment Process). *Assume $d = \Omega(\log(m/\delta))$ with $\delta \in (0, 1)$, for any $\epsilon_1 > 0$, under Assumption 1 with $10k\zeta \le \epsilon_1^2 = o(1)$ and Assumptions 2, 3 such that $\sigma = o(\frac{poly(\epsilon_1)\|\boldsymbol{v}\|}{\sqrt{d}})$ in our random Gaussian initialization, and the stepsize satisfies $\eta = o(\frac{\sigma\sqrt{d}\epsilon_1^2}{\|\boldsymbol{v}\|})$, then there exist a $T_1 = \Theta(\frac{\epsilon_1^2}{\eta})$, for $0 \le t \le T_1$, the following statements hold with probability at least $1 - \delta$:*

$$s_1 \le \|\boldsymbol{w}_i(t)\| \le s_2 + 2k\eta \|\boldsymbol{v}\| t, \quad \forall i \in [m], \quad \text{with } s_1 := \frac{1}{2}\sigma\sqrt{d}, \quad s_2 := 2\sigma\sqrt{d}, \quad (3)$$

*and*

$$\sin^2\left(\frac{\theta_{i^\star}(t)}{2}\right) - \epsilon_1^2 \le \left(1 + \frac{\eta k \|\boldsymbol{v}\| t}{s_2}\right)^{-\frac{1}{8k}} \left(\sin^2\left(\frac{\theta_{i^\star}(0)}{2}\right) - \epsilon_1^2\right), \quad \forall i \in [m]. \quad (4)$$

**Remark**: Our theorem implies that, during phase 1 of the training, the norm of each student neuron always has an immutable lower bound, while the upper bound increases linearly over time. Additionally, for each student neuron, the angle with its nearest teacher neuron converges linearly within an error range of $\epsilon_1^2$. Compared to the results of Xu and Du (2023), the upper bound of our neuron norm increases $k$ times faster because we have $k$ different teacher neurons, which naturally leads to this outcome. Taking $k = 1$, our convergence rate is faster by a constant factor compared to the results of Xu and Du (2023), and our condition for $\sigma$ is weaker. When the same $\sigma$ is selected, the total duration of phase 1, denoted as $T_1$, they are the same.

Then, we briefly introduce our proof technique, due to the presence of multiple teacher neurons, the gradient expression in Eq. (15) contains $2(k-1)$ cross terms including $\theta_{il}$ with detailed interactions, which do not exist in Xu and Du (2023). To handle this challenge, we provide additional analysis on alignment related to these cross terms in phase 1. Specifically, we prove these results by induction.

**Proof of Eq. (3)**: This formula provides the upper and lower bounds of $\|\boldsymbol{w}\|$ during the training. For the lower bound, based on the gradient expression $\nabla_i$ in Eq. (2), we prove $\langle \boldsymbol{w}_i(t), \nabla_i(t)\rangle \le 0$, which ensures that the norm $\|\boldsymbol{w}\|$ increases monotonically such that $\|\boldsymbol{w}_i(t)\| \ge s_1$. For the upper bound, we need to bound the norm of gradient using Eq. (2). Then, applying the triangle inequality, we can obtain the desired result.

**Proof of Eq. (4)**: This formula illustrates the angle dynamics (i.e., the alignment process) between the student neuron and its closest teacher neuron during phase 1. For larger $\theta_{i^\star}$, it is easy to prove

that the $\theta_{i^\star}$ decreases monotonically. However, when $\theta_{i^\star}$ is very small, we cannot guarantee the monotonic decreasing property of $\theta_{i^\star}$. To this end, we build the connection between $\sin^2(\theta_{i^\star}(t)/2)$ and the following angle difference

$$\cos(\theta_{i^\star}(t+1)) - \cos(\theta_{i^\star}(t)) := I_2 + I_3\,, \qquad \text{the first-order term } I_2 \text{ and the second-order term } I_3\,.$$

By estimating $I_2$ and $I_3$, we can track the dynamics of the angle difference and then prove that $\sin\theta_{i^\star}$ converges linearly to a very small neighborhood (i.e., $\epsilon_1^2$).

At the end of phase 1, we conclude the following result:

**Corollary 1** (Final State of Phase 1). *Under the same conditions as Theorem 3, at time $T_1$, the following statements hold with probability at least $1 - \delta$:*

$$\theta_{i^\star}(T_1) \leq 4\epsilon_1\,, \quad \text{and} \quad h_{i^\star}(T_1) \leq 2h_{j^\star}(T_1)\,, \quad \forall i, j \in [m]\,. \tag{5}$$

**Remark**: By the end of phase 1, each student neuron will align with its nearest teacher neuron with the residual angle at the order of $\mathcal{O}(\epsilon_1)$. Additionally, the projection lengths of these student neurons in the direction of their corresponding teacher neurons are relatively balanced, with a rough upper bound of 2.

**Proof of Eq. (5)**: For the first part, substituting the parameters from Theorem 3 into Eq. (4) will yield the result. For the second part, firstly, we derive the upper and lower bounds for $h_{i^\star}(t+1) - h_{i^\star}(t)$ and then accumulate these bounds. Next, we prove that before a certain time (e.g., $t := T_1/50$), the upper bound of $h_{i^\star}(t)$ is relatively small compared to this accumulated value. This allows us to establish the upper and lower bounds for all $h_{i^\star}(T_1)$ and thereby determine the maximum ratio of $h_{i^\star}(T_1)$ among different student neurons.

### 4.3.2 Phase 2 - Tangential Growth

In this section, we present the results of the second phase, in which each student neuron grows along the tangential direction of the teacher neuron aligned in phase 1 as below.The detailed derivation can be found in Appendix G.

**Theorem 4** (Phase 2: Tangential Growth Process). *Assume $d = \Omega(\log(m/\delta))$ with $\delta \in (0, 1)$, for any $\epsilon_1 > 0, \epsilon_2 > 0$, under Assumption 1 with $\epsilon_2 = o(1), \epsilon_1^2 = o(poly(\epsilon_2))$, Assumptions 2, 3 such that $\sigma = o(\frac{poly(\epsilon_1)\|\boldsymbol{v}\|}{\sqrt{d}})$ in our random Gaussian initialization, and the stepsize satisfies $\eta = o(\frac{\sigma\sqrt{d}\epsilon_1^2}{\|\boldsymbol{v}\|})$, then by setting there exist a $T_2 = T_1 + \Theta(\frac{1}{\eta}\ln(\frac{1}{\epsilon_2}))$, then $\forall T_1 \leq t \leq T_2$, we define $H_l(t) := \|\boldsymbol{v}\| - \sum_{i=1}^m \mathbb{I}_{\tau_i = l} h_{i^\star}(t)$ for $l \in [k]$, the following statements hold with probability at least $1 - \delta$:*

$$h_{i^\star}(t) \leq 2h_{j^\star}(t)\,, \quad \text{and} \quad \frac{2\|\boldsymbol{v}\|}{m_{\tau_i}} \geq h_{i^\star}(t) \geq \frac{s_1}{2}\,, \quad \forall i, j \in [m] \text{ and } \tau_i = \tau_j\,. \tag{6}$$

$$\left(1 - \frac{\eta m}{9k}\right)^{t-T_1}\|\boldsymbol{v}\| + 8\pi\epsilon_2\|\boldsymbol{v}\| \geq H_l(t) \geq \frac{2}{3}\|\boldsymbol{v}\|\left(1 - \frac{3\eta m}{2k}\right)^{t-T_1} - 8\pi\epsilon_2\|\boldsymbol{v}\| \geq 24\pi\epsilon_2\|\boldsymbol{v}\|\,, \forall l \in [k]\,, \tag{7}$$

*and*

$$\theta_{i^\star}(t) \leq \epsilon_2, \forall i \in [m]\,. \tag{8}$$

**Remark**: This theorem tells us that during phase 2:

1). The norm of student neurons close to the same teacher neuron remains relatively balanced, with each neuron having strict upper and lower bounds (Eq. (6)). It is worth noting that, unlike in phase one, see Eq. (5), this balance is not maintained for all neurons.

2). The projections of the student neurons near each teacher neuron will gradually increase, and the difference from $\|\boldsymbol{v}\|$ will approach zero at a linear convergence rate (Eq. (7)). This result implies that as training progresses, the loss gradually decreases. We will further prove that by the end of phase 2, the loss has decreased to a sufficiently small value.

3). The angle between each student neuron and its nearest teacher neuron stays within a small range (Eq. (8)). However, the angle is slightly larger than that of Phase 1 because additional cost/movement

is required to handle the convergence for tangential difference and the decrease of loss. For example, we have $\|\nabla_i(t)\| \le 2k\|\boldsymbol{v}\|$ in Phase 1 but it changes to $\|\nabla_i(t)\| \le 15k\|\boldsymbol{v}\|$ in Phase 2.

4). Taking $k = 1$, our condition for $\epsilon_2$ is similar to that of Xu and Du (2023), but we have relaxed the learning rate condition by a factor of $m$. And the total duration of phase 2, is reduced by a constant factor of $\frac{1}{2}$.

Then, we briefly introduce our proof technique. Compared to one teacher setting (Xu and Du, 2023), the tangential analysis requires a new dynamical system analysis regarding the dynamics of $\{H_l(t)\}_{l=1}^k$ due to the coupling tangential components among student/teacher neurons. Besides, the loss function becomes more complex Eq. (14) and we have to control the loss below a certain threshold in the presence of these interactions, which requires additional quantities to estimate. Specifically, we prove these results by induction.

**Proof of Eq. (6)**: For the first part, we follow the proof of Eq. (5) to build the connection between $h_{i^\star}(t+1) - h_{i^\star}(t)$ and $H_l$ in a weighted sum relationship, with an additional constant term $Q_i$. For two different student neurons close to the same teacher neuron, these weights are the same. By studying the changes of $\theta_{i^\star}$ and $\theta_{il}$ during this phase, $|Q_i(t)|$ will be bounded by a small quantity. Then we conclude the result by summing and combining the results with Eq. (5). For the second part, based on Eq. (7), we can derive $H_l \ge 0$ and finish the upper bound by combining the results from the first part. For the lower bound, we derive $h_{i^\star}(t+1) - h_{i^\star}(t) \ge 0$, which implies that $h_{i^\star}$ is monotonic increasing. Combining this with Eqs. (3) and (5), the proof is complete.

**Proof of Eq. (7)**: Using the above analysis about $h_{i^\star}(t+1) - h_{i^\star}(t) \ge 0$ and the relationship between $h_{i^\star}$ and $H_{\tau_i}$, we can establish a recursive relationship between $H(t+1)$ and $H(t)$ as well. Note that there is a coupling between different $\boldsymbol{H}$ and interference from small quantities $Q_i$, so we express the iterative formula in matrix form. To be specific, by denoting $\boldsymbol{H} := \{H_l\}_{l=1}^k$ (we write it in a matrix formulation), it admits the following recursive relationship:

$$\boldsymbol{H}(t+1) = \boldsymbol{A}\boldsymbol{H}(t) + \boldsymbol{Q}(t) \qquad \text{for a certain transition matrix } \boldsymbol{A} \text{ and } \boldsymbol{Q}(t) \text{ depends on } Q_i(t)\,.$$

By analyzing the eigenvalues of the transition matrix $\boldsymbol{A}$, we estimate the upper and lower bounds of such a dynamic system. For the small quantities $Q_i$, we adopt the same approach used in proving Eq. (4). Finally, we prove that $\boldsymbol{H}$ converges to a small value at a linear convergence rate.

**Proof of Eq. (8)**: The proof here is similar to Eq. (4), as it also analyzes the dynamics of $\cos\theta_{i^\star}$. However, the difficulty lies in that at this phase, the influence of $\boldsymbol{w}$ in the gradient is no longer negligible compared to $\boldsymbol{v}$, making the iterative relationship between angles more complex. First, by proving

$$\frac{\|\boldsymbol{w}_i(t)\|}{\|\boldsymbol{w}_j(t)\|} = \Theta(1)\,, \forall i, j \in [m],\ T_1 \le t \le T_2\,,$$

we are able to analyse the dynamics of $\cos\theta_{i^\star}$ (i.e., $I_2$ and $I_3$ in Eq. (4)) based on two cases $\tau_i = (\ne)\tau_j$. First, we use some properties of trigonometric functions to decouple this relationship so that it only involves the coupling between each student neuron and its nearest teacher neuron. Then, we estimate the difference $\sin^2\left(\frac{\theta_{i^\star}(t+1)}{2}\right) - \sin^2\left(\frac{\theta_{i^\star}(t)}{2}\right)$ for the final $\sin\theta_{i^\star}(t)$. Unlike in phase 1, here we obtain an upper bound for the linear growth of the angle $\theta_{i^\star}$. However, we can still prove that within the range of $T_2$, the angle remains small.

At the end of phase 2, we can draw the following results:

**Corollary 2** (Final state of Phase 2). *Under the same conditions as Theorem 4, at time $T_2$, the following statements hold with probability at least $1 - \delta$:*

$$\frac{\|\boldsymbol{v}\|}{3m_{\tau_i}} \le \|\boldsymbol{w}_i(T_2)\| \le \frac{3\|\boldsymbol{v}\|}{m_{\tau_i}},\ \forall i \in [m]\,, \quad and \quad L(\boldsymbol{W}(T_2)) \le \frac{1}{2}k^2\epsilon_2^{0.05}\|\boldsymbol{v}\|^2\,. \tag{9}$$

**Remark**: After phase 2, the norms of each student neuron have balanced, and the loss has decreased to a very small value. This provides the foundation for proving local convergence in phase three.

**Proof of first part of Eq. (9)**: We use the results in Theorem 4 to prove this result. For the lower bound, we first observe from Eq. (7) that $H_l$ is very small at time $T_2$, meaning the sum of $h$ among student neurons near each teacher neuron is close to $\|\boldsymbol{v}\|$, i.e., $H_l(T_2) \le \frac{\|\boldsymbol{v}\|}{3}$. Using the balance

of them in Eq. (6), we can then establish a lower bound for $h_{i^\star}(\|\boldsymbol{w}_i\| \cos \theta_{i^\star})$, which further allows us to derive a lower bound for $\|\boldsymbol{w}_i\|$. Similarly, for the upper bound, we first observe from Eq. (7) that at time $T_2$, the sum of $h$ among student neurons near each teacher neuron is close to but still less than $\|\boldsymbol{v}\|$. Using the balance of them in Eq. (6), we can then establish an upper bound for $h_{i^\star}(\|\boldsymbol{w}_i\| \cos \theta_{i^\star})$. Given that the angle between each student neuron and its nearest teacher neuron is very small (second part in Eq. (8)), we can further derive a lower bound for $\|\boldsymbol{w}_i\|$.

**Proof of second part of Eq. (9)**: The key point of this proof involves introducing an auxiliary function $g$ to help decompose the $L$. The loss $L$ can be expressed in the summation of $g$, see Appendix D for details. First, based on the upper bound of the angle in phase 2 (second part in Eq. (8)), we know that there are two scenarios for the angle in the closed form of the loss: close to 0 and nearly orthogonal. We discuss the upper and lower bounds of auxiliary function $g$ in these two cases. Then, according to Eq. (7), we find that at time $T_2$, the sum of the norms of the student neurons near each teacher neuron close to the norm of teacher neurons, i.e. $\sum_{i=1}^{m} \mathbb{I}_{\tau_i=l} h_{i^\star}(T_2) \geq (1 - o(1)) \|\boldsymbol{v}\|$. Combining these two results, we can derive an upper bound for the loss $L$.

### 4.3.3 PHASE 3 - LOCAL CONVERGENCE

In this section, we present the results of phase 3 - local convergence. Specifically, we show that when the loss is already small enough, the loss function converges to zero at a rate of $\mathcal{O}(\frac{1}{T^3})$ Theorem 5. Our results build upon the previous works of Xu and Du (2023); Zhou et al. (2021); Safran et al. (2021). The detailed derivation can be found in Appendix H.

**Theorem 5** (Local convergence). *Suppose the initial condition in Lemma 1 and Assumption 1 2 and 3 holds. If we set $\epsilon_2 = o(poly(1))$ and $\eta = o(1)$ in Theorem 4, then $\forall T \in \mathrm{N}$, the following statements hold with probability at least $1 - \delta$:*

$$L(\boldsymbol{W}(T + T_2)) \leq \frac{1}{\left( L(\boldsymbol{W}(T_2))^{-\frac{1}{3}} + \Omega\left( k^{-4} \|\boldsymbol{v}\|^{-\frac{2}{3}} \right) \eta T \right)^3}, \tag{10}$$

*and*

$$\frac{\|\boldsymbol{v}\|}{4 m_{\tau_i}} \leq \|\boldsymbol{w}_i(T + T_2)\| \leq \frac{4 \|\boldsymbol{v}\|}{m_{\tau_i}} \quad \forall i \in [m]. \tag{11}$$

**Remark**: This theorem shows that, under the condition that the loss is less than a very small value and the neurons remain balanced at the end of phase two, GD training can achieve the global minimum with a convergence rate of $\frac{1}{T^3}$. This result is consistent with the previous result in Xu and Du (2023) and is superior to $\frac{1}{T}$ in Zhou et al. (2021). Furthermore, this result also indicates that, without using regularization during training, every student neuron will implicitly converge to the directions of specific teacher neurons, and there is a balance among student neurons that converge to the direction of the same teacher neuron.

**Proof of Eq. (10)**: The proof framework of Theorem 5 is standard based on the local convergence analysis, e.g., (Zhou et al., 2021; Xu and Du, 2023). The key point is utilizing the result of classic optimization in Appendix H.4 and the lower bound of the gradient to satisfy the conditions of (Xu and Du, 2023, Lemma 24). First, we follow (Zhou et al., 2021) to derive several lemmas related to the properties of the loss function. Based on these lemmas, we can obtain the lower bound of the gradient in terms of the loss. Then, similar to Safran et al. (2021), we deduce that when the neurons maintain a certain balance, the loss is locally smooth. This allows us to directly apply the classic optimization theory conclusion regarding the relationship between adjacent iterations of gradient descent Appendix H.4. Finally, we build the final convergence result by Xu and Du (2023, Lemma 24). Additionally, our proof requires that the balance condition of the neurons is consistently maintained Eq. (11), which can be proven using induction and convergence results alternately.

Finally, by combining results from Sections 4.3.1 to 4.3.3 with the hyper-parameter selection in Appendix B, we obtain the global convergence result in Theorem 2.

When $k = 1$, compared to the results of Xu and Du (2023), our paper needs stronger requirements on $\sigma$, $\eta$ and time. This is due to the upper bound of the loss after phase 2 in Eq. (9) and its relationship with $\epsilon_2$. Due to multiple teacher neurons, the number of student neurons converging to each teacher neuron directions are different. This leads to different norms for the student neurons, which makes a looser upper bound. However, in the case of $k = 1$, such handling is not necessary. Therefore, our

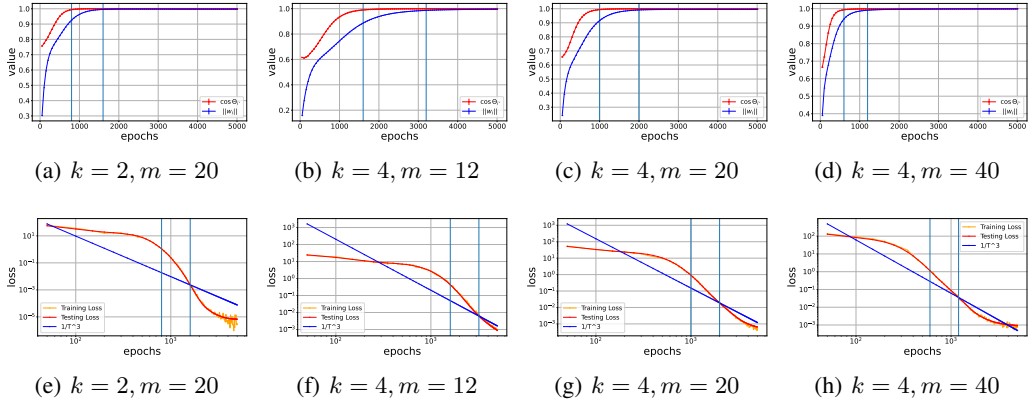

Figure 1: Convergence curves for different $m$ and $k$.

results can cover the results of Xu and Du (2023) with only minor modifications and have a better constant factor in phase 1 discussed before.

## 5 NUMERICAL VALIDATION

In this section, we empirically validate our theoretical results by plotting the convergence curves under the following setting: we set $\|v\| = 5$, data dimension $d = 100$, batch size of 512, and a total of 5000 batches. The total number of training samples (equivalent to the previously mentioned $T^\star + T$) is $2.56 \times 10^6$. Besides, we have added a $1/T^3$ reference line in the log-log plot for better comparison. We selected four sets of parameters $k$ and $m$: $k = 2, m = 20$, $k = 4, m = 12$, $k = 4, m = 20$, and $k = 4, m = 40$ with initialization variance $\sigma = 10^{-6}$ and learning rate $\eta = 5 \times 10^{-4}$. The plots in Fig. 1 show the cosine of the angle and norm convergence during training (top row) and the log-log plot of the loss during training (bottom row) for different values of $k$ and $m$. The results show that larger $k$ values lead to longer $t_1$ and $t_2$ and slower convergence rates, while larger m values result in shorter $t_1$ and $t_2$ but have little effect on the convergence rate. This matches our theoretical results such that using more (student) neurons decreases the time for alignment. We admit that learning more (teacher) neurons generally requires more time but this is given under the same initialization strategy. Instead, our initialization strategy depends on $m$ and $k$, leading to different learning dynamics.

Regarding the timescale experiments, we divided the training dynamics into three phases for analysis. We can observe the clear "align then fit" phenomena where in phases 1 and 2, the angle aligns and the tangential grows until the norm of neurons' weights is unchanged. In phase 3, the loss function decreases for fitting data. The phase transition from Phase 1 to 2 is not very clear in the experiments but can still be observed with a distinct difference in that Phase 2 finishes later than Phase 1. Nonetheless, we have marked the figure's approximate endpoints of the first and second phases.

## 6 CONCLUSION

Our three-phase analysis framework provides a comprehensive analysis on global convergence, i.e., 1) *alignment*: the angle decreases $\theta_{i^\star}(T_1) \leq \mathcal{O}(\epsilon_1)$ satisfying the balance condition but the norm of student neuron gradually increases with $T_1$; 2) *tangential growth*: the projection of the student neurons near teacher neurons gradually increases. The angle is still small but slightly larger than that of phase 1 due to the additional cost of handling the convergence of tangential difference; 3) local convergence: the loss is close to zero and the neurons are still well-balanced thus achieving the global convergence at the rate of $\mathcal{O}(T^{-3})$.

One potential drawback of this work is the weak recovery which simplifies the analysis. However, without weak recovery, the analysis will be quite complex, remaining unsolved, and thus we leave it as future work.

ACKNOWLEDGEMENTS

We are also thankful to the reviewers for providing constructive feedback. This work was supported by Hasler Foundation Program: Hasler Responsible AI (project number 21043). Research was sponsored by the Army Research Office and was accomplished under Grant Number W911NF-24-1-0048. This project has received funding from the European Research Council (ERC) under the European Union's Horizon 2020 research and innovation programme (grant agreement n° 725594 - time-data).

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

APPENDIX INTRODUCTION

The Appendix is organized as follows:

- In Appendix A, we provide a comprehensive introduction to the notations used in this paper.
- In Appendix B, we discuss the selection of hyperparameters in this paper.
- In Appendix C, we provide some additional numerical results.
- In Appendix D, we provide a detailed explanation of the closed-form expression for the loss and its gradient as mentioned in the main text.
- In Appendix E, we provide a detailed analysis of Assumption 1.
- In Appendix F, we present the main results of phase 1 along with detailed proofs.
- In Appendix G, we present the main results of phase 2 along with detailed proofs.
- In Appendix H, we present the main results of phase 3 along with detailed proofs.

## A  SYMBOLS AND NOTATION

Table 2: Core symbols and notations used in this paper.

| Symbol | Dimension(s) | Definition |
|---|---|---|
| $\mathcal{N}(\mu, \sigma^2)$ | - | Gaussian distribution with mean $\mu$ and variance $\sigma^2$ |
| $[n]$ | - | Shorthand for $\{1, 2, \ldots, n\}$ |
| $\mathcal{O}, o, \Omega, \Theta$ | - | Bachmann–Landau asymptotic notation |
| $\|\boldsymbol{v}\|_2$ | - | Euclidean norm of vector $\boldsymbol{v}$ |
| $\|\boldsymbol{M}\|_2$ | - | Spectral norm of matrix $\boldsymbol{M}$ |
| $\|\boldsymbol{M}\|_{\mathrm{F}}$ | - | Frobenius norm of matrix $\boldsymbol{M}$ |
| $\boldsymbol{v}_i$ | $\mathbb{R}^d$ | Weight vector of the $i$-th teacher neuron |
| $\boldsymbol{w}_i$ | $\mathbb{R}^d$ | Weight vector of the $i$-th student neuron |
| $\langle \boldsymbol{u}, \boldsymbol{v} \rangle$ | - | Dot product of vectors $\boldsymbol{u}$ and $\boldsymbol{v}$ |
| $T, T_1, T_2$ | - | Total training time and durations of different phases |
| $\bar{\boldsymbol{a}}$ | $\mathbb{R}^d$ | Normalized vector: $\bar{\boldsymbol{a}} = \frac{\boldsymbol{a}}{\|\boldsymbol{a}\|}$ |
| $h_{il}$ | $\mathbb{R}$ | Projection of $\boldsymbol{w}_i$ in the direction of $\boldsymbol{v}_l$ |
| $\phi(\cdot)$ | - | ReLU activation function |
| $\nabla_i$ | $\mathbb{R}^d$ | Gradient of loss with respect to $\boldsymbol{w}_i$ |
| $\tau_i$ | - | Index of teacher neuron closest to $\boldsymbol{w}_i$ at initialization |
| $\zeta_i$ | - | Angular offset between $\boldsymbol{w}_i$ and nearest teacher neuron |
| $\zeta$ | - | Angular offset between $\boldsymbol{w}_i$ and other teacher neurons |
| $R(\boldsymbol{x})$ | $\mathbb{R}$ | Residual of the network output |
| $m_l$ | - | Number of student neurons close to teacher neuron $\boldsymbol{v}_l$ |
| $\boldsymbol{r}_j$ | $\mathbb{R}^d$ | Difference between $\boldsymbol{v}_j$ and sum of nearby student neurons |
| $\theta_{il}$ | - | Angle between $\boldsymbol{w}_i$ and $\boldsymbol{v}_l$ |
| $\varphi_{ij}$ | - | Angle between $\boldsymbol{w}_i$ and $\boldsymbol{w}_j$ |
| $L(\boldsymbol{W})$ | - | Loss function: $L(\boldsymbol{W}) = \frac{1}{2}\mathbb{E}_{\boldsymbol{x}} R(\boldsymbol{x})^2$ |
| $d$ | - | Input dimension of the data |
| $m$ | - | Number of student neurons |
| $k$ | - | Number of teacher neurons |
| $\eta$ | - | Learning rate for gradient descent |
| $\sigma$ | - | Initialization variance of student neurons |

## B  SELECTION OF HYPER-PARAMETERS

- We set $\epsilon_2 = o(m^{-60}k^{-100}) = o(\mathrm{poly}(1))$ in Theorem 4 as required by Theorem 5.
- We set $\epsilon_1^2 = o(\epsilon_2^{\Theta(k)}/m) = o(\mathrm{poly}(\epsilon_2))$ in Theorem 3 as required by Theorem 4.

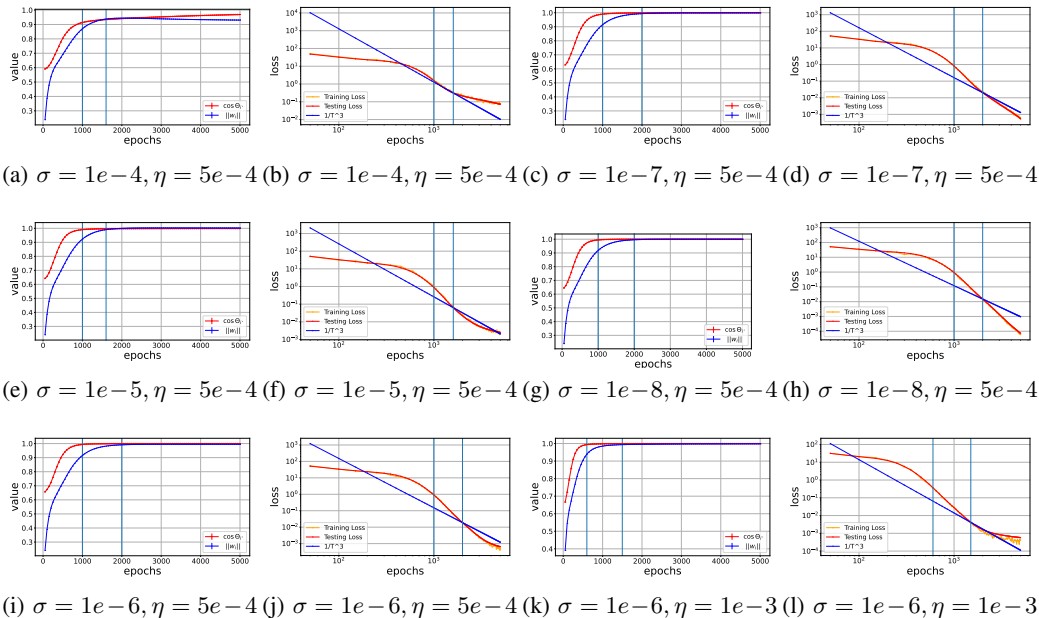

(a) $\sigma = 1e-4, \eta = 5e-4$ (b) $\sigma = 1e-4, \eta = 5e-4$ (c) $\sigma = 1e-7, \eta = 5e-4$ (d) $\sigma = 1e-7, \eta = 5e-4$

(e) $\sigma = 1e-5, \eta = 5e-4$ (f) $\sigma = 1e-5, \eta = 5e-4$ (g) $\sigma = 1e-8, \eta = 5e-4$ (h) $\sigma = 1e-8, \eta = 5e-4$

(i) $\sigma = 1e-6, \eta = 5e-4$ (j) $\sigma = 1e-6, \eta = 5e-4$ (k) $\sigma = 1e-6, \eta = 1e-3$ (l) $\sigma = 1e-6, \eta = 1e-3$

Figure 2: Convergence curves for different $\sigma$ and $\eta$.

- We set $\sigma \leq \frac{\epsilon_1^{16k+2}\|\boldsymbol{v}\|}{10000m\sqrt{d}} = o(\frac{\text{poly}(\epsilon_1^2)\|\boldsymbol{v}\|}{\sqrt{d}})$ in Theorem 3 as required by Theorems 3 and 4.

- We set $\eta = o\left(\frac{m\epsilon_1^2 s_1^2}{k^2\|\boldsymbol{v}\|^2}\right) \leq o\left(\frac{\epsilon_1^{32k+6}}{mk}\right) = o(\text{poly}(\epsilon_1^2))$ in Theorem 3 as required by Theorem 4.

- We set $T_1 := \frac{\epsilon_1^2}{100\eta km} = \Theta(\frac{\epsilon_1^2}{\eta})$ in Theorem 3.

- We set $T_2 = T_1 + \frac{k}{2\eta m}\ln\left(\frac{1}{48\pi\epsilon_2}\right) = \Theta(\frac{1}{\eta}\ln\epsilon_2^{-1}) = \Omega(\frac{1}{\eta})$ in Theorem 4.

## C  ADDITIONAL NUMERICAL VALIDATION

In this section, we selected four sets of parameters $\sigma$ and $\eta$: $\sigma = 10^{-4}, \eta = 5 \times 10^{-4}, \sigma = 10^{-5}, \eta = 5 \times 10^{-4}, \sigma = 10^{-6}, \eta = 5 \times 10^{-4}, \sigma = 10^{-7}, \eta = 5 \times 10^{-4}$ and $\sigma = 10^{-8}, \eta = 10^{-3}$ with $k = 4$ and $m = 20$. The plots in Fig. 2 show the cosine of the angle and norm convergence during training (top row) and the log-log plot of the loss during training (bottom row) for different values of $\sigma$ and $\eta$.

The results demonstrate that smaller initialization variances are crucial for ensuring the training dynamics in the first two phases align with theoretical predictions, thereby facilitating faster and more predictable convergence in the third phase. In contrast, larger initialization variances, while potentially shortening the duration of the first two phases, disrupt the balance of the student neuron norms by the end of the second phase, leading to slower convergence rates in the third phase. This underscores the importance of adopting smaller initialization variances and stepsizes to maintain stable training dynamics across all phases and achieve consistency with the predicted $T^{-3}$ convergence rate.

## D  EXPRESSION OF LOSS $L$ AND ITS GRADIENT $\nabla L$

In this section, we provide a detailed explanation of the closed-form expression for the loss and its gradient as mentioned in the main text. The main content of this section follows (Safran and Shamir, 2018, Section 4.1.1). Besides, the bounded norm of $\|\boldsymbol{w}_i\|$ for any $i \in [m]$ is also given in this subsection. We include these results here just for completeness.

For notational simplicity, we introduce the following auxiliary function:

$$g(\boldsymbol{a}, \boldsymbol{b}) := \mathbb{E}_{\boldsymbol{x}}[\phi(\boldsymbol{a}^\top \boldsymbol{x})\phi(\boldsymbol{b}^\top \boldsymbol{x})] = \frac{\|\boldsymbol{a}\| \|\boldsymbol{b}\|}{2\pi}\left(\sin\angle(\boldsymbol{a}, \boldsymbol{b}) + (\pi - \angle(\boldsymbol{a}, \boldsymbol{b}))\cos\angle(\boldsymbol{a}, \boldsymbol{b})\right), \quad (12)$$

which implies that

- if $\boldsymbol{a}$ and $\boldsymbol{b}$ are orthogonal, i.e., $\langle \boldsymbol{a}, \boldsymbol{b}\rangle = 0$, then $g(\boldsymbol{a}, \boldsymbol{b}) = \frac{\|\boldsymbol{a}\|\|\boldsymbol{b}\|}{2\pi}$.
- If $\boldsymbol{a} = \boldsymbol{b}$, then $g(\boldsymbol{a}, \boldsymbol{b}) = \frac{1}{2}\|\boldsymbol{a}\| \|\boldsymbol{b}\| = \frac{1}{2}\|\boldsymbol{a}\|^2 = \frac{1}{2}\|\boldsymbol{b}\|^2$.

Then we can derive that the gradient for $g(\boldsymbol{a}, \boldsymbol{b})$ w.r.t $\boldsymbol{a}$ as follows:

$$g'(\boldsymbol{a}, \boldsymbol{b}) = \frac{\partial g(\boldsymbol{a}, \boldsymbol{b})}{\partial \boldsymbol{a}} = \frac{1}{2\pi}\left(\|\boldsymbol{b}\|\sin\angle(\boldsymbol{a}, \boldsymbol{b})\frac{\boldsymbol{a}}{\|\boldsymbol{a}\|} + (\pi - \angle(\boldsymbol{a}, \boldsymbol{b}))\boldsymbol{b}\right). \quad (13)$$

Using this auxiliary function, we can rewrite the loss function in Eq. (1) as the following form:

$$\begin{aligned}
L(\boldsymbol{W}) &= \frac{1}{2}\mathbb{E}_{\boldsymbol{x} \sim \mathcal{N}(0,1)}\left(\sum_{i=1}^m \phi(\boldsymbol{w}_i^\top \boldsymbol{x}) - \sum_{i=1}^k \phi(\boldsymbol{v}_i^\top \boldsymbol{x})\right)^2 \\
&= \frac{1}{2}\sum_{i=1}^m\sum_{j=1}^m g(\boldsymbol{w}_i, \boldsymbol{w}_j) + \frac{1}{2}\sum_{i=1}^k\sum_{j=1}^k g(\boldsymbol{v}_i, \boldsymbol{v}_j) - \sum_{i=1}^m\sum_{j=1}^k g(\boldsymbol{w}_i, \boldsymbol{v}_j).
\end{aligned} \quad (14)$$

Accordingly, when $\boldsymbol{w}_i \neq \boldsymbol{0}$. for $\forall i \in [n]$, the loss function is differentiable with gradient given by:

$$\begin{aligned}
\nabla_i &:= \frac{\partial L(\boldsymbol{W})}{\partial \boldsymbol{w}_i} \\
&= \sum_{j=1, j\neq i}^m \frac{\partial g(\boldsymbol{w}_i, \boldsymbol{w}_j)}{\partial \boldsymbol{w}_i} + \frac{1}{2}\frac{\partial g(\boldsymbol{w}_i, \boldsymbol{w}_i)}{\partial \boldsymbol{w}_i} - \sum_{l=1}^k \frac{\partial g(\boldsymbol{w}_i, \boldsymbol{v}_l)}{\partial \boldsymbol{w}_i} \\
&= \frac{\boldsymbol{w}_i}{2} + \frac{1}{2\pi}\sum_{j=1, j\neq i}^m\left(\|\boldsymbol{w}_j\|\sin\varphi_{ij}\frac{\boldsymbol{w}_i}{\|\boldsymbol{w}_i\|} + (\pi - \varphi_{ij})\boldsymbol{w}_j\right) - \frac{1}{2\pi}\sum_{l=1}^k\left(\|\boldsymbol{v}\|\sin\theta_{il}\frac{\boldsymbol{w}_i}{\|\boldsymbol{w}_i\|} + (\pi - \theta_{il})\boldsymbol{v}_l\right) \\
&= \frac{1}{2}\sum_{j=1}^m \boldsymbol{w}_j - \frac{1}{2}\sum_{l=1}^k \boldsymbol{v}_l + \frac{1}{2\pi}\left[\frac{\boldsymbol{w}_i}{\|\boldsymbol{w}_i\|}\left(\sum_{j=1, j\neq i}^m \sin\varphi_{ij}\|\boldsymbol{w}_j\| - \sum_{l=1}^k \sin\theta_{il}\|\boldsymbol{v}\|\right) - \sum_{j=1, j\neq i}^m \varphi_{ij}\boldsymbol{w}_j + \sum_{l=1}^k \theta_{il}\boldsymbol{v}_l\right].
\end{aligned} \quad (15)$$

The bounded norm $\|\boldsymbol{w}_i\|$ at initialization can be given as below.

**Lemma 1** (Adapted from Lemma 3 in (Xu and Du, 2023)). *Let $s_1 := \frac{1}{2}\sigma\sqrt{d}$. $s_2 := 2\sigma\sqrt{d}$, if $d = \Omega(\log(m/\delta))$, with probability at least $1 - \delta$, the following properties hold at the initialization:*

$$s_1 \leq \|\boldsymbol{w}_i(0)\| \leq s_2, \qquad \forall i \in [m].$$

**Remark:** This is a standard fact in high-dimensional statistics, and on this basis, our result only involves this randomness. In the rest of the analysis in this paper is deterministic.

## E   DETAILED ANALYSIS FOR ASSUMPTION 1

Here we prove the following lemma:

**Lemma 2.** *When $d = \Omega(\frac{\log(mk/\delta))}{\zeta^2})$ with $\zeta = o(poly(m^{-k^2}, k^{-k^2}))$, then with probability at least $1 - \delta$, the following property hold at the initialization:*

$$\frac{\pi}{2} - \zeta \leq \theta_{ij}(0) \leq \frac{\pi}{2} + \zeta. \qquad \forall i \in [m], j \in [k].$$

*Proof.* According to Lemma 1, we have for $\forall i \in [m], j \in [k]$, we have:

$$|\langle \boldsymbol{w}_i(0), \bar{\boldsymbol{v}}_j \rangle| \leq \frac{\zeta}{4} \sigma \sqrt{d} \wedge \|\boldsymbol{w}_i(0)\| \geq \frac{1}{2} \sigma \sqrt{d} \Rightarrow |\cos \theta_{ij}(0)| \leq \frac{\zeta}{2} \Rightarrow \frac{\pi}{2} - \zeta \leq \theta_{ij}(0) \leq \frac{\pi}{2} + \zeta.$$

By concentration inequality of Gaussian, we have:

$$\mathbb{P}\left( |\langle \boldsymbol{w}_i(0), \bar{\boldsymbol{v}}_j \rangle| \geq \frac{\zeta}{4} \sigma \sqrt{d} \right) \leq 2 \exp\left( -\frac{(\frac{\zeta}{4} \sigma \sqrt{d})^2}{2\sigma^2} \right) \leq \frac{\delta}{3mk}.$$

Then:

$$\mathbb{P}\left( \theta_{ij}(0) \geq \frac{\pi}{2} + \zeta \vee \theta_{ij}(0) \leq \frac{\pi}{2} - \zeta \forall j \in [k] \right) \leq \frac{\delta}{3mk} * k + \frac{\delta}{3m} = \frac{2\delta}{3m}.$$

Applying the union bound for $\forall i \in [m]$, which finishes the proof. $\qquad\square$

## F   GLOBAL CONVERGENCE: PHASE 1 (ALIGNMENT)

In Phase 1, we are interested in the dynamics of $\theta_{i^\star}$ as well as the angle difference between the student neuron and its closest teacher neuron. The theorem we prove below is a combination of Theorem 3 and Corollary 1 from the main text.

**Theorem 6** (Phase 1: Alignment). *Assume $d = \Omega(\log(m/\delta))$ with $\delta \in (0,1)$, for any $\epsilon_1 > 0$, under Assumption 1 with $10k\zeta \leq \epsilon_1^2 = o(\zeta_i^3)$ and Assumptions 2, 3 such that $\sigma \leq \frac{\epsilon_1^{16k+2} \|\boldsymbol{v}\|}{10000m\sqrt{d}}$ in our random Gaussian initialization, and the stepsize satisfies $\eta \leq \frac{\sigma\sqrt{d}\epsilon_1^2}{100k^2\|\boldsymbol{v}\|}$, then by setting $T_1 := \frac{\epsilon_1^2}{100\eta km}$, for $0 \leq t \leq T_1$, the following statements hold with probability at least $1 - \delta$:*

$$s_1 \leq \|\boldsymbol{w}_i(t)\| \leq s_2 + 2k\eta\|\boldsymbol{v}\|t, \quad \forall i \in [m], \quad \text{with } s_1 := \frac{1}{2}\sigma\sqrt{d}, \quad s_2 := 2\sigma\sqrt{d}, \tag{16}$$

*and*

$$\sin^2\left(\frac{\theta_{i^\star}(t)}{2}\right) - \epsilon_1^2 \leq \left(1 + \frac{\eta k \|\boldsymbol{v}\| t}{s_2}\right)^{-\frac{1}{8k}} \left(\sin^2\left(\frac{\theta_{i^\star}(0)}{2}\right) - \epsilon_1^2\right), \quad \forall i \in [m]. \tag{17}$$

*After Phase 1, we have:*

$$\theta_{i^\star}(T_1) \leq 4\epsilon_1, \quad \forall i \in [m]. \tag{18}$$

*and*

$$h_{i^\star}(T_1) \leq 2h_{j^\star}(T_1), \quad \forall i,j \in [m]. \tag{19}$$

*Proof.* The proof is given by induction. We firstly prove Eqs. (16) and (17) and then Eqs. (18) and (19).

At the initialization time $t = 0$, Eq. (16) and Eq. (17) directly hold according to Lemma 1. Note that the probability in this work only relates to the random initialization as given by Lemma 1. For description simplicity, we do not include this probability during the derivation but just mention it in our theorem.

Before proving Eqs. (16) and (17), we first analyse the learning dynamics of $\theta_{i^\star}$. For any $\forall i \in [m]$ and $0 < t < T_1$, according to the inductive hypothesis, we have:

$$\sin^2\left(\frac{\theta_{i^\star}(t)}{2}\right) \leq \max\left\{\sin^2\left(\frac{\theta_{i^\star}(0)}{2}\right), \epsilon_1^2\right\} = \sin^2\left(\frac{\theta_{i^\star}(0)}{2}\right) = \sin^2\left(\frac{\pi}{4} - \frac{\zeta_i}{2}\right),$$

where the right part of the above inequality is given by the following fact with Assumption 1:

$$\sin^2\left(\frac{\theta_{i^\star}(0)}{2}\right) = \sin^2\left(\frac{\pi}{4} - \frac{\zeta_i}{2}\right) = \frac{1}{2} + \frac{1}{2}\sin(\zeta_i) = \Theta(1) \geq \epsilon_1^2 = o(\zeta_i^3) = o(1),$$

which means:

$$\theta_{i^\star}(t) \le \frac{\pi}{2} - \zeta_i \,. \tag{20}$$

Then we assume Eqs. (16) and (17) hold for any $0 < t < T_1$ to prove Eqs. (16) and (17) for $t + 1$.

**Proof of right part of Eq. (16)**:

According to the inductive hypothesis and $s_2 := 2\sigma\sqrt{d}$ in Lemma 1, we have:

$$\|\boldsymbol{w}_i(t)\| \le s_2 + 2k\eta \|\boldsymbol{v}\| T_1 \le \frac{\epsilon_1^{16k+2} \|\boldsymbol{v}\|}{50m} + \frac{\epsilon_1^2 \|\boldsymbol{v}\|}{50m} \le \frac{\epsilon_1^2 \|\boldsymbol{v}\|}{48m} = o\left(\frac{\|\boldsymbol{v}\|}{m}\right) \le \frac{\|\boldsymbol{v}\|}{3m}, \quad \forall i \in [m] \,. \tag{21}$$

That means the teacher neuron's norm controls all of the student neurons' norm at $t \in [0, T_1]$. Then by triangle inequality and Eq. (21), the gradient norm can be upper bounded by

$$
\begin{aligned}
&\|\nabla_i(t)\| \\
&\le \left\| \frac{1}{2} \sum_{j=1}^m \boldsymbol{w}_j(t) \right\| + \left\| \frac{1}{2} \sum_{l=1}^k \boldsymbol{v}_l \right\| \\
&+ \left\| \frac{1}{2\pi} \left[ \frac{\boldsymbol{w}_i(t)}{\|\boldsymbol{w}_i(t)\|} \left( \sum_{j=1,j\neq i}^m \sin\varphi_{ij}(t) \|\boldsymbol{w}_j(t)\| - \sum_{l=1}^k \sin\theta_{il}(t) \|\boldsymbol{v}\| \right) - \sum_{j=1,j\neq i}^m \varphi_{ij}(t)\boldsymbol{w}_j(t) + \sum_{l=1}^k \theta_{il}(t)\boldsymbol{v}_l(t) \right] \right\| \\
&\le \frac{m}{2} \times \frac{\|\boldsymbol{v}\|}{3m} + \frac{k}{2} \|\boldsymbol{v}\| + \frac{1}{2\pi} \left( m \times \frac{\|\boldsymbol{v}\|}{3m} + k \|\boldsymbol{v}\| + m\pi \times \frac{\|\boldsymbol{v}\|}{3m} + k\pi \|\boldsymbol{v}\| \right) \\
&< 2k \|\boldsymbol{v}\|, \quad \forall i \in [m] \,.
\end{aligned}
\tag{22}
$$

One can see that, the gradient norm is upper bounded by all of the teacher neuron's norm. Accordingly, based on the gradient iteration, by the above results, we have:

$$\|\boldsymbol{w}_i(t+1)\| = \|\boldsymbol{w}_i(t) - \eta\nabla_i(t)\| \le \|\boldsymbol{w}_i(t)\| + \|\eta\nabla_i(t)\| \le s_2 + 2k\eta \|\boldsymbol{v}\| (t+1), \quad \forall i \in [m], \tag{23}$$

which concludes the proof.

**Proof of left part of Eq. (16)**:

Here we need to prove the lower bound, we have:

$$\|\boldsymbol{w}_i(t+1)\| \ge \|\boldsymbol{w}_i(t)\| \ge s_1, \quad \forall i \in [m] \,.$$

According to the gradient iteration:

$$\|\boldsymbol{w}_i(t+1)\|^2 - \|\boldsymbol{w}_i(t)\|^2 = \|\boldsymbol{w}_i(t) - \eta\nabla_i(t)\|^2 - \|\boldsymbol{w}_i(t)\|^2 = -2\eta \langle \boldsymbol{w}_i(t), \nabla_i(t) \rangle + \eta^2 \|\nabla_i(t)\|^2, \quad \forall i \in [m],$$

we only need to prove $\forall i \in [m]$, $\langle \boldsymbol{w}_i(t), \nabla_i(t) \rangle \le 0$. To be specific, we split $\langle \boldsymbol{w}_i(t), \nabla_i(t) \rangle$ into two parts:

$$\langle \boldsymbol{w}_i(t), \nabla_i(t) \rangle$$

$$= \left\langle \boldsymbol{w}_i(t), \frac{1}{2} \sum_{j=1}^m \boldsymbol{w}_j(t) - \frac{1}{2} \sum_{l=1}^k \boldsymbol{v}_l \right\rangle$$

$$+ \left\langle \boldsymbol{w}_i(t), \frac{1}{2\pi} \left[ \frac{\boldsymbol{w}_i(t)}{\|\boldsymbol{w}_i(t)\|} \left( \sum_{j=1, j\neq i}^m \sin \varphi_{ij}(t) \|\boldsymbol{w}_j(t)\| - \sum_{l=1}^k \sin \theta_{il}(t) \|\boldsymbol{v}\| \right) - \sum_{j=1, j\neq i}^m \varphi_{ij}(t) \boldsymbol{w}_j(t) + \sum_{l=1}^k \theta_{il}(t) \boldsymbol{v}_l \right] \right\rangle$$

$$= \frac{1}{2} \sum_{j=1}^m \langle \boldsymbol{w}_i(t), \boldsymbol{w}_j(t) \rangle - \frac{1}{2} \sum_{l=1}^k \langle \boldsymbol{w}_i(t), \boldsymbol{v}_l \rangle$$

$$+ \frac{1}{2\pi} \|\boldsymbol{w}_i(t)\| \sum_{j=1, j\neq i}^m \sin \varphi_{ij}(t) \|\boldsymbol{w}_j(t)\| - \frac{1}{2\pi} \|\boldsymbol{w}_i(t)\| \sum_{l=1}^k \sin \theta_{il}(t) \|\boldsymbol{v}\|$$

$$- \frac{1}{2\pi} \sum_{j=1, j\neq i}^m \varphi_{ij}(t) \langle \boldsymbol{w}_i(t), \boldsymbol{w}_j(t) \rangle + \frac{1}{2\pi} \sum_{l=1}^k \theta_{il}(t) \langle \boldsymbol{w}_i(t), \boldsymbol{v}_l \rangle$$

$$= \frac{1}{2} \|\boldsymbol{w}_i(t)\| \left( \sum_{j=1}^m \|\boldsymbol{w}_j(t)\| \cos \varphi_{ij}(t) - \sum_{l=1}^k \|\boldsymbol{v}\| \cos \theta_{il}(t) \right.$$

$$+ \frac{1}{\pi} \sum_{j=1, j\neq i}^m \|\boldsymbol{w}_j(t)\| \sin \varphi_{ij}(t) - \frac{1}{\pi} \sum_{l=1}^k \|\boldsymbol{v}\| \sin \theta_{il}(t)$$

$$\left. - \frac{1}{\pi} \sum_{j=1, j\neq i}^m \|\boldsymbol{w}_j(t)\| \cos \varphi_{ij}(t) \varphi_{ij}(t) + \frac{1}{\pi} \sum_{l=1}^k \|\boldsymbol{v}\| \cos \theta_{il}(t) \theta_{il}(t) \right)$$

$$= \frac{1}{2\pi} \|\boldsymbol{w}_i(t)\| \|\boldsymbol{v}\| \sum_{l=1}^k \left( \underbrace{-\pi \cos \theta_{il}(t) - \sin \theta_{il}(t) + \cos \theta_{il}(t) \theta_{il}(t)}_{I_1} \right)$$

$$+ \frac{1}{2\pi} \|\boldsymbol{w}_i(t)\| \sum_{j=1}^m \|\boldsymbol{w}_j(t)\| \left( \underbrace{\pi \cos \varphi_{ij}(t) + \sin \varphi_{ij}(t) - \cos \varphi_{ij}(t) \varphi_{ij}(t)}_{\tilde{I}_1} \right),$$

where the last equality holds by including the additional term related to $\varphi_{ii} = 0$ for any $i \in [m]$.

One hand, for $I_1$, by Eq. (17), $I_1$ is a monotonically increase function of $\theta_{il}(t)$ on the interval $[0, \pi]$. Then by Eq. (20), we have $\theta_{il}(t) \leq \theta_{i^\star}(t) + \frac{\pi}{2} \leq \pi - \zeta_i$, which implies that:

$$\begin{aligned} I_1 &= -\pi \cos \theta_{il}(t) - \sin \theta_{il}(t) + \cos \theta_{il}(t) \theta_{il}(t) \\ &\leq -\pi \cos(\pi - \zeta_i) - \sin(\pi - \zeta_i) + \cos(\pi - \zeta_i)(\pi - \zeta_i) \\ &= \zeta_i \cos(\zeta_i) - \sin(\zeta_i) \\ &\leq -\frac{\zeta_i^3}{4}, \end{aligned}$$

where the last inequality holds by the fact that $\zeta_i \cos(\zeta_i) - \sin(\zeta_i) \leq -\frac{\zeta_i^3}{4}$ is always true on the interval $[0, \frac{\pi}{2}]$.

On the other hand, to estimate $\tilde{I}_1$, recall $\|\boldsymbol{w}_i(t)\| \leq \frac{\epsilon_1^2 \|\boldsymbol{v}\|}{48m} = o(1)$ in Eq. (21) and the fact $|\tilde{I}_1| \leq \pi$, we have:

$$\frac{1}{2\pi} \|\boldsymbol{w}_i(t)\| \sum_{j=1}^m \|\boldsymbol{w}_j(t)\| \tilde{I}_1 \leq \frac{1}{2\pi} \|\boldsymbol{w}_i(t)\| \sum_{j=1}^m \|\boldsymbol{w}_j(t)\| |\tilde{I}_1| \leq \frac{1}{96} \|\boldsymbol{w}_i(t)\| \|\boldsymbol{v}\| \epsilon_1^2.$$

Accordingly, combining the above derivation over $I_1$ and $\tilde{I}_1$, we have:

$$\langle \boldsymbol{w}_i(t), \nabla_i(t) \rangle \leq \frac{1}{2\pi} \|\boldsymbol{w}_i(t)\| \|\boldsymbol{v}\| \left( -\frac{k\zeta_i^3}{4} + \Theta(\epsilon_1^2) \right) \leq 0 \,,$$

due to $\epsilon_1^2 = o(\zeta_i^3)$, we conclude that $\|\boldsymbol{w}_i(t+1)\| \geq \|\boldsymbol{w}_i(t)\| \geq s_1$ and finish the proof for Eq. (16).

**Proof of Eq. (17)**:

We analyze the learning dynamics of $\cos\theta_{i^\star}$ by splitting it into two parts (first-order term and the second-order term) as follows:

$$
\begin{aligned}
&\cos\theta_{i^\star}(t+1) - \cos\theta_{i^\star}(t) \\
&= \frac{\langle \boldsymbol{w}_i(t+1), \boldsymbol{v}_{\tau_i} \rangle}{\|\boldsymbol{w}_i(t+1)\| \|\boldsymbol{v}\|} - \frac{\langle \boldsymbol{w}_i(t), \boldsymbol{v}_{\tau_i} \rangle}{\|\boldsymbol{w}_i(t)\| \|\boldsymbol{v}\|} \\
&= \frac{\|\boldsymbol{w}_i(t)\| \langle \boldsymbol{w}_i(t+1), \boldsymbol{v}_{\tau_i} \rangle - \|\boldsymbol{w}_i(t+1)\| \langle \boldsymbol{w}_i(t), \boldsymbol{v}_{\tau_i} \rangle}{\|\boldsymbol{w}_i(t+1)\| \|\boldsymbol{w}_i(t)\| \|\boldsymbol{v}\|} \\
&= \frac{\|\boldsymbol{w}_i(t)\| \langle \boldsymbol{w}_i(t) - \eta\nabla_i(t), \boldsymbol{v}_{\tau_i} \rangle - \|\boldsymbol{w}_i(t+1)\| \langle \boldsymbol{w}_i(t), \boldsymbol{v}_{\tau_i} \rangle}{\|\boldsymbol{w}_i(t+1)\| \|\boldsymbol{w}_i(t)\| \|\boldsymbol{v}\|} \\
&= \frac{(\|\boldsymbol{w}_i(t)\| - \|\boldsymbol{w}_i(t+1)\|) \langle \boldsymbol{w}_i(t), \boldsymbol{v}_{\tau_i} \rangle - \|\boldsymbol{w}_i(t)\| \langle \eta\nabla_i(t), \boldsymbol{v}_{\tau_i} \rangle}{\|\boldsymbol{w}_i(t+1)\| \|\boldsymbol{w}_i(t)\| \|\boldsymbol{v}\|} \\
&= \frac{\left( \frac{\|\boldsymbol{w}_i(t)\|^2 - \|\boldsymbol{w}_i(t+1)\|^2}{\|\boldsymbol{w}_i(t)\| + \|\boldsymbol{w}_i(t+1)\|} \right) \langle \boldsymbol{w}_i(t), \boldsymbol{v}_{\tau_i} \rangle - \|\boldsymbol{w}_i(t)\| \langle \eta\nabla_i(t), \boldsymbol{v}_{\tau_i} \rangle}{\|\boldsymbol{w}_i(t+1)\| \|\boldsymbol{w}_i(t)\| \|\boldsymbol{v}\|} \\
&= \frac{\left( \frac{2\eta\langle \boldsymbol{w}_i(t), \nabla_i(t) \rangle - \eta^2 \|\nabla_i(t)\|^2}{\|\boldsymbol{w}_i(t)\| + \|\boldsymbol{w}_i(t+1)\|} \right) \langle \boldsymbol{w}_i(t), \boldsymbol{v}_{\tau_i} \rangle - \eta \|\boldsymbol{w}_i(t)\| \langle \nabla_i(t), \boldsymbol{v}_{\tau_i} \rangle}{\|\boldsymbol{w}_i(t+1)\| \|\boldsymbol{w}_i(t)\| \|\boldsymbol{v}\|} \\
&= \underbrace{\frac{\eta}{\|\boldsymbol{w}_i(t+1)\|} \langle \langle \bar{\boldsymbol{w}}_i(t), \bar{\boldsymbol{v}}_{\tau_i} \rangle \bar{\boldsymbol{w}}_i(t) - \bar{\boldsymbol{v}}_{\tau_i}, \nabla_i(t) \rangle}_{I_2} \\
&\quad + \underbrace{\frac{\eta \langle \bar{\boldsymbol{w}}_i(t), \bar{\boldsymbol{v}}_{\tau_i} \rangle}{\|\boldsymbol{w}_i(t+1)\|} \left( \frac{\langle \bar{\boldsymbol{w}}_i(t), \nabla_i(t) \rangle (\|\boldsymbol{w}_i(t)\| - \|\boldsymbol{w}_i(t+1)\|) - \eta \|\nabla_i(t)\|^2}{\|\boldsymbol{w}_i(t+1)\| + \|\boldsymbol{w}_i(t)\|} \right)}_{I_3} \,.
\end{aligned}
\tag{24}
$$

One can see that we need to estimate the respective two parts $I_2$ and $I_3$. For term $I_2$, note that $\langle \langle \bar{\boldsymbol{w}}_i(t), \bar{\boldsymbol{v}}_{\tau_i} \rangle \bar{\boldsymbol{w}}_i(t) - \bar{\boldsymbol{v}}_{\tau_i}, \boldsymbol{w}_i(t) \rangle = 0$, then we have:

$$I_2 = \frac{\eta}{\|\boldsymbol{w}_i(t+1)\|} \left\langle \left\langle \bar{\boldsymbol{w}}_i(t), \bar{\boldsymbol{v}}_{\tau_i} \right\rangle \bar{\boldsymbol{w}}_i(t) - \bar{\boldsymbol{v}}_{\tau_i}, \nabla_i(t) \right\rangle$$

$$= \frac{\eta}{\|\boldsymbol{w}_i(t+1)\|} \left( \left\langle \left\langle \bar{\boldsymbol{w}}_i(t), \bar{\boldsymbol{v}}_{\tau_i} \right\rangle \bar{\boldsymbol{w}}_i(t) - \bar{\boldsymbol{v}}_{\tau_i}, \frac{1}{2} \sum_{j=1}^{m} \boldsymbol{w}_j(t) - \frac{1}{2} \sum_{l=1}^{k} \boldsymbol{v}_l \right\rangle \right.$$

$$\left. + \left\langle \left\langle \bar{\boldsymbol{w}}_i(t), \bar{\boldsymbol{v}}_{\tau_i} \right\rangle \bar{\boldsymbol{w}}_i(t) - \bar{\boldsymbol{v}}_{\tau_i}, \frac{1}{2\pi} \left[ - \sum_{j=1,j\neq i}^{m} \varphi_{ij}(t)\boldsymbol{w}_j(t) + \sum_{l=1}^{k} \theta_{il}(t)\boldsymbol{v}_l \right] \right\rangle \right)$$

$$= \frac{\eta}{\|\boldsymbol{w}_i(t+1)\|} \left( \frac{1}{2} \left\langle \left\langle \bar{\boldsymbol{w}}_i(t), \bar{\boldsymbol{v}}_{\tau_i} \right\rangle \bar{\boldsymbol{w}}_i(t) - \bar{\boldsymbol{v}}_{\tau_i}, \sum_{j=1}^{m} \boldsymbol{w}_j(t) \right\rangle - \frac{1}{2} \left\langle \left\langle \bar{\boldsymbol{w}}_i(t), \bar{\boldsymbol{v}}_{\tau_i} \right\rangle \bar{\boldsymbol{w}}_i(t) - \bar{\boldsymbol{v}}_{\tau_i}, \sum_{l=1}^{k} \boldsymbol{v}_l \right\rangle \right.$$

$$\left. - \frac{1}{2\pi} \left\langle \left\langle \bar{\boldsymbol{w}}_i(t), \bar{\boldsymbol{v}}_{\tau_i} \right\rangle \bar{\boldsymbol{w}}_i(t) - \bar{\boldsymbol{v}}_{\tau_i}, \sum_{j=1,j\neq i}^{m} \varphi_{ij}(t)\boldsymbol{w}_j(t) \right\rangle + \frac{1}{2\pi} \left\langle \left\langle \bar{\boldsymbol{w}}_i(t), \bar{\boldsymbol{v}}_{\tau_i} \right\rangle \bar{\boldsymbol{w}}_i(t) - \bar{\boldsymbol{v}}_{\tau_i}, \sum_{l=1}^{k} \theta_{ij}(t)\boldsymbol{v}_l \right\rangle \right)$$

$$= \frac{\eta}{2\pi \|\boldsymbol{w}_i(t+1)\|} \left( \left\langle \cos\theta_{i\star}(t)\bar{\boldsymbol{w}}_i(t) - \bar{\boldsymbol{v}}_{\tau_i}, \sum_{j=1}^{m} (\pi - \varphi_{ij}(t))\boldsymbol{w}_j(t) \right\rangle - \left\langle \cos\theta_{i\star}(t)\bar{\boldsymbol{w}}_i(t) - \bar{\boldsymbol{v}}_{\tau_i}, \sum_{l=1}^{k} (\pi - \theta_{il}(t))\boldsymbol{v}_l \right\rangle \right)$$

$$= \frac{\eta}{2\pi \|\boldsymbol{w}_i(t+1)\|} \left( \left\langle \cos\theta_{i\star}(t)\bar{\boldsymbol{w}}_i(t), \sum_{j=1}^{m} (\pi - \varphi_{ij}(t))\boldsymbol{w}_j(t) \right\rangle - \left\langle \bar{\boldsymbol{v}}_{\tau_i}, \sum_{j=1}^{m} (\pi - \varphi_{ij}(t))\boldsymbol{w}_j(t) \right\rangle \right.$$

$$\left. - \left\langle \cos\theta_{i\star}(t)\bar{\boldsymbol{w}}_i(t), \sum_{l=1}^{k} (\pi - \theta_{il}(t))\boldsymbol{v}_l \right\rangle + \left\langle \bar{\boldsymbol{v}}_{\tau_i}, \sum_{l=1}^{k} (\pi - \theta_{il}(t))\boldsymbol{v}_l \right\rangle \right)$$

$$= \frac{\eta}{2\pi \|\boldsymbol{w}_i(t+1)\|} \left( \sum_{j=1}^{m} \left( \|\boldsymbol{w}_j(t)\| (\pi - \varphi_{ij}(t)) \cos\theta_{i\star}(t) \cos\varphi_{ij}(t) \right) - \sum_{j=1}^{m} \left( \|\boldsymbol{w}_j(t)\| (\pi - \varphi_{ij}(t)) \cos\theta_{j\tau_i}(t) \right) \right.$$

$$\left. - \sum_{l=1,l\neq\tau_i}^{k} \left( \|\boldsymbol{v}\| (\pi - \theta_{il}(t)) \cos\theta_{i\star}(t) \cos\theta_{il}(t) \right) + \|\boldsymbol{v}\| \sin^2\theta_{i\star}(t)(\pi - \theta_{i\star}(t)) \right)$$

$$\geq \frac{\eta \|\boldsymbol{v}\|}{2\pi \|\boldsymbol{w}_i(t+1)\|} \left( \sin^2\theta_{i\star}(t)(\pi - \theta_{i\star}(t)) - \sum_{l=1,l\neq\tau_i}^{k} \left( (\pi - \theta_{il}(t)) \cos\theta_{i\star}(t) \cos\theta_{il}(t) \right) - \frac{\pi}{12}\epsilon_1^2 \right) \quad \text{[Eq. (21)]}$$

$$\geq \frac{\eta \|\boldsymbol{v}\|}{2\pi \|\boldsymbol{w}_i(t+1)\|} \left( \frac{\pi}{2} \sin^2\theta_{i\star}(t) - 2k\pi\zeta - \frac{\pi}{12}\epsilon_1^2 \right)$$

$$\geq \frac{\eta \|\boldsymbol{v}\|}{4 \|\boldsymbol{w}_i(t+1)\|} \left( \sin^2\theta_{i\star}(t) - \frac{17}{30}\epsilon_1^2 \right),$$

$$(25)$$

which builds the connection between $I_2$ and $\sin^2\theta_{i\star}(t)$. For term $I_3$:

$$I_3 = \frac{\eta \langle \bar{\boldsymbol{w}}_i(t), \bar{\boldsymbol{v}}_{\tau_i} \rangle}{\|\boldsymbol{w}_i(t+1)\|} \left( \frac{\langle \bar{\boldsymbol{w}}_i(t), \nabla_i(t) \rangle (\|\boldsymbol{w}_i(t)\| - \|\boldsymbol{w}_i(t+1)\|) - \eta \|\nabla_i(t)\|^2}{\|\boldsymbol{w}_i(t+1)\| + \|\boldsymbol{w}_i(t)\|} \right)$$

$$\geq -\frac{\eta}{\|\boldsymbol{w}_i(t+1)\|} \left( \frac{\|\nabla_i(t)\| \|\eta\nabla_i(t)\| + \eta \|\nabla_i(t)\|^2}{\|\boldsymbol{w}_i(t+1)\| + \|\boldsymbol{w}_i(t)\|} \right) \quad \text{[using Eq. (23)]}$$

$$\geq -\frac{\eta}{\|\boldsymbol{w}_i(t+1)\|} \left( \frac{\|\nabla_i(t)\| \|\eta\nabla_i(t)\| + \eta \|\nabla_i(t)\|^2}{2s_1} \right) \quad \text{[using Eq. (16)]} \qquad (26)$$

$$= -\frac{\eta^2 \|\nabla_i(t)\|^2}{s_1 \|\boldsymbol{w}_i(t+1)\|}$$

$$\geq -\frac{4k^2\eta^2 \|\boldsymbol{v}\|^2}{s_1 \|\boldsymbol{w}_i(t+1)\|} . \qquad \text{[using Eq. (22)]}$$

Take Eq. (25) and Eq. (26) into Eq. (24), we have:

$$\cos\theta_{i^\star}(t+1) - \cos\theta_{i^\star}(t) = I_2 + I_3$$

$$\geq \frac{\eta\|\boldsymbol{v}\|}{4\|\boldsymbol{w}_i(t+1)\|}\left(\sin^2\theta_{i^\star}(t) - \frac{17}{30}\epsilon_1^2 - \frac{16k^2\eta\|\boldsymbol{v}\|}{s_1}\right)$$

$$\geq \frac{\eta\|\boldsymbol{v}\|}{4\|\boldsymbol{w}_i(t+1)\|}\left(\sin^2\theta_{i^\star}(t) - \frac{133}{150}\epsilon_1^2\right)$$

$$\geq \frac{\eta\|\boldsymbol{v}\|}{4\|\boldsymbol{w}_i(t+1)\|}\left(\sin^2\theta_{i^\star}(t) - \epsilon_1^2\right).$$

Accordingly, we transform the dynamics analysis on $\theta_{i^\star}$ from $\cos$ to $\sin$, which allows for estimating Eq. (17) as below. Recall $\cos 2x = 1 - 2\sin^2 x$, the above inequality implies:

$$
\begin{aligned}
&\sin^2\left(\frac{\theta_{i^\star}(t)}{2}\right) - \sin^2\left(\frac{\theta_{i^\star}(t+1)}{2}\right) \\
&= \frac{\cos\theta_{i^\star}(t+1) - \cos\theta_{i^\star}(t)}{2} \\
&\geq \frac{\eta\|\boldsymbol{v}\|}{8\|\boldsymbol{w}_i(t+1)\|}\left(\sin^2\theta_{i^\star}(t) - \epsilon_1^2\right) \\
&= \frac{\eta\|\boldsymbol{v}\|}{8\|\boldsymbol{w}_i(t+1)\|}\left(4\sin^2\left(\frac{\theta_{i^\star}(t)}{2}\right)\cos^2\left(\frac{\theta_{i^\star}(t)}{2}\right) - \epsilon_1^2\right) \\
&\geq \frac{\eta\|\boldsymbol{v}\|}{4\|\boldsymbol{w}_i(t+1)\|}\left(\sin^2\left(\frac{\theta_{i^\star}(t)}{2}\right) - \epsilon_1^2\right),
\end{aligned}
\tag{27}
$$

which implies:

$$
\begin{aligned}
\sin^2\left(\frac{\theta_{i^\star}(t+1)}{2}\right) - \epsilon_1^2 &\leq \left(1 - \frac{\eta\|\boldsymbol{v}\|}{4\|\boldsymbol{w}_i(t+1)\|}\right)\left(\sin^2\left(\frac{\theta_{i^\star}(t)}{2}\right) - \epsilon_1^2\right) \\
&\leq \left(1 - \frac{\eta\|\boldsymbol{v}\|}{4(s_2 + 2\eta k\|\boldsymbol{v}\|(t+1))}\right)\left(\sin^2\left(\frac{\theta_{i^\star}(t)}{2}\right) - \epsilon_1^2\right) \quad \text{[using Eq. (16)]} \\
&\leq \prod_{u=1}^{t+1}\left(1 - \frac{\eta\|\boldsymbol{v}\|}{4(s_2 + 2\eta k\|\boldsymbol{v}\|u)}\right)\left(\sin^2\left(\frac{\theta_{i^\star}(0)}{2}\right) - \epsilon_1^2\right) \\
&\leq \exp\left(\int_{u=1}^{t+2} -\frac{\eta\|\boldsymbol{v}\|}{4(s_2 + 2\eta k\|\boldsymbol{v}\|u)}\mathrm{d}u\right)\left(\sin^2\left(\frac{\theta_{i^\star}(0)}{2}\right) - \epsilon_1^2\right) \quad \text{[using } 1 - x \leq e^{-x}] \\
&= \exp\left(-\frac{1}{8k}\ln\left(\frac{s_2 + 2\eta k\|\boldsymbol{v}\|(t+2)}{s_2 + 2\eta k\|\boldsymbol{v}\|}\right)\right)\left(\sin^2\left(\frac{\theta_{i^\star}(0)}{2}\right) - \epsilon_1^2\right) \\
&\leq \left(1 + \frac{\eta k\|\boldsymbol{v}\|(t+1)}{s_2}\right)^{-\frac{1}{8k}}\left(\sin^2\left(\frac{\theta_{i^\star}(0)}{2}\right) - \epsilon_1^2\right).
\end{aligned}
$$

Accordingly, we finish the proof of Eq. (17).

**Proof of Eq. (18)**:

Let $t_0 := \frac{T}{50} \in \mathbb{N}$, for any $t \in [t_0, T_1]$, using Eq. (17) and definitions of $s_2, \sigma$, we have:

$$\sin^2\left(\frac{\theta_{i^\star}(t)}{2}\right) - \epsilon_1^2 \leq \left(1 + \frac{\eta k \|\boldsymbol{v}\| t}{s_2}\right)^{-\frac{1}{8k}} \left(\sin^2\left(\frac{\theta_{i^\star}(0)}{2}\right) - \epsilon_1^2\right)$$

$$\leq \left(1 + \frac{\eta k \|\boldsymbol{v}\| t}{s_2}\right)^{-\frac{1}{8k}}$$

$$\leq \left(\frac{\eta k \|\boldsymbol{v}\| t_0}{s_2}\right)^{-\frac{1}{8k}}$$

$$= \left(\frac{\eta k \|\boldsymbol{v}\| T_1}{100\sigma\sqrt{d}}\right)^{-\frac{1}{8k}}$$

$$\leq \left(\frac{\|\boldsymbol{v}\| \epsilon_1^2}{10000 m \sigma\sqrt{d}}\right)^{-\frac{1}{8k}}$$

$$\leq \epsilon_1^2.$$

That means: $\sin^2\left(\frac{\theta_{i^\star}(t))}{2}\right) \leq 2\epsilon_1^2$. So $\forall t \in [\frac{T_1}{50}, T_1]$ and $\forall i \in [m]$, we have $\theta_{i^\star}(t) \leq 4\epsilon_1$. Consequently, each student neuron has aligned to a teacher neuron by the end of phase 1.

**Proof of Eq. (19)**: For any $t \in [T_1/50, T_1]$, we study the dynamics of $h_{i^\star}$ (i.e., the inner product between the projection of gradient and teacher neuron) admitting the following formulation:

$$h_{i^\star}(t+1) - h_{i^\star}(t)$$
$$= \langle \boldsymbol{w}_i(t+1), \bar{\boldsymbol{v}}_{\tau_i} \rangle - \langle \boldsymbol{w}_i(t), \bar{\boldsymbol{v}}_{\tau_i} \rangle$$
$$= - \eta \langle \nabla_i(t), \bar{\boldsymbol{v}}_{\tau_i} \rangle$$
$$= - \frac{\eta}{2} \left\langle \sum_{j=1}^m \boldsymbol{w}_j(t) - \boldsymbol{v}_{\tau_i}, \bar{\boldsymbol{v}}_{\tau_i} \right\rangle$$
$$- \frac{\eta}{2\pi} \left\langle \frac{\boldsymbol{w}_i(t)}{\|\boldsymbol{w}_i(t)\|} \left( \sum_{j=1,j\neq i}^m \sin\varphi_{ij}(t) \|\boldsymbol{w}_j(t)\| - \sum_{l=1}^k \sin\theta_{il}(t) \|\boldsymbol{v}\| \right) - \sum_{j=1,j\neq i}^m \varphi_{ij}(t)\boldsymbol{w}_j(t) + \theta_{i^\star}(t)\boldsymbol{v}_{\tau_i}, \bar{\boldsymbol{v}}_{\tau_i} \right\rangle$$
$$= \frac{\eta}{2} \left( \|\boldsymbol{v}\| - \sum_{j=1}^m h_{j\tau_i}(t) \right)$$
$$- \frac{\eta}{2\pi} \left( \cos\theta_{i^\star}(t) \left( \sum_{j=1,j\neq i}^m \sin\varphi_{ij}(t) \|\boldsymbol{w}_j(t)\| - \sum_{l=1}^k \sin\theta_{il}(t) \|\boldsymbol{v}\| \right) - \sum_{j=1,j\neq i}^m \varphi_{ij}(t)h_{j\tau_i}(t) + \theta_{i^\star}(t) \|\boldsymbol{v}\| \right).$$
$$(28)$$

To analyse this dynamics, we need to study the $\sin\theta_{il}(t)$ at first. According to Assumption 2, we have:

$$\frac{\pi}{2} - \theta_{i^\star}(t) \leq \theta_{il}(t) \leq \frac{\pi}{2} + \theta_{i^\star}(t), \quad \forall i \in [m], \tau_i \neq l \in [k].$$

So we have:

$$-\theta_{i^\star}(t) \leq \frac{\pi}{2} - \theta_{il}(t) \leq \theta_{i^\star}(t), \quad \forall i \in [m], \tau_i \neq l \in [k].$$

That is:

$$1 \geq \sin\theta_{il}(t) = \cos\left(\frac{\pi}{2} - \theta_{il}(t)\right) \geq \cos\theta_{i^\star}(t) \geq \cos(4\epsilon_1) \geq 1 - 8\epsilon_1^2, \quad \forall i \in [m], \tau_i \neq l \in [k].$$

Then taking it back to Eq. (28), we have:

$$
\begin{aligned}
h_{i^\star}(t+1) - h_{i^\star}(t) &\leq \frac{\eta}{2}\|\boldsymbol{v}\| - \frac{\eta}{2\pi}\left(\cos\theta_{i^\star}(t)\left(-\sum_{l=1}^{k}\sin\theta_{il}(t)\|\boldsymbol{v}\|\right) - \sum_{j=1,j\neq i}^{m}\varphi_{ij}(t)h_{j\tau_i}(t)\right) \\
&\leq \frac{\eta}{2}\|\boldsymbol{v}\| + \frac{\eta}{2\pi}\left(\sum_{l=1}^{k}\sin\theta_{il}(t)\|\boldsymbol{v}\| + \sum_{j=1,j\neq i}^{m}\varphi_{ij}(t)\|\boldsymbol{w}_j(t)\|\right)\left[\text{using }\cos\theta_{i^\star}(t)\leq 1\right] \\
&\leq \frac{\eta}{2}\|\boldsymbol{v}\| + \frac{\eta}{2\pi}\left(k\|\boldsymbol{v}\| + (m-1)\pi\frac{\epsilon_1^2\|\boldsymbol{v}\|}{48m}\right) \qquad \left[\text{using Eq. (21) and }\varphi_{ij}<\pi\right] \\
&\leq \frac{k+\pi-0.5}{2\pi}\eta\|\boldsymbol{v}\|, \quad \forall i\in[m].
\end{aligned}
$$

Similarly, we can derive that:

$$
h_{i^\star}(t+1) - h_{i^\star}(t) \geq \frac{k+\pi-1.5}{2\pi}\eta\|\boldsymbol{v}\|, \quad \forall i\in[m].
$$

Then, we accumulate over the time:

$$
\frac{49(k+\pi-1.5)}{100\pi}\eta T_1\|\boldsymbol{v}\| \leq h_{i^\star}(T_1) - h_{i^\star}\left(\frac{T_1}{50}\right) \leq \frac{49(k+\pi-0.5)}{100\pi}\eta T_1\|\boldsymbol{v}\|, \quad \forall i\in[m]. \tag{29}
$$

The remaining thing left is to bound $h_{i^\star}\left(\frac{T_1}{50}\right)$:

$$
\left|h_{i^\star}\left(\frac{T_1}{50}\right)\right| \leq \left\|\boldsymbol{w}_i\left(\frac{T_1}{50}\right)\right\| \leq s_2 + 2k\eta\|\boldsymbol{v}\|\frac{T_1}{50} \leq \frac{k}{20}\eta T_1\|\boldsymbol{v}\|, \quad \forall i\in[m]. \tag{30}
$$

Combine Eqs. (29) and (30), we have:

$$
\frac{49k+49\pi-5\pi k-73.5}{100\pi}\eta T_1\|\boldsymbol{v}\| \leq h_{i^\star}(T_1) \leq \frac{49k+49\pi+5\pi k-24.5}{100\pi}\eta T_1\|\boldsymbol{v}\|, \quad \forall i\in[m]. \tag{31}
$$

Hence we finish the proof of Eq. (19). $\qquad\square$

## G  GLOBAL CONVERGENCE: PHASE 2 (BEHAVIORS ON THE TANGENTIAL GROWTH)

In Phase 2, we are interested in the dynamics of $h_i^\star$ as well as the tangential difference between the student neuron and its closest teacher neuron.

### G.1  GLOBAL CONVERGENCE: PHASE 2 (TANGENTIAL GROWTH PROCESS)

In this section, we will restate and prove Theorem 4.

**Theorem 7** (Phase 2: Tangential Growth, restate version of Theorem 4). *Assume $d = \Omega(\log(m/\delta))$ with $\delta \in (0,1)$, for any $\epsilon_1 > 0, \epsilon_2 > 0$, under Assumption 1 with $10k\zeta \leq \epsilon_1^2 = o(\zeta_i^3) = o(\epsilon_2^{\Theta(k)}/m)$, $\epsilon_2 = o(1)$, Assumptions 2, 3 such that $\sigma \leq \frac{\epsilon_1^{16k+2}\|\boldsymbol{v}\|}{10000m\sqrt{d}}$ in our random Gaussian initialization, and the stepsize satisfies $\eta = o\left(\frac{m\epsilon_1^2 s_1^2}{k^2\|\boldsymbol{v}\|^2}\right) \leq \frac{\sigma\sqrt{d}\epsilon_1^2}{100k^2\|\boldsymbol{v}\|}$, then by setting $T_1 := \frac{\epsilon_1^2}{100\eta km}$ and $T_2 = T_1 + \frac{k}{2\eta m}\ln\left(\frac{1}{48\pi\epsilon_2}\right)$, then $\forall T_1 \leq t \leq T_2$, we define $H_l(t) := \|\boldsymbol{v}\| - \sum_{i=1}^{m}\mathbb{I}_{\tau_i=l}h_{i^\star}(t)$ for $l \in [k]$, the following statements hold with probability at least $1 - \delta$:*

$$h_{i^\star}(t) \le 2h_{j^\star}(t), \forall i, j \in [m] \text{ and } \tau_i = \tau_j. \tag{32}$$

$$\left(1-\frac{\eta m}{9k}\right)^{t-T_1}\|\boldsymbol{v}\|+8\pi\epsilon_2\|\boldsymbol{v}\| \ge H_l(t) \ge \frac{2}{3}\|\boldsymbol{v}\|\left(1-\frac{3\eta m}{2k}\right)^{t-T_1}-8\pi\epsilon_2\|\boldsymbol{v}\| \ge 24\pi\epsilon_2\|\boldsymbol{v}\|, \forall l \in [k]. \tag{33}$$

$$\frac{2\|\boldsymbol{v}\|}{m_{\tau_i}} \ge h_{i^\star}(t) \ge \frac{s_1}{2}, \forall i \in [m], \tag{34}$$

*and*

$$\theta_{i^\star}(t) \le \epsilon_2, \forall i \in [m]. \tag{35}$$

*Proof.* We use induction to prove this theorem.

First, for $t = T_1$, according to Eq. (19) and Eq. (18), we have Eq. (32) and Eq. (35) hold directly.

For Eq. (33), by Eq. (21), we have:

$$\|\boldsymbol{v}\| \ge H_l(T_1) = \|\boldsymbol{v}\| - \sum_{i=1}^{m}\mathbb{I}_{\tau_i=l}h_{i^\star}(t) \ge \frac{2}{3}\|\boldsymbol{v}\|, \forall l \in [k]. \tag{36}$$

For Eq. (34), for the left part, by Eq. (21) we have:

$$h_{i^\star}(T_1) \le \|\boldsymbol{w}_i(T_1)\| \le \frac{2\|\boldsymbol{v}\|}{m} \le \frac{2\|\boldsymbol{v}\|}{m_j}, \forall i \in [m],$$

and for the right part, by Eq. (18) and Lemma 1, we have:

$$h_{i^\star}(T_1) = \|\boldsymbol{w}_i(T_1)\|\cos\theta_{i^\star}(T_1) \ge (1-8\epsilon_1^2)\|\boldsymbol{w}_i(T_1)\| \ge \frac{s_1}{2}, \forall i \in [m].$$

Next step, we assume Eqs. (32) to (35) hold for $T_1, T_1 + 1, \ldots, t$ for any $T_1 < t < T_2$, and then prove Eqs. (32) to (35) for $t + 1$.

**Proof of Eq. (32)**:

By Eq. (28), for any $i \in [m]$, we decompose the tangential difference $h_{i^\star}(t+1) - h_{i^\star}(t)$ as below:

$$h_{i^\star}(t+1) - h_{i^\star}(t)$$

$$= \frac{\eta}{2}\left( \|\boldsymbol{v}\| - \sum_{j=1}^m h_{j\tau_i}(t) \right)$$

$$- \frac{\eta}{2\pi}\left( \cos\theta_{i^\star}(t)\left( \sum_{j=1,j\neq i}^m \sin\varphi_{ij}(t)\|\boldsymbol{w}_j(t)\| - \sum_{l=1}^k \sin\theta_{il}(t)\|\boldsymbol{v}\| \right) - \sum_{j=1,j\neq i}^m \varphi_{ij}(t)h_{j\tau_i}(t) + \theta_{i^\star}(t)\|\boldsymbol{v}\| \right)$$

$$= \frac{\eta}{2}\left( H_{\tau_i}(t) - \sum_{j=1}^m \mathbb{I}_{\tau_j\neq\tau_i} h_{j\tau_i}(t) \right)$$

$$- \frac{\eta}{2\pi}\cos\theta_{i^\star}(t)\sum_{l=1,l\neq\tau_i}^k \left( \sum_{j=1}^m \mathbb{I}_{\tau_j=l}\sin\varphi_{ij}(t)\|\boldsymbol{w}_j(t)\| - \sin\theta_{il}(t)\|\boldsymbol{v}\| \right)$$

$$- \frac{\eta}{2\pi}\cos\theta_{i^\star}(t)\left( \sum_{j=1}^m \mathbb{I}_{\tau_j=\tau_i}\sin\varphi_{ij}(t)\|\boldsymbol{w}_j(t)\| - \sin\theta_{i^\star}(t)\|\boldsymbol{v}\| \right)$$

$$+ \frac{\eta}{2\pi}\left( \sum_{j=1,j\neq i}^m \varphi_{ij}(t)h_{j\tau_i}(t) + \theta_{i^\star}(t)\|\boldsymbol{v}\| \right)$$

$$= \frac{\eta}{2}\left( H_{\tau_i}(t) - \sum_{j=1}^m \mathbb{I}_{\tau_j\neq\tau_i} h_{j\tau_i}(t) \right) + \frac{\eta}{2\pi}\cos\theta_{i^\star}(t)\sum_{l=1,l\neq\tau_i}^k \left( \sin\theta_{il}(t)H_l(t) \right)$$

$$- \frac{\eta}{2\pi}\cos\theta_{i^\star}(t)\sum_{l=1,l\neq\tau_i}^k \left( \sum_{j=1}^m \mathbb{I}_{\tau_j=l}\|\boldsymbol{w}_j(t)\|\left[ \sin\varphi_{ij}(t) - \cos\theta_{jl}(t)\sin\theta_{il}(t) \right] \right)$$

$$- \frac{\eta}{2\pi}\cos\theta_{i^\star}(t)\left( \sum_{j=1}^m \mathbb{I}_{\tau_j=\tau_i}\sin\varphi_{ij}(t)\|\boldsymbol{w}_j(t)\| - \sin\theta_{i^\star}(t)\|\boldsymbol{v}\| \right)$$

$$+ \frac{\eta}{2\pi}\left( \sum_{j=1,j\neq i}^m \varphi_{ij}(t)h_{j\tau_i}(t) + \theta_{i^\star}(t)\|\boldsymbol{v}\| \right)$$

$$= \frac{\eta}{2}H_{\tau_i}(t) + \frac{\eta}{2\pi}\sum_{l=1,l\neq\tau_i}^k H_l(t)$$

$$- \frac{\eta}{2}\sum_{j=1}^m \mathbb{I}_{\tau_j\neq\tau_i} h_{j\tau_i}(t) + \frac{\eta}{2\pi}\sum_{l=1,l\neq\tau_i}^k \left( \left[ \cos\theta_{i^\star}(t)\sin\theta_{il}(t) - 1 \right]H_l(t) \right)$$

$$- \frac{\eta}{2\pi}\cos\theta_{i^\star}(t)\sum_{l=1,l\neq\tau_i}^k \left( \sum_{j=1}^m \mathbb{I}_{\tau_j=l}\|\boldsymbol{w}_j(t)\|\left[ \sin\varphi_{ij}(t) - \cos\theta_{jl}(t)\sin\theta_{il}(t) \right] \right)$$

$$- \frac{\eta}{2\pi}\cos\theta_{i^\star}(t)\left( \sum_{j=1}^m \mathbb{I}_{\tau_j=\tau_i}\sin\varphi_{ij}(t)\|\boldsymbol{w}_j(t)\| - \sin\theta_{i^\star}(t)\|\boldsymbol{v}\| \right)$$

$$+ \frac{\eta}{2\pi}\left( \sum_{j=1}^m \mathbb{I}_{\tau_i\neq\tau_j}\varphi_{ij}(t)h_{j\tau_i}(t) + \sum_{j=1}^m \mathbb{I}_{\tau_i=\tau_j}\varphi_{ij}(t)h_{j\tau_i}(t) + \theta_{i^\star}(t)\|\boldsymbol{v}\| \right)$$

$$:= \frac{\eta}{2}H_{\tau_i}(t) + \frac{\eta}{2\pi}\sum_{l=1,l\neq\tau_i}^k H_l(t) + Q_i(t),$$

$$(37)$$

where the $Q_i(t)$ is defined as:

$$Q_i(t) := -\frac{\eta}{2} \sum_{j=1}^{m} \mathbb{I}_{\tau_j \neq \tau_i} h_{j\tau_i}(t) + \frac{\eta}{2\pi} \sum_{l=1, l \neq \tau_i}^{k} \left( \left[ \cos \theta_{i^\star}(t) \sin \theta_{il}(t) - 1 \right] H_l(t) \right)$$

$$- \frac{\eta}{2\pi} \cos \theta_{i^\star}(t) \sum_{l=1, l \neq \tau_i}^{k} \left( \sum_{j=1}^{m} \mathbb{I}_{\tau_j = l} \|\boldsymbol{w}_j(t)\| \left[ \sin \varphi_{ij}(t) - \cos \theta_{jl}(t) \sin \theta_{il}(t) \right] \right)$$

$$- \frac{\eta}{2\pi} \cos \theta_{i^\star}(t) \left( \sum_{j=1}^{m} \mathbb{I}_{\tau_j = \tau_i} \sin \varphi_{ij}(t) \|\boldsymbol{w}_j(t)\| - \sin \theta_{i^\star}(t) \|\boldsymbol{v}\| \right)$$

$$+ \frac{\eta}{2\pi} \left( \sum_{j=1}^{m} \mathbb{I}_{\tau_i \neq \tau_j} \varphi_{ij}(t) h_{j\tau_i}(t) + \sum_{j=1}^{m} \mathbb{I}_{\tau_i = \tau_j} \varphi_{ij}(t) h_{j\tau_i}(t) + \theta_{i^\star}(t) \|\boldsymbol{v}\| \right).$$

To bound $Q_i$, we need to estimate $\varphi_{ij}$ and $\theta_{il}$ at first. By Eq. (35) and Assumption 2, we have that for $\tau_j = l$ and $\tau_i \neq l$:

$$\frac{\pi}{2} - 2\epsilon_2 \leq \frac{\pi}{2} - \theta_{i^\star}(t) - \theta_{j^\star}(t) \leq \varphi_{ij}(t) \leq \frac{\pi}{2} + \theta_{i^\star}(t) + \theta_{j^\star}(t) \leq \frac{\pi}{2} + 2\epsilon_2.$$

And for a similar reason, we have:

$$\frac{\pi}{2} - \epsilon_2 \leq \theta_{il}(t) \leq \frac{\pi}{2} + \epsilon_2, \quad \text{and} \quad -\epsilon_2 \leq \theta_{jl}(t) \leq \epsilon_2,$$

which implies that for a sufficient small $\epsilon_2$:

$$\sin \varphi_{ij}(t) - \cos \theta_{jl}(t) \sin \theta_{il}(t) \leq |\sin \varphi_{ij}(t) - 1| + |1 - \cos \theta_{jl}(t) \sin \theta_{il}(t)|$$

$$= \left( 1 - \cos \left( \frac{\pi}{2} - \varphi_{ij}(t) \right) \right) + \left( 1 - \cos \theta_{jl}(t) \cos \left( \frac{\pi}{2} - \theta_{il}(t) \right) \right)$$

$$\cong (1 - \cos 2\epsilon_2) + (1 - \cos^2 \epsilon_2)$$

$$\leq 2\epsilon_2^2 + \epsilon_2^2$$

$$= 3\epsilon_2^2.$$

Then using this result as well as Eqs. (34) and (35) to bound $|Q_i(t)|$, for $\forall i \in [m]$, we have:

$$
\begin{aligned}
|Q_i(t)| \leq & \frac{\eta}{2} \sum_{j=1}^{m} \mathbb{I}_{\tau_j \neq \tau_i} \frac{2\|\boldsymbol{v}\| \sin\theta_{j^\star}(t)}{m_{\tau_j} \cos\theta_{j^\star}(t)} + \frac{\eta}{2\pi}(k-1)\sin^2\theta_{i^\star}(t)\|\boldsymbol{v}\| \\
& + \frac{\eta}{2\pi}\cos\theta_{i^\star}(t) \sum_{l=1,l\neq\tau_i}^{k} \left( \sum_{j=1}^{m} \mathbb{I}_{\tau_j=l} \frac{2\|\boldsymbol{v}\|}{m_{\tau_j}\cos\theta_{j^\star}(t)} 3\epsilon_2^2 \right) \\
& + \frac{\eta}{2\pi}\cos\theta_{i^\star}(t) \sum_{j=1}^{m} \mathbb{I}_{\tau_j=\tau_i} \frac{2\|\boldsymbol{v}\|\sin\varphi_{ij}(t)}{m_{\tau_i}\cos\theta_{j^\star}(t)} + \frac{\eta}{2\pi}\frac{\sin 2\theta_{i^\star}(t)}{2}\|\boldsymbol{v}\| \\
& + \frac{\eta}{2\pi}\sum_{j=1}^{m}\mathbb{I}_{\tau_i\neq\tau_j}\left(\frac{\pi}{2}+2\epsilon_2\right)\frac{2\|\boldsymbol{v}\|\sin\theta_{j^\star}(t)}{m_{\tau_j}\cos\theta_{j^\star}(t)} + \frac{\eta}{2\pi}\sum_{j=1}^{m}\mathbb{I}_{\tau_j=\tau_i}2\epsilon_2\frac{2\|\boldsymbol{v}\|}{m_{\tau_j}} + \frac{\eta}{2\pi}\theta_{i^\star}(t)\|\boldsymbol{v}\| \\
\leq & \eta \sum_{j=1}^{m}\mathbb{I}_{\tau_j\neq\tau_i}\frac{\|\boldsymbol{v}\|\epsilon_2(1+\epsilon_2^2)}{m_{\tau_j}} + \frac{\eta}{2\pi}k\epsilon_2^2\|\boldsymbol{v}\| \\
& + \frac{\eta}{2\pi}\sum_{l=1,l\neq\tau_i}^{k}\left(\sum_{j=1}^{m}\mathbb{I}_{\tau_j=l}\frac{2\|\boldsymbol{v}\|(1+\epsilon_2^2)}{m_{\tau_j}}3\epsilon_2^2\right) \\
& + \frac{\eta}{2\pi}\sum_{j=1}^{m}\mathbb{I}_{\tau_j=\tau_i}\frac{2\|\boldsymbol{v}\|2\epsilon_2(1+\epsilon_2^2)}{m_{\tau_i}} + \frac{\eta}{2\pi}\frac{2\epsilon_2}{2}\|\boldsymbol{v}\| \\
& + \frac{\eta}{2\pi}\sum_{j=1}^{m}\mathbb{I}_{\tau_i\neq\tau_j}\left(\frac{\pi}{2}+2\epsilon_2\right)\frac{2\|\boldsymbol{v}\|\epsilon_2(1+\epsilon_2^2)}{m_{\tau_j}} + \frac{\eta}{2\pi}\sum_{j=1}^{m}\mathbb{I}_{\tau_j=\tau_i}2\epsilon_2\frac{2\|\boldsymbol{v}\|}{m_{\tau_j}} + \frac{\eta}{2\pi}\epsilon_2\|\boldsymbol{v}\| \\
\leq & 1.1\eta k\epsilon_2\|\boldsymbol{v}\| + \eta k\epsilon_2^2\|\boldsymbol{v}\| + 2\eta k\epsilon_2^2\|\boldsymbol{v}\| + 0.7\eta\epsilon_2\|\boldsymbol{v}\| + 0.2\eta\epsilon_2\|\boldsymbol{v}\| + 0.6\eta k\epsilon_2\|\boldsymbol{v}\| + 0.7\eta\epsilon_2\|\boldsymbol{v}\| + 0.2\eta\epsilon_2\|\boldsymbol{v}\| \\
\leq & 4\eta k\epsilon_2\|\boldsymbol{v}\| \\
\leq & \frac{1}{3}\left(\frac{\eta}{2}H_{\tau_i}(t) + \frac{\eta}{2\pi}\sum_{l=1,l\neq\tau_i}^{k}H_l(t)\right),
\end{aligned}
$$

$$(38)$$

where the last inequality use Eq. (33).

Then $\forall i, j \in [m]$ and $\tau_i = \tau_j$, we have:

$$
\begin{aligned}
h_{i^\star}(t+1) =& h_{i^\star}(t) + \frac{\eta}{2}H_{\tau_i}(t) + \frac{\eta}{2\pi}\sum_{l=1,l\neq\tau_i}^{k}H_l(t) + Q_i(t) \\
\leq & 2h_{j^\star}(t) + 2\left(\frac{\eta}{2}H_{\tau_j}(t) + \frac{\eta}{2\pi}\sum_{l=1,l\neq\tau_j}^{k}H_l(t) + Q_j(t)\right) \\
\leq & 2h_{j^\star}(t+1),
\end{aligned}
$$

which finishes the proof of Eq. (32).

**Proof of Eq. (33)**:

Then we derive the dynamics of $H_l(t)$, for any $l \in [k]$, we have:

$$H_l(t+1) = H_l(t) - \sum_{i=1}^m \mathbb{I}_{\tau_i = l} \left( h_{i^\star}(t+1) - h_{i^\star}(t) \right)$$

$$= H_l(t) - \sum_{i=1}^m \mathbb{I}_{\tau_i = l} \left( \frac{\eta}{2} H_{\tau_i}(t) + \frac{\eta}{2\pi} \sum_{j=1, j \neq l}^k H_j(t) + Q_i(t) \right)$$

$$= \left( 1 - \frac{m_l \eta}{2} \right) H_l(t) - \frac{m_l \eta}{2\pi} \sum_{j=1, j \neq \tau_i}^k H_j(t) + \sum_{i=1}^m \mathbb{I}_{\tau_i = l} Q_i(t) \,.$$

For ease of description, we write the recursive iteration in a matrix form

$$\boldsymbol{H}(t+1) = \left( \boldsymbol{I} - \frac{\eta}{2\pi} \boldsymbol{Diag}(m) \left( \mathbf{1}\mathbf{1}^\top + (\pi - 1)\boldsymbol{I} \right) \right) \boldsymbol{H}(t) + \boldsymbol{Q}(t) \,.$$

by defining the following quantities

$$\boldsymbol{H}(t) := [H_1(t), H_2(t), \ldots, H_k(t)]^\top \in \mathbb{R}^k,$$

$$\boldsymbol{Diag}(m) := \boldsymbol{Diag}(m_1, m_2, \ldots, m_k) \in \mathbb{R}^{k \times k},$$

$$\boldsymbol{Q}(t) := [\textstyle\sum_{i=1}^m \mathbb{I}_{\tau_i=1} Q_i(t), \sum_{i=1}^m \mathbb{I}_{\tau_i=2} Q_i(t), \ldots, \sum_{i=1}^m \mathbb{I}_{\tau_i=k} Q_i(t)]^\top \in \mathbb{R}^k \,.$$

In the next, we aim to derive the upper and lower bound of $\boldsymbol{H}(t+1)$. Denote $\boldsymbol{A} := [\frac{8\pi k \epsilon_2 \|\boldsymbol{v}\|}{\pi + k - 1}, \frac{8\pi k \epsilon_2 \|\boldsymbol{v}\|}{\pi + k - 1}, \ldots, \frac{8\pi k \epsilon_2 \|\boldsymbol{v}\|}{\pi + k - 1}]^\top \in \mathbb{R}^k$, according to Eq. (38) and Assumption 3, we have:

$$\boldsymbol{H}(t+1) - \boldsymbol{A} \preccurlyeq \left( \boldsymbol{I} - \frac{\eta}{2\pi} \boldsymbol{Diag}(m) \left( \mathbf{1}\mathbf{1}^\top + (\pi - 1)\boldsymbol{I} \right) \right) \boldsymbol{H}(t) + 4\eta k \epsilon_2 \|\boldsymbol{v}\| \boldsymbol{Diag}(m)\mathbf{1} - \boldsymbol{A} \,.$$

$$= \left( \boldsymbol{I} - \frac{\eta}{2\pi} \boldsymbol{Diag}(m) \left( \mathbf{1}\mathbf{1}^\top + (\pi - 1)\boldsymbol{I} \right) \right) \left( \boldsymbol{H}(t) - \boldsymbol{A} \right)$$

$$\preccurlyeq \left( \boldsymbol{I} - \frac{\eta}{2\pi} \frac{m}{3k} (\pi - 1)\boldsymbol{I} \right) \boldsymbol{H}(t)$$

$$\preccurlyeq \left( 1 - \frac{\eta m (\pi - 1)}{6\pi k} \right) \boldsymbol{H}(t) \,.$$

Here $\preccurlyeq$ means that all elements of the previous vector are smaller than the following vector. Then for $l \in [k]$, we have:

$$H_l(t+1) \leq \left( 1 - \frac{\eta m (\pi - 1)}{6\pi k} \right)^{t+1-T_1} H_l(T_1) + \frac{8\pi k \epsilon_2 \|\boldsymbol{v}\|}{\pi + k - 1}$$

$$\leq \left( 1 - \frac{\eta m (\pi - 1)}{6\pi k} \right)^{t+1-T_1} \|\boldsymbol{v}\| + \frac{8\pi k \epsilon_2 \|\boldsymbol{v}\|}{\pi + k - 1}$$

$$\leq \left( 1 - \frac{\eta m}{9k} \right)^{t+1-T_1} \|\boldsymbol{v}\| + 8\pi \epsilon_2 \|\boldsymbol{v}\| \,.$$

Similarly, we have

$$\boldsymbol{H}(t+1) + \boldsymbol{A} \succcurlyeq \left( \boldsymbol{I} - \frac{\eta}{2\pi} \frac{3m}{k} \left( \mathbf{1}\mathbf{1}^\top + (\pi - 1)\boldsymbol{I} \right) \right) \left( \boldsymbol{H}(t) + \boldsymbol{A} \right)$$

$$\succcurlyeq \left( \boldsymbol{I} - \frac{3\eta m}{2\pi k} \left( \mathbf{1}\mathbf{1}^\top + (\pi - 1)\boldsymbol{I} \right) \right) \boldsymbol{H}(t) \,.$$

Here $\succcurlyeq$ means that all elements of the previous vector are greater than the following vector. The eigenvalues of matrix $\boldsymbol{I} - \frac{3\eta m}{2\pi k}\left(\mathbf{1}\mathbf{1}^\top + (\pi - 1)\boldsymbol{I}\right)$ is calculated to be one $1 - \frac{3\eta m(k+\pi-1)}{2\pi k}$ and the rest $k - 1$ are $1 - \frac{3\eta m(\pi-1)}{2\pi k}$. Then according to Eq. (36),for $l \in [k]$, we have:

$$
H_l(t+1) \geq \frac{2}{3}\|\boldsymbol{v}\|\left(1 - \frac{3\eta m(k+\pi-1+(k-1)(\pi-1))}{2\pi k^2}\right)^{t+1-T_1} - 8\pi\epsilon_2\|\boldsymbol{v}\|
$$
$$
= \frac{2}{3}\|\boldsymbol{v}\|\left(1 - \frac{3\eta m}{2k}\right)^{t+1-T_1} - 8\pi\epsilon_2\|\boldsymbol{v}\| .
$$

Based on the above results, for $l \in [k]$, we have:

$$
\left(1 - \frac{\eta m}{9k}\right)^{t+1-T_1}\|\boldsymbol{v}\| + 8\pi\epsilon_2\|\boldsymbol{v}\| \geq H_l(t) \geq \frac{2}{3}\|\boldsymbol{v}\|\left(1 - \frac{3\eta m}{2k}\right)^{t+1-T_1} - 8\pi\epsilon_2\|\boldsymbol{v}\| . \quad (39)
$$

Due to $\eta m \ll 1$, we have $(1 - x) \geq \exp(-1.5x)$ with $x := \eta m$. Using this fact, for any $t \leq T_2$, the last inequality can be further lower bounded by:

$$
\frac{2}{3}\|\boldsymbol{v}\|\left(1 - \frac{3\eta m}{2k}\right)^{t-T_1} - 8\pi\epsilon_2
$$
$$
\geq \frac{2}{3}\|\boldsymbol{v}\|\exp\left(-\frac{2\eta m}{k}\frac{k}{2\eta m}\ln\left(\frac{1}{48\pi\epsilon_2}\right)\right) - 8\pi\epsilon_2\|\boldsymbol{v}\|
$$
$$
= 24\pi\epsilon_2\|\boldsymbol{v}\| .
$$

**Proof of Eq. (34):**

To prove the left part, by Eq. (33), we have: $H_l(t+1) = \|\boldsymbol{v}\| - \sum_{i=1}^m \mathbb{I}_{\tau_i=l}h_{i^\star}(t+1) \geq 0$. Then we have:

$$
\|\boldsymbol{v}\| \geq \sum_{i=1}^m \mathbb{I}_{\tau_i=l}h_{i^\star}(t+1) \geq \frac{m_{\tau_i}}{2}h_{i^\star}(t+1), \ \forall i \in [m] .
$$

For the right part, we have:

$$
h_{i^\star}(t+1) - h_{i^\star}(t) = \frac{\eta}{2}H_{\tau_i}(t) + \frac{\eta}{2\pi}\sum_{l=1,l\neq\tau_i}^k H_l(t) + Q_i(t) \geq \frac{\eta k}{2\pi}24\pi\epsilon_2\|\boldsymbol{v}\| - 4\eta k\epsilon_2\|\boldsymbol{v}\| \geq 0 .
$$

So we have $h_{i^\star}(t+1) \geq h_{i^\star}(t) \geq h_{i^\star}(T_1) \geq \frac{s_1}{2}$ .

**Proof of Eq. (35):**

First, we prove that for $\forall i,j \in [m]$, $T_1 \leq t \leq T_2$, we have $\frac{\|\boldsymbol{w}_i(t)\|}{\|\boldsymbol{w}_j(t)\|} = \Theta(1)$.

When $t = T_1$, according to Eq. (31) we have:

$$
\frac{1}{2} \leq \frac{h_{i^\star}(T_1)}{h_{j^\star}(T_1)} \leq 2, \qquad \forall i,j \in [m],
$$

which implies:

$$
\frac{\|\boldsymbol{w}_i(T_1)\|}{\|\boldsymbol{w}_j(T_1)\|} = \frac{h_{i^\star}(t)\cos\theta_{i^\star}(T_1)}{h_{j^\star}(t)\cos\theta_{j^\star}(T_1)} = \Theta(1), \quad \forall i,j \in [m]. \quad (40)
$$

Then by defining $t_s = \frac{9k\ln(2)}{\eta m} + T_1$, when $T_1 \leq t \leq t_s$, according to Eq. (33), for any $l \in [k]$, we have:

$$H_l(t) \geq \frac{2}{3} \|\boldsymbol{v}\| \left(1 - \frac{3\eta m}{2k}\right)^{t-T_1} - 8\pi\epsilon_2 \|\boldsymbol{v}\|$$

$$\geq \frac{2}{3} \|\boldsymbol{v}\| \exp\left(-\frac{2\eta m}{k} \frac{9k\ln(2)}{\eta m}\right) - 8\pi\epsilon_2 \|\boldsymbol{v}\|$$

$$\geq \frac{2}{3}\left(\frac{1}{2}\right)^{18} \|\boldsymbol{v}\| - 8\pi\epsilon_2 \|\boldsymbol{v}\| \,.$$

So for $\forall l_1, l_2 \in [k]$, we have $\frac{H_{l_1}(t)}{H_{l_2}(t)} = \Theta(1)$.

Then for $\forall i, j \in [m]$, according to Eq. (37), for $T_1 \leq t_0 < t$, we have $\frac{h_{i^\star}(t_0+1)-h_{i^\star}(t_0)}{h_{j^\star}(t_0+1)-h_{j^\star}(t_0)} = \Theta(1)$.

Then consider Eq. (31), we have $\frac{h_{i^\star}(t)}{h_{j^\star}(t)} = \Theta(1)$.

That means for $\forall i, j \in [m]$, when $T_1 \leq t \leq t_s$, we have:

$$\frac{\|\boldsymbol{w}_i(t)\|}{\|\boldsymbol{w}_j(t)\|} = \frac{h_{i^\star}(t)\cos\theta_{i^\star}(t)}{h_{j^\star}(t)\cos\theta_{j^\star}(t)} = \Theta(1) \,. \tag{41}$$

When $t_s \leq t \leq T_2$, according to Eq. (33), for $\forall l \in [k]$, we have:

$$H_l(t) \leq \left(1 - \frac{\eta m}{9k}\right)^{t-T_1} \|\boldsymbol{v}\| + 8\pi\epsilon_2 \|\boldsymbol{v}\|$$

$$\leq \exp\left(-\frac{\eta m}{9k} \frac{9k\ln(2)}{\eta m}\right) \|\boldsymbol{v}\| + 8\pi\epsilon_2 \|\boldsymbol{v}\| \qquad [\text{using } (1-x) \leq \exp(-x), \forall x \geq 0]$$

$$= \frac{1}{2} \|\boldsymbol{v}\| + 8\pi\epsilon_2 \|\boldsymbol{v}\| \,.$$

Then we have:

$$\sum_{i=1}^{m} \mathbb{I}_{\tau_i=l} h_{i^\star}(t) = \|\boldsymbol{v}\| - H_l(t) \geq \frac{1}{2} \|\boldsymbol{v}\| - 8\pi\epsilon_2 \|\boldsymbol{v}\| \geq \frac{1}{3} \|\boldsymbol{v}\| \,.$$

Then for $\forall i, j \in [m]$, we have:

$$h_{i^\star}(t) \geq \frac{\sum_{l=i}^{m} \mathbb{I}_{\tau_i=\tau_l} h_{l^\star}(t)}{2m_{\tau_i}}$$

$$\geq \frac{\|\boldsymbol{v}\|}{6m_{\tau_i}}$$

$$\geq \frac{\sum_{l=i}^{m} \mathbb{I}_{\tau_j=\tau_l} h_{l^\star}(t)}{6m_{\tau_i}}$$

$$\geq \frac{m_{\tau_j} h_{j^\star}(t)}{12m_{\tau_i}}$$

$$\geq \frac{h_{j^\star}(t)}{108} \,.$$

That means for $\forall i, j \in [m]$, when $t_s \leq t \leq T_2$, we have:

$$\frac{\|\boldsymbol{w}_i(t)\|}{\|\boldsymbol{w}_j(t)\|} = \frac{h_{i^\star}(t)\cos\theta_{i^\star}(t)}{h_{j^\star}(t)\cos\theta_{j^\star}(t)} = \Theta(1) \,. \tag{42}$$

So combine Eqs. (40) to (42), for $\forall i, j \in [m]$, when $T_1 \leq t \leq T_2$, we have:

$$\frac{\|\boldsymbol{w}_i(t)\|}{\|\boldsymbol{w}_j(t)\|} = \frac{h_{i^\star}(t)\cos\theta_{i^\star}(t)}{h_{j^\star}(t)\cos\theta_{j^\star}(t)} = \Theta(1) \,. \tag{43}$$

Then, we analyze the change in angle, recall the dynamics of $\cos\theta_{i^\star}$ in Eq. (24) is given by:

$$\cos\theta_{i^\star}(t+1) - \cos\theta_{i^\star}(t) =: I_2 + I_3 \,.$$

For $I_2$, we have:

$$
\begin{aligned}
I_2 &= \frac{\eta}{\|\boldsymbol{w}_i(t+1)\|} \left\langle \langle \bar{\boldsymbol{w}}_i(t), \bar{\boldsymbol{v}}_{\tau_i}\rangle \bar{\boldsymbol{w}}_i(t) - \bar{\boldsymbol{v}}_{\tau_i}, \nabla_i(t) \right\rangle \\
&= \frac{\eta}{2\pi \|\boldsymbol{w}_i(t+1)\|} \left( \sum_{j=1}^{m} \big( \|\boldsymbol{w}_j(t)\| (\pi - \varphi_{ij}(t))(\cos\theta_{i^\star}(t)\cos\varphi_{ij}(t) - \cos\theta_{j\tau_i}(t)) \big) \right. \\
&\qquad \left. - \sum_{l=1, l\neq\tau_i}^{k} \big( \|\boldsymbol{v}\| (\pi - \theta_{il}(t))\cos\theta_{i^\star}(t)\cos\theta_{il}(t)) + \|\boldsymbol{v}\| \sin^2\theta_{i^\star}(t)(\pi - \theta_{i^\star}(t)) \big) \right) \,.
\end{aligned}
$$

To bound $I_2$, we need handle $\cos\theta_{i^\star}(t)\cos\varphi_{ij}(t) - \cos\theta_{j\tau_i}(t)$ at first. For $\tau_i \neq \tau_j$, without loss of generality, we assume that: $\bar{\boldsymbol{v}}_{\tau_i} = [1, 0, 0, \ldots, 0]^\top \in \mathbb{R}^d$ and $\bar{\boldsymbol{v}}_{\tau_j} = [0, 1, 0, 0, \ldots, 0]^\top \in \mathbb{R}^d$. Let $\bar{\boldsymbol{w}}_i = [w_{i1}, w_{i2}, \ldots, w_{id}]^\top \in \mathbb{R}^d$ and $\bar{\boldsymbol{w}}_j = [w_{j1}, w_{j2}, \ldots, w_{jd}]^\top \in \mathbb{R}^d$, then we have:

$$
\begin{aligned}
&\cos\theta_{i^\star}(t)\cos\varphi_{ij}(t) - \cos\theta_{j\tau_i}(t) \\
&= \langle \bar{\boldsymbol{w}}_i, \bar{\boldsymbol{v}}_{\tau_i}\rangle \langle \bar{\boldsymbol{w}}_i, \bar{\boldsymbol{w}}_j\rangle - \langle \bar{\boldsymbol{w}}_j, \bar{\boldsymbol{v}}_{\tau_i}\rangle \\
&= w_{i1}(t) \sum_{l=1}^{d} w_{il}(t)w_{jl}(t) - w_{j1}(t) \\
&= w_{i1}(t)\left( w_{i1}(t)w_{j1}(t) + w_{i2}(t)w_{j2}(t) + \sum_{l=3}^{d} w_{il}(t)w_{jl}(t) \right) - w_{j1}(t) \\
&= w_{i1}(t)\left( w_{i2}(t)w_{j2}(t) + \sum_{l=3}^{d} w_{il}(t)w_{jl}(t) \right) - \sin^2\theta_{i^\star}(t)w_{j1}(t) \\
&\geq -\left| w_{i2}(t)w_{j2}(t) + \sum_{l=3}^{d} w_{il}(t)w_{jl}(t) \right| - \sin^2\theta_{i^\star}(t)|w_{j1}(t)| \\
&\geq -|w_{i2}(t)w_{j2}(t)| - \left| \sum_{l=3}^{d} w_{il}(t)w_{jl}(t) \right| - \sin^2\theta_{i^\star}(t)|w_{j1}(t)| \\
&\geq -|w_{i2}(t)w_{j2}(t)| - \left| \left( \sum_{l=3}^{d} w_{il}(t)^2 \right)^{\frac{1}{2}} \left( \sum_{l=3}^{d} w_{jl}(t)^2 \right)^{\frac{1}{2}} \right| - \sin^2\theta_{i^\star}(t)|w_{j1}(t)| \quad \text{[Cauchy–Schwarz inequality]} \\
&\geq -|w_{i2}(t)w_{j2}(t)| - \sin\theta_{i^\star}(t)\sin\theta_{j^\star}(t) - \sin^2\theta_{i^\star}(t)|w_{j1}(t)| \\
&\geq -\sin\theta_{i^\star}(t)\sin\theta_{j^\star}(t) - 2\zeta \,.
\end{aligned}
$$

For $\tau_i = \tau_j$, without loss of generality, we assume that: $\bar{\boldsymbol{v}}_{\tau_i} = \bar{\boldsymbol{v}}_{\tau_j} = [1, 0, 0, \ldots, 0]^\top \in \mathbb{R}^d$. Then, we let $\bar{\boldsymbol{w}}_i = [w_{i1}, w_{i2}, \ldots, w_{id}]^\top \in \mathbb{R}^d$ and $\bar{\boldsymbol{w}}_j = [w_{j1}, w_{j2}, \ldots, w_{jd}]^\top \in \mathbb{R}^d$. Then we have:

$$\cos \theta_{i^\star}(t) \cos \varphi_{ij}(t) - \cos \theta_{j\tau_i}(t)$$

$$= \langle \bar{\boldsymbol{w}}_i, \bar{\boldsymbol{v}}_{\tau_i} \rangle \langle \bar{\boldsymbol{w}}_i, \bar{\boldsymbol{w}}_j \rangle - \langle \bar{\boldsymbol{w}}_j, \bar{\boldsymbol{v}}_{\tau_i} \rangle$$

$$= w_{i1}(t) \sum_{l=1}^{d} w_{il}(t) w_{jl}(t) - w_{j1}(t)$$

$$= w_{i1}(t) \left( w_{i1}(t) w_{j1}(t) + \sum_{l=2}^{d} w_{il}(t) w_{jl}(t) \right) - w_{j1}(t)$$

$$= w_{i1}(t) \left( \sum_{l=2}^{d} w_{il}(t) w_{jl}(t) \right) - \sin^2 \theta_{i^\star}(t) w_{j1}(t)$$

$$= \cos \theta_{i^\star}(t) \left( \sum_{l=2}^{d} w_{il}(t) w_{jl}(t) \right) - \sin^2 \theta_{i^\star}(t) \cos \theta_{j\tau_i}(t)$$

$$\geq - \left( \sum_{l=2}^{d} w_{il}(t)^2 \right)^{\frac{1}{2}} \left( \sum_{l=2}^{d} w_{jl}(t)^2 \right)^{\frac{1}{2}} - \sin^2 \theta_{i^\star}(t) \quad \text{[Cauchy–Schwarz inequality]}$$

$$= - \sin \theta_{i^\star}(t) \sin \theta_{j^\star}(t) - \sin^2 \theta_{i^\star}(t).$$

Then we have:

$$I_2 = \frac{\eta}{2\pi \|\boldsymbol{w}_i(t+1)\|} \left( \sum_{j=1}^{m} \left( \|\boldsymbol{w}_j(t)\| (\pi - \varphi_{ij}(t))(\cos \theta_{i^\star}(t) \cos \varphi_{ij}(t) - \cos \theta_{j\tau_i}(t)) \right) \right.$$

$$\left. - \sum_{l=1, l \neq \tau_i}^{k} \left( \|\boldsymbol{v}\| (\pi - \theta_{il}(t)) \cos \theta_{i^\star}(t) \cos \theta_{il}(t)) + \|\boldsymbol{v}\| \sin^2 \theta_{i^\star}(t)(\pi - \theta_{i^\star}(t)) \right) \right)$$

$$\geq - \frac{\eta}{\|\boldsymbol{w}_i(t+1)\|} \left( \sum_{j=1}^{m} \left[ \|\boldsymbol{w}_j(t)\| (2\zeta + \sin \theta_{i^\star}(t) \sin \theta_{j^\star}(t)) \right] + \sum_{j=1}^{m} \mathbb{I}_{\tau_j = \tau_i} \left( \|\boldsymbol{w}_j(t)\| \sin^2 \theta_{i^\star}(t) \right) \right.$$

$$\left. + (k-1) \|\boldsymbol{v}\| \pi\zeta - (\pi - \theta_{i^\star}(t)) \|\boldsymbol{v}\| \sin^2 \theta_{i^\star}(t) \right)$$

$$\geq -C^\star \eta \sin \theta_{i^\star}(t) \sum_{j=1}^{m} \sin \theta_{j^\star}(t) - \frac{6k\eta\zeta \|\boldsymbol{v}\|}{\|\boldsymbol{w}_i(t+1)\|} \quad \text{[Eq. (43)]}$$

$$\geq -C^\star \eta \sin \theta_{i^\star}(t) \sum_{j=1}^{m} \sin \theta_{j^\star}(t) - \frac{12k\eta\zeta \|\boldsymbol{v}\|}{s_1} \quad \text{[Eq. (34)]}.$$

$$(44)$$

In the next, we aim to bound $I_3$, which requires the estimation of the gradient. Similar to Eq. (22), we have:

$$\|\nabla_i(t)\|$$

$$\leq \left\|\frac{1}{2}\sum_{j=1}^{m}\boldsymbol{w}_j(t)\right\| + \left\|\frac{1}{2}\sum_{l=1}^{k}\boldsymbol{v}_l\right\|$$

$$+ \left\|\frac{1}{2\pi}\left[\frac{\boldsymbol{w}_i(t)}{\|\boldsymbol{w}_i(t)\|}\left(\sum_{j=1,j\neq i}^{m}\sin\varphi_{ij}(t)\|\boldsymbol{w}_j(t)\| - \sum_{l=1}^{k}\sin\theta_{il}(t)\|\boldsymbol{v}\|\right) - \sum_{j=1,j\neq i}^{m}\varphi_{ij}(t)\boldsymbol{w}_j(t) + \sum_{l=1}^{k}\theta_{il}(t)\boldsymbol{v}_l(t)\right]\right\|$$

$$\leq \frac{m}{2}\times\frac{9k\|\boldsymbol{v}\|}{m} + \frac{k}{2}\|\boldsymbol{v}\| + \frac{1}{2\pi}\left(m\times\frac{9k\|\boldsymbol{v}\|}{m} + k\|\boldsymbol{v}\| + m\pi\times\frac{9k\|\boldsymbol{v}\|}{m} + k\pi\|\boldsymbol{v}\|\right)$$

$$< 15k\|\boldsymbol{v}\| .$$

$$(45)$$

Combining with this result, we can derive the lower bound for $I_3$:

$$I_3 = \frac{\eta\langle\bar{\boldsymbol{w}}_i(t),\bar{\boldsymbol{v}}_{\tau_i}\rangle}{\|\boldsymbol{w}_i(t+1)\|}\left(\frac{\langle\bar{\boldsymbol{w}}_i(t),\nabla_i(t)\rangle(\|\boldsymbol{w}_i(t)\| - \|\boldsymbol{w}_i(t+1)\|) - \eta\|\nabla_i(t)\|^2}{\|\boldsymbol{w}_i(t+1)\| + \|\boldsymbol{w}_i(t)\|}\right)$$

$$\geq -\frac{\eta}{\|\boldsymbol{w}_i(t+1)\|}\left(\frac{\|\nabla_i(t)\|\|\eta\nabla_i(t)\| + \eta\|\nabla_i(t)\|^2}{s_1}\right)$$

$$= -\frac{4\eta^2\|\nabla_i(t)\|^2}{s_1^2}$$

$$\geq -\frac{900k^2\eta^2\|\boldsymbol{v}\|^2}{s_1^2} .$$

$$(46)$$

Subsequently, we need to estimate the difference $\sin^2\left(\frac{\theta_{i^\star}(t+1)}{2}\right) - \sin^2\left(\frac{\theta_{i^\star}(t)}{2}\right)$ for our final estimation for $\sin\theta_{i^\star}$. Hence, similar to Eq. (27), combining Eq. (44), for $\forall i \in [m]$, we have:

$$\sin^2\left(\frac{\theta_{i^\star}(t+1)}{2}\right) - \sin^2\left(\frac{\theta_{i^\star}(t)}{2}\right)$$

$$= -\frac{1}{2}\left(\cos\theta_{i^\star}(t+1) - \cos\theta_{i^\star}(t)\right)$$

$$\leq -\frac{1}{2}\left(-C^\star\eta\sin\theta_{i^\star}(t)\sum_{j=1}^{m}\sin\theta_{j^\star}(t) - \frac{12k\eta\zeta\|\boldsymbol{v}\|}{s_1} - \frac{900k^2\eta^2\|\boldsymbol{v}\|^2}{s_1^2}\right) \qquad \text{[using Eq. (44) and Eq. (46)]}$$

$$\leq 2C^\star\eta\sin\left(\frac{\theta_{i^\star}(t)}{2}\right)\sum_{j=1}^{m}\sin\left(\frac{\theta_{j^\star}(t)}{2}\right) + \frac{6k\eta\zeta\|\boldsymbol{v}\|}{s_1} + \frac{450k^2\eta^2\|\boldsymbol{v}\|^2}{s_1^2} .$$

Summing over all student neurons yields:

$$\sum_{i=1}^{m}\sin^2\left(\frac{\theta_{i^\star}(t+1)}{2}\right) - \sum_{i=1}^{m}\sin^2\left(\frac{\theta_{i^\star}(t)}{2}\right)$$

$$\leq \sum_{i=1}^{m}\left[2C^\star\eta\sin\left(\frac{\theta_{i^\star}(t)}{2}\right)\sum_{j=1}^{m}\sin\left(\frac{\theta_{j^\star}(t)}{2}\right) + \frac{6k\zeta\eta\|\boldsymbol{v}\|}{s_1} + \frac{450k^2\eta^2\|\boldsymbol{v}\|^2}{s_1^2}\right]$$

$$= 2C^\star\eta\sum_{i=1}^{m}\sin\left(\frac{\theta_{i^\star}(t)}{2}\right)\sum_{j=1}^{m}\sin\left(\frac{\theta_{j^\star}(t)}{2}\right) + \frac{6km\zeta\eta\|\boldsymbol{v}\|}{s_1} + \frac{450k^2m\eta^2\|\boldsymbol{v}\|^2}{s_1^2}$$

$$\leq 2C^\star\eta m\sum_{i=1}^{m}\sin^2\left(\frac{\theta_{i^\star}(t)}{2}\right) + \frac{6km\zeta\eta\|\boldsymbol{v}\|}{s_1} + \frac{450k^2m\eta^2\|\boldsymbol{v}\|^2}{s_1^2} . \qquad \text{[using AM-GM ineuqality]}$$

$$(47)$$

Then we have:

$$
\begin{aligned}
&\sum_{i=1}^{m} \sin^2\left(\frac{\theta_{i^\star}(t+1)}{2}\right) \\
&\leq \sum_{i=1}^{m} \sin^2\left(\frac{\theta_{i^\star}(t+1)}{2}\right) + \frac{3k\zeta\|\boldsymbol{v}\|}{C^\star s_1} + \frac{225k^2\eta\|\boldsymbol{v}\|^2}{C^\star s_1^2} \\
&\leq (1+2C^\star\eta m)\left(\sum_{i=1}^{m}\sin^2\left(\frac{\theta_{i^\star}(t)}{2}\right) + \frac{3k\zeta\|\boldsymbol{v}\|}{C^\star s_1} + \frac{225k^2\eta\|\boldsymbol{v}\|^2}{C^\star s_1^2}\right) \quad \text{[Eq. (47)]} \\
&\leq (1+2C^\star\eta m)^{t+1-T_1}\left(\sum_{i=1}^{m}\sin^2\left(\frac{\theta_{i^\star}(T_1)}{2}\right) + \frac{3k\zeta\|\boldsymbol{v}\|}{C^\star s_1} + \frac{225k^2\eta\|\boldsymbol{v}\|^2}{C^\star s_1^2}\right) \\
&\leq (1+2C^\star\eta m)^{t+1-T_1}4m\epsilon_1^2 \quad \text{[by Assumption 1, choosing } \zeta = o\left(\frac{m\epsilon_1^2 s_1}{k\|\boldsymbol{v}\|}\right)] \\
&\leq \exp\left(2C^\star\eta m\frac{k}{2\eta m}\ln\left(\frac{1}{48\pi\epsilon_2}\right)\right)4m\epsilon_1^2 \quad \text{[using } 1+x\leq\exp(x)] \\
&\leq \frac{4m\epsilon_1^2}{(48\pi\epsilon_2)^{C^\star k}} \\
&\leq \frac{\epsilon_2^2}{16},
\end{aligned}
$$

where the last inequality needs $\epsilon_1^2 \leq \frac{(48\pi\epsilon_2)^{C^\star k}\epsilon_2^2}{64m}$.

Finally we finish the proof for Eq. (35), i.e.,

$$
\theta_{i^\star}(t+1) \leq \epsilon_2, \qquad \forall i \in [m].
$$

which finishes the proof.

$\square$

### G.2 GLOBAL CONVERGENCE: PHASE 2 (FINAL STATE)

Here we prove the bounds on the student neurons and the loss function at the end of phase 2.

**Lemma 3** (Final state of Phase 2, restate version of Corollary 2). *Under the same conditions as Theorem 7, at time $T_2$, we have the following statements hold with probability at least $1-\delta$:*

$$
\frac{\|\boldsymbol{v}\|}{3m_{\tau_i}} \leq \|\boldsymbol{w}_i(T_2)\| \leq \frac{3\|\boldsymbol{v}\|}{m_{\tau_i}}, \ \forall i \in [m],
$$

*and*

$$
L(\boldsymbol{W}(T_2)) \leq \frac{1}{2}k^2\epsilon_2^{0.05}\|\boldsymbol{v}\|^2.
$$

*Proof.* Firstly we derive the bound for the $\|\boldsymbol{w}_i(T_2)\|$. By Eq. (33) in Theorem 7, for any $l \in [k]$, we have:

$$H_l(T_2) \leq \left(1 - \frac{\eta m}{9k}\right)^{T_2 - T_1} \|\boldsymbol{v}\| + 8\pi\epsilon_2 \|\boldsymbol{v}\|$$

$$\leq \exp\left(-\frac{\eta m}{9k}(T_2 - T_1)\right) \|\boldsymbol{v}\| + 8\pi\epsilon_2 \|\boldsymbol{v}\|$$

$$= (48\pi\epsilon_2)^{\frac{1}{18}} \|\boldsymbol{v}\| + 8\pi\epsilon_2 \|\boldsymbol{v}\|$$

$$\leq (49\pi\epsilon_2)^{\frac{1}{18}} \|\boldsymbol{v}\|$$

$$\leq \frac{1}{3} \|\boldsymbol{v}\| \,,$$

which implies

$$\frac{2}{3} \|\boldsymbol{v}\| \leq \|\boldsymbol{v}\| - H_{\tau_i}(T_2) = \sum_{j=1}^{m} \mathbb{I}_{\tau_j = \tau_i} h_{j^\star}(T_2) \leq 2m_{\tau_i} h_{i^\star}(T_2)\,, \qquad \forall i \in [m]\,.$$

So we have the lower bound $\|\boldsymbol{w}_i(T_2)\| \geq h_{i^\star}(T_2) \geq \frac{\|\boldsymbol{v}\|}{3m_{\tau_i}}$. For the upper bound, for any $i \in [m]$, we have $H_{\tau_i}(T_2) \geq 0$:

$$\|\boldsymbol{v}\| \geq \sum_{j=1}^{m} \mathbb{I}_{\tau_j = \tau_i} h_{j^\star}(T_2) \geq \frac{1}{2} m_{\tau_i} h_{i^\star}(T_2) = \frac{1}{2} m_{\tau_i} \|\boldsymbol{w}_i(T_2)\| \cos\theta_{i^\star}(T_2) \geq \frac{1}{3} m_{\tau_i} \|\boldsymbol{w}_i(T_2)\| \,,$$

which implies $\|\boldsymbol{w}_i(T_2)\| \leq \frac{3\|\boldsymbol{v}\|}{m_{\tau_i}}$ and the following estimation which is used for estimating the loss. To be specific, for any $l \in [k]$, we have:

$$\sum_{i=1}^{m} \mathbb{I}_{\tau_i = l} \|\boldsymbol{w}_i(T_2)\| = \sum_{i=1}^{m} \mathbb{I}_{\tau_i = l} \frac{h_{i^\star}(T_2)}{\cos\theta_{i^\star}(T_2)} \leq (1 + \epsilon_2^2) \sum_{i=1}^{m} \mathbb{I}_{\tau_i = l} h_{i^\star}(T_2) \leq (1 + \epsilon_2^2) \|\boldsymbol{v}\| \,,$$

and

$$\sum_{i=1}^{m} \mathbb{I}_{\tau_i = l} \|\boldsymbol{w}_i(T_2)\| \geq \sum_{i=1}^{m} \mathbb{I}_{\tau_i = l} h_{i^\star}(T_2) \geq \left(1 - (49\pi\epsilon_2)^{\frac{1}{18}}\right) \|\boldsymbol{v}\| \geq \left(1 - \epsilon_2^{0.05}\right) \|\boldsymbol{v}\| \,.$$

Combine the lower and upper bound, we have:

$$\left(1 - \epsilon_2^{0.05}\right) \|\boldsymbol{v}\| \leq \sum_{i=1}^{m} \mathbb{I}_{\tau_i = l} \|\boldsymbol{w}_i(T_2)\| \leq (1 + \epsilon_2^2) \|\boldsymbol{v}\| \,. \tag{48}$$

Before we bound the loss, we need to analyze $g(\boldsymbol{a}, \boldsymbol{b})$ defined in Eq. (12). If $\angle(\boldsymbol{a}, \boldsymbol{b}) \leq 2\epsilon_2$ we have:

$$\frac{\pi - 2\epsilon_2}{2\pi} \|\boldsymbol{a}\| \|\boldsymbol{b}\| \leq g(\boldsymbol{a}, \boldsymbol{b}) = \frac{\|\boldsymbol{a}\| \|\boldsymbol{b}\|}{2\pi} \left(\sin\angle(\boldsymbol{a}, \boldsymbol{b}) + (\pi - \angle(\boldsymbol{a}, \boldsymbol{b})) \cos\angle(\boldsymbol{a}, \boldsymbol{b})\right) \leq \frac{1}{2} \|\boldsymbol{a}\| \|\boldsymbol{b}\| \,, \tag{49}$$

Besides, if $-2\epsilon_2 \leq \frac{\pi}{2} - \angle(\boldsymbol{a}, \boldsymbol{b}) \leq 2\epsilon_2$, we have:

$$\frac{1 - 4\epsilon_2}{2\pi} \|\boldsymbol{a}\| \|\boldsymbol{b}\| \leq g(\boldsymbol{a}, \boldsymbol{b}) \leq \frac{1 + 4\epsilon_2}{2\pi} \|\boldsymbol{a}\| \|\boldsymbol{b}\| \,. \tag{50}$$

According to Eq. (35) in Theorem 7, then when $\tau_i = \tau_j$, we have $\varphi_{ij} \leq 2\epsilon_2$ and when $\tau_i \neq \tau_j$, we have $-2\epsilon_2 \leq \frac{\pi}{2} - \varphi_{ij} \leq 2\epsilon_2$.

Then, according to Eqs. (48) to (50), we have:

$$L(\boldsymbol{W}(T_2)) = \frac{1}{2}\sum_{i=1}^{m}\sum_{j=1}^{m}g(\boldsymbol{w}_i(T_2), \boldsymbol{w}_j(T_2)) + \frac{1}{2}\sum_{i=1}^{k}\sum_{j=1}^{k}g(\boldsymbol{v}_i, \boldsymbol{v}_j) - \sum_{i=1}^{m}\sum_{j=1}^{k}g(\boldsymbol{w}_i(T_2), \boldsymbol{v}_j)$$

$$\leq \frac{1}{2}\sum_{l=1}^{k}\sum_{i=1}^{m}\mathbb{I}_{\tau_i=l}\sum_{j=1}^{m}\mathbb{I}_{\tau_j=l}\frac{1}{2}\|\boldsymbol{w}_i(T_2)\|\|\boldsymbol{w}_j(T_2)\| + \frac{1}{2}\sum_{i=1}^{m}\sum_{j=1}^{m}\mathbb{I}_{\tau_i\neq\tau_l}\frac{1+4\epsilon_2}{2\pi}\|\boldsymbol{w}_i(T_2)\|\|\boldsymbol{w}_j(T_2)\|$$

$$+ \frac{k\|\boldsymbol{v}\|^2}{2}\frac{}{2} + \frac{k(k-1)}{2}\frac{\|\boldsymbol{v}\|^2}{2\pi}$$

$$- \sum_{l=1}^{k}\sum_{i=1}^{m}\mathbb{I}_{\tau_i=l}\frac{\pi-2\epsilon_2}{2\pi}\|\boldsymbol{w}_i(T_2)\|\|\boldsymbol{v}_l\| - \sum_{l=1}^{k}\mathbb{I}_{\tau_i\neq l}\frac{1-4\epsilon_2}{2\pi}\|\boldsymbol{w}_i(T_2)\|\|\boldsymbol{v}_l\|$$

$$= \frac{1}{2}\sum_{l=1}^{k}\sum_{i=1}^{m}\mathbb{I}_{\tau_i=l}\sum_{j=1}^{m}\mathbb{I}_{\tau_j=l}\frac{1}{2}\|\boldsymbol{w}_i(T_2)\|\|\boldsymbol{w}_j(T_2)\| + \frac{1}{2}\sum_{i=1}^{m}\sum_{j=1}^{m}\mathbb{I}_{\tau_i\neq\tau_j}\frac{1}{2\pi}\|\boldsymbol{w}_i(T_2)\|\|\boldsymbol{w}_j(T_2)\|$$

$$+ \frac{k\|\boldsymbol{v}\|^2}{4} + \frac{k(k-1)\|\boldsymbol{v}\|^2}{4\pi} - \sum_{l=1}^{k}\sum_{i=1}^{m}\mathbb{I}_{\tau_i=l}\frac{1}{2}\|\boldsymbol{w}_i(T_2)\|\|\boldsymbol{v}_l\| - \sum_{l=1}^{k}\sum_{i=1}^{m}\mathbb{I}_{\tau_i\neq l}\frac{1}{2\pi}\|\boldsymbol{w}_i(T_2)\|\|\boldsymbol{v}_l\|$$

$$+ \sum_{i=1}^{m}\sum_{j=1}^{m}\mathbb{I}_{\tau_i\neq\tau_j}\frac{\epsilon_2}{\pi}\|\boldsymbol{w}_i(T_2)\|\|\boldsymbol{w}_j(T_2)\| + \sum_{l=1}^{k}\sum_{i=1}^{m}\mathbb{I}_{\tau_i=l}\frac{\epsilon_2}{\pi}\|\boldsymbol{w}_i(T_2)\|\|\boldsymbol{v}_l\|$$

$$+ \sum_{l=1}^{k}\sum_{i=1}^{m}\mathbb{I}_{\tau_i\neq l}\frac{2\epsilon_2}{\pi}\|\boldsymbol{w}_i(T_2)\|\|\boldsymbol{v}_l\| \quad [\text{ Eqs. (49) and (50)}]$$

$$\leq \frac{k(1+\epsilon_2^2)^2\|\boldsymbol{v}\|^2}{4} + \frac{k(k-1)(1+\epsilon_2^2)^2\|\boldsymbol{v}\|^2}{4\pi} + \frac{k\|\boldsymbol{v}\|^2}{4} + \frac{k(k-1)\|\boldsymbol{v}\|^2}{4\pi}$$

$$- \frac{k(1-\epsilon_2^{0.05})\|\boldsymbol{v}\|^2}{2} - \frac{k(k-1)(1-\epsilon_2^{0.05})\|\boldsymbol{v}\|^2}{2\pi}$$

$$+ \frac{k(k-1)(1+\epsilon_2^2)^2\epsilon_2\|\boldsymbol{v}\|^2}{\pi} + \frac{k(1+\epsilon_2^2)\epsilon_2\|\boldsymbol{v}\|^2}{\pi} + \frac{2k(k-1)(1+\epsilon_2^2)\epsilon_2\|\boldsymbol{v}\|^2}{\pi} \quad [\text{ Eq. (48)}]$$

$$\leq \frac{1}{2}k^2\epsilon_2^{0.05}\|\boldsymbol{v}\|^2 ,$$

$$(51)$$

which concludes the proof. $\qquad\square$

# H  GLOBAL CONVERGENCE: PHASE 3 (LOCAL CONVERGENCE)

In phase 3, we focus on the local convergence of the network when the loss function has an upper bound. First, we introduce some structural lemmas related to the loss function of neural network.

## H.1  STRUCTURAL LEMMAS

**Lemma 4.** *We define that* $\boldsymbol{w}_i^\star := \frac{h_{i\star}}{\sum_{j=1}^{m}\mathbb{I}_{\tau_j=\tau_i}h_{j\star}}\boldsymbol{v}_{\tau_i}$, *and* $\theta_{\max} := \max_{i\in[m]}\theta_{i\star}$, *then we have:*

$$\sum_{i=1}^{m}\left\langle\frac{\partial}{\partial\boldsymbol{w}_i}L(\boldsymbol{W}), \boldsymbol{w}_i - \boldsymbol{w}_i^\star\right\rangle \geq 2L(\boldsymbol{W}) - \mathcal{O}(k\theta_{\max}^2\sum_{l=1}^{k}\|\boldsymbol{r}_l\|\|\boldsymbol{v}\|) .$$

*Proof.* First, we decomposes the residual function $R(\boldsymbol{x})$ into two terms:

$$R(\boldsymbol{x}) := \sum_{i=1}^{m} \phi(\boldsymbol{w}_i^\top \boldsymbol{x}) - \sum_{l=1}^{k} \phi(\boldsymbol{v}_l^\top \boldsymbol{x})$$

$$= \sum_{i=1}^{m} (\boldsymbol{w}_i^\top \boldsymbol{x}) \phi'(\boldsymbol{w}_i^\top \boldsymbol{x}) - \sum_{l=1}^{k} (\boldsymbol{v}_l^\top \boldsymbol{x}) \phi'(\boldsymbol{v}_l^\top \boldsymbol{x}) \quad \text{[using ReLU property: } \phi(x) = x\phi'(x)\text{]}$$

$$= \sum_{i=1}^{m} (\boldsymbol{w}_i^\top \boldsymbol{x}) \phi'(\boldsymbol{w}_i^\top \boldsymbol{x}) - \sum_{l=1}^{k} ((\sum_{i=1}^{m} \mathbb{I}_{\tau_i=l} \boldsymbol{w}_i - \boldsymbol{r}_l)^\top \boldsymbol{x}) \phi'(\boldsymbol{v}_l^\top \boldsymbol{x}) \quad \text{[using definition of } r_l\text{]}$$

$$= \sum_{i=1}^{m} (\boldsymbol{w}_i^\top \boldsymbol{x}) \phi'(\boldsymbol{w}_i^\top \boldsymbol{x}) - \sum_{l=1}^{k} ((\sum_{i=1}^{m} \mathbb{I}_{\tau_i=l} \boldsymbol{w}_i)^\top \boldsymbol{x}) \phi'(\boldsymbol{v}_l^\top \boldsymbol{x}) + \sum_{l=1}^{k} (\boldsymbol{r}_l^\top \boldsymbol{x}) \phi'(\boldsymbol{v}_l^\top \boldsymbol{x})$$

$$:= \underbrace{\sum_{l=1}^{k} \sum_{i=1}^{m} \mathbb{I}_{\tau_i=l} (\boldsymbol{w}_i^\top \boldsymbol{x}) \Big( \phi'(\boldsymbol{w}_i^\top \boldsymbol{x}) - \phi'(\boldsymbol{v}_l^\top \boldsymbol{x}) \Big)}_{R_1(\boldsymbol{x})} + \underbrace{\sum_{l=1}^{k} (\boldsymbol{r}_l^\top \boldsymbol{x}) \phi'(\boldsymbol{v}_l^\top \boldsymbol{x})}_{R_2(\boldsymbol{x})} \; .$$

Then we can derive the lower bound for $\sum_{i=1}^{m} \left\langle \frac{\partial}{\partial \boldsymbol{w}_i} L(\boldsymbol{W}), \boldsymbol{w}_i - \boldsymbol{w}_i^\star \right\rangle$ that:

$$\sum_{i=1}^{m} \left\langle \frac{\partial}{\partial \boldsymbol{w}_i} L(\boldsymbol{W}), \boldsymbol{w}_i - \boldsymbol{w}_i^\star \right\rangle$$

$$= \sum_{i=1}^{m} \mathbb{E}_{\boldsymbol{x}} \Big( R(\boldsymbol{x}) \phi'(\boldsymbol{w}_i^\top \boldsymbol{x}) \boldsymbol{x}^\top (\boldsymbol{w}_i - \boldsymbol{w}_i^\star) \Big)$$

$$= \mathbb{E}_{\boldsymbol{x}} \left[ R(\boldsymbol{x}) \sum_{i=1}^{m} \Big( \phi(\boldsymbol{w}_i^\top \boldsymbol{x}) - \phi'(\boldsymbol{w}_i^\top \boldsymbol{x}) \boldsymbol{x}^\top \boldsymbol{w}_i^\star \Big) \right]$$

$$= 2L(\boldsymbol{W}) + \mathbb{E}_{\boldsymbol{x}} \left[ R(\boldsymbol{x}) \Big( \sum_{i=1}^{m} \Big( \phi(\boldsymbol{w}_i^\top \boldsymbol{x}) - \phi'(\boldsymbol{w}_i^\top \boldsymbol{x}) \boldsymbol{x}^\top \boldsymbol{w}_i^\star \Big) - R(\boldsymbol{x}) \Big) \right] \quad \text{[using } L(\boldsymbol{W}) = \frac{1}{2} \mathbb{E}_{\boldsymbol{x}} R(\boldsymbol{x})^2 \text{]}$$

$$= 2L(\boldsymbol{W}) + \mathbb{E}_{\boldsymbol{x}} \left[ R(\boldsymbol{x}) \Big( \sum_{l=1}^{k} \phi(\boldsymbol{v}_l^\top \boldsymbol{x}) - \sum_{i=1}^{m} \Big( \phi'(\boldsymbol{w}_i^\top \boldsymbol{x}) \boldsymbol{x}^\top \boldsymbol{w}_i^\star \Big) \Big) \right] \quad \text{[using definition of } R(\boldsymbol{x})\text{]}$$

$$= 2L(\boldsymbol{W}) + \mathbb{E}_{\boldsymbol{x}} \left[ R(\boldsymbol{x}) \Big( \sum_{i=1}^{m} \Big( \phi'(\boldsymbol{x}^\top \boldsymbol{w}_i^\star) \boldsymbol{x}^\top \boldsymbol{w}_i^\star \Big) - \sum_{i=1}^{m} \Big( \phi'(\boldsymbol{w}_i^\top \boldsymbol{x}) \boldsymbol{x}^\top \boldsymbol{w}_i^\star \Big) \Big) \right] \quad \text{[using definition of } R(\boldsymbol{w}_i^\star)\text{]}$$

$$= 2L(\boldsymbol{W}) + \mathbb{E}_{\boldsymbol{x}} \left[ R(\boldsymbol{x}) \sum_{i=1}^{m} (\boldsymbol{x}^\top \boldsymbol{w}_i^\star) \Big( \phi'(\boldsymbol{x}^\top \boldsymbol{w}_i^\star) - \phi'(\boldsymbol{w}_i^\top \boldsymbol{x}) \Big) \right]$$

$$:= 2L(\boldsymbol{W}) + \underbrace{\mathbb{E}_{\boldsymbol{x}} \left[ R_1(\boldsymbol{x}) \sum_{i=1}^{m} (\boldsymbol{x}^\top \boldsymbol{w}_i^\star) \Big( \phi'(\boldsymbol{x}^\top \boldsymbol{w}_i^\star) - \phi'(\boldsymbol{w}_i^\top \boldsymbol{x}) \Big) \right]}_{I_4}$$

$$+ \underbrace{\mathbb{E}_{\boldsymbol{x}} \left[ R_2(\boldsymbol{x}) \sum_{i=1}^{m} (\boldsymbol{x}^\top \boldsymbol{w}_i^\star) \Big( \phi'(\boldsymbol{x}^\top \boldsymbol{w}_i^\star) - \phi'(\boldsymbol{w}_i^\top \boldsymbol{x}) \Big) \right]}_{I_5} \; .$$

For term $I_4$, note that for $\forall i \in [m]$, when $\boldsymbol{w}_i^\top \boldsymbol{x} \geq 0$, we have $\phi'(\boldsymbol{w}_i^\top \boldsymbol{x}) = 1$, which means $\phi'(\boldsymbol{w}_i^\top \boldsymbol{x}) - \phi'(\boldsymbol{v}_l^\top \boldsymbol{x}) \geq 0$. Then we have $R_1(\boldsymbol{x}) \geq 0$. Similar, we have $\sum_{i=1}^{m} (\boldsymbol{x}^\top \boldsymbol{w}_i^\star) \Big( \phi'(\boldsymbol{x}^\top \boldsymbol{w}_i^\star) - \phi'(\boldsymbol{w}_i^\top \boldsymbol{x}) \Big) \geq 0$. So we have $I_4 \geq 0$.

For term $I_5$, we have:

$$
\begin{aligned}
I_5 &= \mathbb{E}_{\boldsymbol{x}} \sum_{l=1}^{k} (\boldsymbol{r}_l^\top \boldsymbol{x}) \phi'(\boldsymbol{v}_l^\top \boldsymbol{x}) \sum_{i=1}^{m} (\boldsymbol{x}^\top \boldsymbol{w}_i^\star) \Big( \phi'(\boldsymbol{x}^\top \boldsymbol{w}_i^\star) - \phi'(\boldsymbol{w}_i^\top \boldsymbol{x}) \Big) \\
&= \sum_{l=1}^{k} \sum_{i=1}^{m} \mathbb{E}_{\boldsymbol{x}} (\boldsymbol{r}_l^\top \boldsymbol{x}) \phi'(\boldsymbol{v}_l^\top \boldsymbol{x}) (\boldsymbol{x}^\top \boldsymbol{w}_i^\star) \Big( \phi'(\boldsymbol{x}^\top \boldsymbol{w}_i^\star) - \phi'(\boldsymbol{w}_i^\top \boldsymbol{x}) \Big) \\
&\geq - \sum_{l=1}^{k} \sum_{i=1}^{m} \mathcal{O}(\|\boldsymbol{r}_l\| \, \theta_{i\star}^2 \, \|\boldsymbol{w}_i^\star\|) ,
\end{aligned}
$$

where the last inequality is from the proof of Xu and Du (2023, Lemma 8).

Thus we have:

$$
\begin{aligned}
\sum_{i=1}^{m} \left\langle \frac{\partial}{\partial \boldsymbol{w}_i} L(\boldsymbol{W}), \boldsymbol{w}_i - \boldsymbol{w}_i^\star \right\rangle &= 2L(\boldsymbol{W}) + I_4 + I_5 \\
&\geq 2L(\boldsymbol{W}) - \sum_{l=1}^{k} \sum_{i=1}^{m} \mathcal{O}(\|\boldsymbol{r}_l\| \, \theta_{i\star}^2 \, \|\boldsymbol{w}_i^\star\|) \\
&\geq 2L(\boldsymbol{W}) - \mathcal{O}(k\theta_{\max}^2 \sum_{l=1}^{k} \|\boldsymbol{r}_l\| \, \|\boldsymbol{v}\|) ,
\end{aligned}
$$

which finishes the proof. $\qquad\square$

**Lemma 5** (Bounds of $\theta_{i\star}$ and $\|\boldsymbol{r}\|$)**.** *Given that $\frac{\|\boldsymbol{v}\|}{3m_{\tau_i}} \leq \|\boldsymbol{w}_i\| \leq \frac{3\|\boldsymbol{v}\|}{m_{\tau_i}}$ and $L(\boldsymbol{W}) = o(\|\boldsymbol{v}\|^2 \, k^{10})$, then we have:*

$$
\|\boldsymbol{r}_l\| \leq \mathcal{O}(k^{\frac{11}{4}} \|\boldsymbol{v}\|^{\frac{1}{4}} L^{\frac{3}{8}}(\boldsymbol{W})), \qquad \forall l \in [l] . \tag{52}
$$

$$
\|\boldsymbol{v}\|^2 \, \theta_{i\star}^3 = \Theta(k^3 L(\boldsymbol{W})), \qquad \forall i \in [m] . \tag{53}
$$

*Proof.* The proof technique here heavily depends on (Zhou et al., 2021), so we simplify our proof here. To be specific, using the same proof method as (Zhou et al., 2021, Lemma C.6), we have:

$$
\sum_{i=1}^{m} \|\boldsymbol{w}_i\|^2 \, \theta_{i\star}^2 = \mathcal{O}(L^{\frac{1}{2}}(\boldsymbol{W})) .
$$

Similarly, following (Zhou et al., 2021, Lemma 12), we have:

$$
\mathbb{E}_{\boldsymbol{x}} R_1(\boldsymbol{x})^2 = \mathcal{O}\Big( k^{\frac{5}{2}} \|\boldsymbol{v}\|^{\frac{1}{2}} L^{\frac{3}{4}}(\boldsymbol{W}) \Big)
$$

Based on Zhou et al. (2021, Lemma 11), we can derive that:

$$
\mathbb{E}_{\boldsymbol{x}} R_2(\boldsymbol{x})^2 = \Omega\left( \frac{\|\boldsymbol{r}_l\|^2}{k^3} \right), \qquad \forall l \in [k] .
$$

Combine the previous results, for any $l \in [k]$, the upper bound of $\|\boldsymbol{r}_l\|$ is:

$$\frac{\|\boldsymbol{r}_l\|}{k^{\frac{3}{2}}} = \mathcal{O}(\mathbb{E}_{\boldsymbol{x}} R_2(\boldsymbol{x}))$$

$$\leq \mathcal{O}(\mathbb{E}_{\boldsymbol{x}} R(\boldsymbol{x}) + \mathbb{E}_{\boldsymbol{x}} R_1(\boldsymbol{x}))$$

$$= \mathcal{O}\left(L^{\frac{1}{2}}(\boldsymbol{W}) + k^{\frac{5}{4}} \|\boldsymbol{v}\|^{\frac{1}{4}} L^{\frac{3}{8}}(\boldsymbol{W})\right)$$

$$\leq \mathcal{O}\left(k^{\frac{5}{4}} \|\boldsymbol{v}\|^{\frac{1}{4}} L^{\frac{3}{8}}(\boldsymbol{W})\right) \quad [\text{using } L(\boldsymbol{W}) = \mathcal{O}(k^{10} \|\boldsymbol{v}\|^2)].$$

Accordingly, we finish the proof of Eq. (52). Based on this, using the same proof method as Zhou et al. (2021, Lemma 9), we can directly obtain Eq. (53). □

**Lemma 6** (Bound of $\|\boldsymbol{w}_i - \boldsymbol{w}_i^\star\|$). *Given that* $\frac{\|\boldsymbol{v}\|}{3m_{\tau_i}} \leq \|\boldsymbol{w}_i\| \leq \frac{3\|\boldsymbol{v}\|}{m_{\tau_i}}$ *and* $L(\boldsymbol{W}) = o(\frac{\|\boldsymbol{v}\|^2}{k^{\frac{22}{3}}})$, *then for* $\forall i \in [m]$, *we have:*

$$\|\boldsymbol{w}_i - \boldsymbol{w}_i^\star\| \leq \mathcal{O}\left(\frac{k^{\frac{2}{3}} m^{\frac{2}{3}} L^{\frac{1}{3}}(\boldsymbol{W})}{\|\boldsymbol{v}\|^{\frac{2}{3}}}\right) \|\boldsymbol{w}_i\| .$$

*Proof.* By Lemma 5, we have $\theta_{i^\star} = \mathcal{O}\left(\frac{k L^{\frac{1}{3}}(\boldsymbol{W})}{\|\boldsymbol{v}\|^{\frac{2}{3}}}\right)$ and $|H_l| = |\langle \boldsymbol{r}_l, \bar{\boldsymbol{v}}_l \rangle| = \|\boldsymbol{r}_l\| \leq \mathcal{O}(k^{\frac{11}{4}} \|\boldsymbol{v}\|^{\frac{1}{4}} L^{\frac{3}{8}}(\boldsymbol{W})) = o(\|\boldsymbol{v}\|)$. Then we have:

$$\|\boldsymbol{w}_i - \boldsymbol{w}_i^\star\| \leq \|\boldsymbol{w}_i - h_{i^\star} \bar{\boldsymbol{v}}_{\tau_i}\| + \|h_{i^\star} \bar{\boldsymbol{v}}_{\tau_i} - \boldsymbol{w}_i^\star\|$$

$$= \|\boldsymbol{w}_i - h_{i^\star} \bar{\boldsymbol{v}}_{\tau_i}\| + \left|h_{i^\star}\left(1 - \frac{\|\boldsymbol{v}\|}{\sum_{j=1}^m \mathbb{I}_{\tau_j = \tau_i} h_{i^\star}}\right)\right|$$

$$= \|\boldsymbol{w}_i\| \sin \theta_{i^\star} + \frac{h_{i^\star} |H_l|}{\|\boldsymbol{v}\| - |H_l|} \quad [\text{using definition of } H_l]$$

$$\leq \|\boldsymbol{w}_i\| \mathcal{O}\left(\frac{k L^{\frac{1}{3}}(\boldsymbol{W})}{\|\boldsymbol{v}\|^{\frac{2}{3}}}\right) + \frac{\|\boldsymbol{w}_i\| \mathcal{O}(k^{\frac{11}{4}} \|\boldsymbol{v}\|^{\frac{1}{4}} L^{\frac{3}{8}}(\boldsymbol{W}))}{\|\boldsymbol{v}\|}$$

$$\leq \mathcal{O}\left(\frac{k L^{\frac{1}{3}}(\boldsymbol{W})}{\|\boldsymbol{v}\|^{\frac{2}{3}}}\right) \|\boldsymbol{w}_i\| + \mathcal{O}\left(\frac{k^{\frac{11}{4}} L^{\frac{3}{8}}(\boldsymbol{W})}{\|\boldsymbol{v}\|^{\frac{3}{4}}}\right) \|\boldsymbol{w}_i\|$$

$$\leq \mathcal{O}\left(\frac{k L^{\frac{1}{3}}(\boldsymbol{W})}{\|\boldsymbol{v}\|^{\frac{2}{3}}}\right) \|\boldsymbol{w}_i\| \quad [\text{using } L(\boldsymbol{W}) = \mathcal{O}(\frac{\|\boldsymbol{v}\|^2}{k^{\frac{7}{2}}})].$$

□

## H.2 GRADIENT LOWER BOUND

In this subsection, we use the structural lemmas in Appendix H.1 to derive the local gradient lower bound.

**Theorem 8.** *Given that* $\frac{\|\boldsymbol{v}\|}{3m_{\tau_i}} \leq \|\boldsymbol{w}_i\| \leq \frac{3\|\boldsymbol{v}\|}{m_{\tau_i}}$ *for* $\forall i \in [m]$ *and* $L(\boldsymbol{W}) = o(\frac{\|\boldsymbol{v}\|^2}{k^{162}})$, *then we have:*

$$\left\|\frac{\partial L(\boldsymbol{W})}{\partial \boldsymbol{W}}\right\| \geq \Omega\left(\frac{L^{\frac{2}{3}}(\boldsymbol{W})}{k^2 \|\boldsymbol{v}\|^{\frac{1}{3}}}\right) .$$

*Proof.* According to Lemmas 4 and 5, we have:

$$\sum_{i=1}^{m} \left\langle \frac{\partial}{\partial \boldsymbol{w}_i} L(\boldsymbol{W}), \boldsymbol{w}_i - \boldsymbol{w}_i^{\star} \right\rangle \geq 2L(\boldsymbol{W}) - \mathcal{O}(k\theta_{\max}^2 \sum_{l=1}^{k} \|\boldsymbol{r}_l\| \|\boldsymbol{v}\|)$$

$$\geq 2L(\boldsymbol{W}) - \mathcal{O}\left(\left(\frac{kL^{\frac{1}{3}}(\boldsymbol{W})}{\|\boldsymbol{v}\|^{\frac{2}{3}}}\right)^2 k^2 (k^{\frac{11}{4}} \|\boldsymbol{v}\|^{\frac{1}{4}} L^{\frac{3}{8}}(\boldsymbol{W})) \|\boldsymbol{v}\|\right)$$

$$\geq 2L(\boldsymbol{W}) - \mathcal{O}\left(\frac{L^{\frac{25}{24}}(\boldsymbol{W})k^{\frac{27}{4}}}{\|\boldsymbol{v}\|^{\frac{1}{12}}}\right)$$

$$\geq L(\boldsymbol{W}) \quad [\text{using } L(\boldsymbol{W}) = \mathcal{O}(\frac{\|\boldsymbol{v}\|^2}{k^{162}})].$$

Then according to Lemma 6, we have:

$$L(\boldsymbol{W}) \leq \sum_{i=1}^{m} \left\langle \frac{\partial}{\partial \boldsymbol{w}_i} L(\boldsymbol{W}), \boldsymbol{w}_i - \boldsymbol{w}_i^{\star} \right\rangle$$

$$\leq \sum_{i=1}^{m} \left\| \frac{\partial}{\partial \boldsymbol{w}_i} L(\boldsymbol{W}) \right\| \|\boldsymbol{w}_i - \boldsymbol{w}_i^{\star}\|$$

$$\leq \left\| \frac{\partial}{\partial \boldsymbol{W}} L(\boldsymbol{W}) \right\| \mathcal{O}\left(\frac{kL^{\frac{1}{3}}(\boldsymbol{W})}{\|\boldsymbol{v}\|^{\frac{2}{3}}}\right) \sum_{i=1}^{m} \|\boldsymbol{w}_i\|$$

$$= \mathcal{O}\left(k^2 L^{\frac{1}{3}}(\boldsymbol{W}) \|\boldsymbol{v}\|^{\frac{1}{3}}\right) \left\| \frac{\partial L(\boldsymbol{W})}{\partial \boldsymbol{W}} \right\|,$$

which concludes the proof.

$\square$

### H.3 LOCAL CONDITIONAL SMOOTHNESS OF LOSS

In this subsection, we deal with the non-smoothness of $L$. We will prove the smoothness of $L$ when the student neuron has upper and lower bounds

**Lemma 7** (Local Conditional Smoothness of $L$). *Given that $\frac{\|\boldsymbol{v}\|}{5m_{\tau_i}} \leq \|\boldsymbol{w}_i\| \leq \frac{5\|\boldsymbol{v}\|}{m_{\tau_i}}$ for any $i \in [m]$, define the Hessian matrix of $L$ as $\boldsymbol{\Lambda} = \frac{\partial^2 L(\boldsymbol{W})}{\partial \boldsymbol{W}^2}$, then we have $\|\boldsymbol{\Lambda}\|_2 \leq \mathcal{O}(m^2)$.*

*Proof.* According to Safran et al. (2021), we have that $L$ is twice differentiable and the closed-form expression of Hessian $\boldsymbol{\Lambda} = \frac{\partial^2 L(\boldsymbol{W})}{\partial \boldsymbol{W}^2} \in \mathbb{R}^{md \times md}$ can be write as:

$$\boldsymbol{\Lambda} = \begin{pmatrix} \boldsymbol{\Lambda}_{1,1} & \cdots & \boldsymbol{\Lambda}_{1,m} \\ \vdots & \ddots & \vdots \\ \boldsymbol{\Lambda}_{m,1} & \cdots & \boldsymbol{\Lambda}_{1,m} \end{pmatrix},$$

where $\boldsymbol{\Lambda}_{i,j} \in \mathbb{R}^{d \times d}, \forall i, j \in [m]$, we will discuss below.

For diagonal elements:

$$\boldsymbol{\Lambda}_{i,i} = \frac{1}{2}\boldsymbol{I} + \sum_{j=1, j \neq i}^{m} \boldsymbol{\Lambda}_1(\boldsymbol{w}_i, \boldsymbol{w}_j) - \sum_{l=1}^{k} \boldsymbol{\Lambda}_1(\boldsymbol{w}_i, \boldsymbol{v}_l), \quad \forall i \in [m],$$

and by defining $\boldsymbol{n}_{\boldsymbol{w}, \boldsymbol{v}} = \bar{\boldsymbol{v}} - \cos \angle(\boldsymbol{w}, \boldsymbol{v}) \bar{\boldsymbol{w}}$, $\boldsymbol{\Lambda}_1$ can be rewritten as:

$$\boldsymbol{\Lambda}_1(\boldsymbol{w}, \boldsymbol{v}) = \frac{\sin \angle(\boldsymbol{w}, \boldsymbol{v}) \|\boldsymbol{v}\|}{2\pi \|\boldsymbol{w}\|} \left(\boldsymbol{I} - \bar{\boldsymbol{w}}\bar{\boldsymbol{w}}^\top + \bar{\boldsymbol{n}}_{\boldsymbol{w}, \boldsymbol{v}} \bar{\boldsymbol{n}}_{\boldsymbol{w}, \boldsymbol{v}}^\top\right).$$

We can bound that

$$\|\mathbf{\Lambda}_1(\boldsymbol{w}, \boldsymbol{v})\| \leq \frac{\|\boldsymbol{v}\|}{\|\boldsymbol{w}\|}.$$

Then we have:

$$\|\mathbf{\Lambda}_{i,i}\| \leq \left\|\frac{1}{2}\boldsymbol{I}\right\| + \sum_{j=1, j\neq i}^{m} \|\mathbf{\Lambda}_1(\boldsymbol{w}_i, \boldsymbol{w}_j)\| + \sum_{l=1}^{k} \|\mathbf{\Lambda}_1(\boldsymbol{w}_i, \boldsymbol{v}_l)\|$$

$$= \mathcal{O}(1) + m\mathcal{O}(1) + k\mathcal{O}\left(\frac{m}{k}\right)$$

$$= \mathcal{O}(m), \quad \forall i \in [m].$$

And non−diagonal elements satisfy that:

$$\mathbf{\Lambda}_{i,j} = \frac{1}{2\pi}\left((\pi - \angle(\boldsymbol{w}_i, \boldsymbol{w}_j))\boldsymbol{I} + \bar{\boldsymbol{n}}_{\boldsymbol{w}_i, \boldsymbol{w}_j}\bar{\boldsymbol{w}}_j^\top + \bar{\boldsymbol{n}}_{\boldsymbol{w}_j, \boldsymbol{w}_i}\bar{\boldsymbol{w}}_i^\top\right), \quad \forall i, j \in [m], \text{and } i \neq j.$$

So we have:

$$\|\mathbf{\Lambda}_{i,j}\| \leq \frac{1}{2\pi}(\pi + 1 + 1) \leq 1, \quad \forall i, j \in [m], \text{and } i \neq j.$$

Combining the above results, we have:

$$\|\mathbf{\Lambda}\| \leq \sum_{i=1}^{m}\sum_{j=1}^{m} \|\mathbf{\Lambda}_{i,j}\| \leq m(m-1) + m\mathcal{O}(m) = \mathcal{O}(m^2).$$

$\square$

## H.4 GENERALIZATION ERROR BOUND

In this subsection, we prove the final convergence result, which is also the generalization error bound.

**Theorem 9.** *Suppose the initial condition in Lemma 1 and Assumption 1 2 and 3 holds. If we set $\epsilon_2 = o(m^{-60}k^{-100})$ and $\eta = o(\frac{1}{m})$ in Theorem 7, then $\forall T \in \mathrm{N}$, we have the following statements hold with probability at least $1 - \delta$:*

$$L(\boldsymbol{W}(T + T_2)) \leq \frac{1}{\left(L(\boldsymbol{W}(T_2))^{-\frac{1}{3}} + \Omega\left(k^{-4}\|\boldsymbol{v}\|^{-\frac{2}{3}}\right)\eta T\right)^3}, \tag{54}$$

*and*

$$\frac{\|\boldsymbol{v}\|}{4m_{\tau_i}} \leq \|\boldsymbol{w}_i(T + T_2)\| \leq \frac{4\|\boldsymbol{v}\|}{m_{\tau_i}} \quad \forall i \in [m]. \tag{55}$$

*Proof.* We prove Eqs. (54) and (55) together inductively.

For $T = 0$, Eq. (54) directly hold and by Lemma 3 we have Eq. (55) holds.

Then we assume Eqs. (54) and (55) hold for $0, 1, \ldots, t$ for any $0 < t < T_1$ to prove Eqs. (54) and (55) for $t + 1$.

**Proof of Eq. (54):**

For $\forall i \in [m]$, similar to Eq. (45), we have $\|\nabla_i(t)\| = \mathcal{O}(k\|\boldsymbol{v}\|)$. Then for $\forall \iota \in [0, 1]$, we have:

$$\|\boldsymbol{w}_i(t) - \iota\eta\nabla_i(t)\| \geq \|\boldsymbol{w}_i(t)\| - \eta\|\nabla_i(t)\| \geq \frac{\|\boldsymbol{v}\|}{4m_{\tau_i}} - \eta\mathcal{O}(k\|\boldsymbol{v}\|) \geq \frac{\|\boldsymbol{v}\|}{5m_{\tau_i}},$$

and

$$\|\boldsymbol{w}_i(t) - \iota\eta\nabla_i(t)\| \leq \|\boldsymbol{w}_i(t)\| + \eta\|\nabla_i(t)\| \leq \frac{4\|\boldsymbol{v}\|}{m_{\tau_i}} + \eta\mathcal{O}(k\|\boldsymbol{v}\|) \leq \frac{5\|\boldsymbol{v}\|}{m_{\tau_i}}.$$

Then, we can use Lemma 7 for $\boldsymbol{W}(t) - \iota\eta\nabla_{\boldsymbol{W}}(t)$ in the following proof.

For $T_2 \leq t \leq T + T_2 - 1$, according to the classic analysis of gradient descent in Nesterov et al. (2018), we have:

$$
\begin{aligned}
L(\boldsymbol{W}(t+1)) &= L(\boldsymbol{W}(t)) + \langle\nabla_{\boldsymbol{W}}(t), -\eta\nabla_{\boldsymbol{W}}(t)\rangle \\
&\quad + \int_{\iota=0}^{1}(1-\iota)(-\eta\nabla_{\boldsymbol{W}}(t))^{\top}\frac{\partial^2 L}{\partial \boldsymbol{W}^2}(\boldsymbol{W}(t) - \iota\eta\nabla_{\boldsymbol{W}}(t))(-\eta\nabla_{\boldsymbol{W}}(t))\mathrm{d}\iota \\
&\leq L(\boldsymbol{W}(t)) - \eta\|\nabla_{\boldsymbol{W}}(t)\|^2 + \int_{\iota=0}^{1}(1-\iota)\eta^2\|\nabla_{\boldsymbol{W}}(t)\|^2\mathcal{O}(m^2)\mathrm{d}\iota \quad [\text{Lemma 7}].
\end{aligned}
$$

Then we have:

$$
\begin{aligned}
L(\boldsymbol{W}(t)) - L(\boldsymbol{W}(t+1)) &\geq \eta\|\nabla_{\boldsymbol{W}}(t)\|^2 - \int_{\iota=0}^{1}(1-\iota)\eta^2\|\nabla_{\boldsymbol{W}}(t)\|^2\mathcal{O}(m^2)\mathrm{d}\iota \\
&= \eta\|\nabla_{\boldsymbol{W}}(t)\|^2 - \frac{1}{2}\eta^2\|\nabla_{\boldsymbol{W}}(t)\|^2\mathcal{O}(m^2) \\
&\geq \frac{1}{2}\eta\|\nabla_{\boldsymbol{W}}(t)\|^2 \\
&\geq \Omega\left(\frac{\eta L^{\frac{4}{3}}(\boldsymbol{W}(t))}{k^4\|\boldsymbol{v}\|^{\frac{2}{3}}}\right).
\end{aligned}
$$

According to Xu and Du (2023, Lemma 24), let $C_s = \Omega\left(k^{-4}\|\boldsymbol{v}\|^{-\frac{2}{3}}\right)$, then we have:

$$L(\boldsymbol{W}(T+T_2)) \leq \frac{1}{\left(L^{-\frac{1}{3}}(\boldsymbol{W}(T_2)) + \Omega\left(k^{-4}\|\boldsymbol{v}\|^{-\frac{2}{3}}\right)\eta T\right)^3}.$$

**Proof of Eq. (55)**: According to Lemma 3, we have $L(\boldsymbol{W}(T_2)) \leq \frac{1}{2}k^2\epsilon_2^{0.05}\|\boldsymbol{v}\|^2 = o(\frac{\|\boldsymbol{v}\|^2}{m^3 k^3})$.

Then for $\forall i \in [m]$, according to (Xu and Du, 2023, Lemma 24), we have:

$$\|\boldsymbol{w}_i(T+T_2)\| \geq \|\boldsymbol{w}_i(T_2)\| - \sum_{t=0}^{T-1} \eta \|\nabla_{\boldsymbol{W}}(t+T_2)\|$$

$$\geq \frac{\|\boldsymbol{v}\|}{3m_{\tau_i}} - 8C_s^{-\frac{1}{2}} o\left(\frac{\|\boldsymbol{v}\|^2}{m^3 k^3}\right)^{\frac{1}{3}}$$

$$\geq \frac{\|\boldsymbol{v}\|}{3m_{\tau_i}} - 8\mathcal{O}\left(k^{-4}\|\boldsymbol{v}\|^{-\frac{2}{3}}\right)^{-\frac{1}{2}} o\left(\frac{\|\boldsymbol{v}\|^2}{m^3 k^3}\right)^{\frac{1}{3}}$$

$$\geq \frac{\|\boldsymbol{v}\|}{3m_{\tau_i}} - o\left(\frac{k\|\boldsymbol{v}\|}{m}\right)$$

$$\geq \frac{\|\boldsymbol{v}\|}{4m_{\tau_i}},$$

and

$$\|\boldsymbol{w}_i(T+T_2)\| \leq \|\boldsymbol{w}_i(T_2)\| + \sum_{t=0}^{T-1} \eta \|\nabla_{\boldsymbol{W}}(t+T_2)\|$$

$$\leq \frac{3\|\boldsymbol{v}\|}{m_{\tau_i}} + 8C_s^{-\frac{1}{2}} o\left(\frac{\|\boldsymbol{v}\|^2}{m^3 k^3}\right)^{\frac{1}{3}}$$

$$\leq \frac{3\|\boldsymbol{v}\|}{m_{\tau_i}} + 8\mathcal{O}\left(k^{-4}\|\boldsymbol{v}\|^{-\frac{2}{3}}\right)^{-\frac{1}{2}} o\left(\frac{\|\boldsymbol{v}\|^2}{m^3 k^3}\right)^{\frac{1}{3}}$$

$$\leq \frac{3\|\boldsymbol{v}\|}{m_{\tau_i}} + o\left(\frac{k\|\boldsymbol{v}\|}{m}\right)$$

$$\leq \frac{4\|\boldsymbol{v}\|}{m_{\tau_i}},$$

which finishes the proof.

$\square$

