# OpenReview forum: "How Gradient descent balances features: A dynamical analysis for two-layer neural networks"
_ICLR.cc/2025/Conference — ICLR 2025 Poster_

### Official Review · Reviewer_FxhA · 2024-10-23

**Soundness:** 3
**Presentation:** 4
**Contribution:** 3
**Rating:** 6
**Confidence:** 3

**Summary:**

The paper theoretically investigates a two-layer ReLU network in the teacher-student setup. The authors manage to derive a global convergence rate of $O(T^{-3})$ for a multi-neuron teacher and a multi-neuron student. The proof follows a three-phase structure, and the authors develop techniques to handle the interactions of multiple teacher and student neurons.

**Strengths:**

The paper is well written and clearly structured. The extension of previous results to a multiple teacher neurons case is a notable advancement towards understanding the learning dynamics of neural networks. The finding that GD balances the student neurons corresponding to the same teacher neuron also provides insights for the implicit minimum-norm bias in GD. The dynamical system analysis that handles the interactions of multiple teacher and student neurons is also an important theoretical contribution of the paper.

**Weaknesses:**

Major point:

1.	The balancing results seems to reply heavily on the special initialization (a direct consequence from assumption 3 and lemma 1). The role of GD is mainly to preserve this balance throughout all three phases. While it’s interesting that GD can maintain the balance, the result feels somewhat limited due to the dependency on this initialization, making it seem more like a consequence of the setup than a profound discovery about GD itself.


Minor point:

1.	$\sigma$ is used for both the nonlinearity and the variance of initialization. It’s better to prevent notation overlap.

2.	There appears to be a typo at line 189, where student neuron should likely refer to teacher neuron.

3.	In theorem 3 (informal), $\epsilon$ should be related to $\zeta$ in assumption 1 but is not stated.

**Questions:**

1.	In phase 2, the authors claim that upper bound of the angle will increase, but this doesn’t necessarily mean that the angle will increase. Also, from the empirics, the angles appear to be monotonic. Will the angle actually increase as the authors state in line 337 that the angle is slightly larger than that of Phase 1?

2.	In phase 3, is it possible to derive the convergence rates for the angle and the norm as well? If so, which factor dominates the overall convergence rate? Understanding this would provide deeper insight into the dynamics of this phase.

3.	In the numerical experiments, how is the boundary between phase 1 and phase 2 determined?

---

### Official Review · Reviewer_iXmg · 2024-11-03

**Soundness:** 3
**Presentation:** 2
**Contribution:** 3
**Rating:** 6
**Confidence:** 2

**Summary:**

The paper presents a comprehensive analysis of training dynamics in a 2-layer  teacher-student model with ReLU activation and iid Gaussian inputs. The authors prove that, under reasonable assumptions, learning has three distinct phases:
1. alignment - each student neuron aligns with one specific teacher neuron, and not too many students cluster around the same teacher neuron
2. tangential growth - student neurons grow in norm
3. final convergence - when all students neurons are sufficiently alinged with their respective teacher neurons, the loss converges at a rate of $T^{-3}$.

**Strengths:**

The paper is overall well-written and the contribution is important, if true. I am not versed in the relevant literature and cannot attest for the validity of the proofs or derivations. I review the paper while accepting the claims of the authors in the main text at face value. The ACs should verify that the other reviewers can judge the content of the proofs.

The paper is very heavy on mathematical notation and all the "juice" is buried in the 30-page appendix. However, the authors provide intuitive informal explanations of the various theorems, which helps a lot in making the manuscript readable.

**Weaknesses:**

- As I wrote above, the notation is elaborate and difficult to follow, and all the actual scientific content of the paper is in the appendix.
- The paper would benefit a lot from a paragraph or two, preferably accompanied by a diagram, that summarizes the main results, showing the 3 phases and the processes that occur in each phase, the bounds for the duration of each phase and so on. To save space, the current Fig. 1 can be safely omitted IMHO.
- On the same note, it seems that some of the notation is introduced but never used (e.g. $r_j$ in line 152) and others is used only once

**Questions:**

- In Fig. 2-3, bottom rows, it seems that in **all cases** the long time behavior of the loss is significantly slower than $T^{-3}$. Is this not in contradiction to the analytical results?

- The authors write in line 58 about sample complexity, though it seems none of the bounds depend on $n$.  Is sample complexity at all investigated in this work?

- Is Assumption 3 (line 206) justified in a generic setting? It seems that if student neurons are initialized at random, the amount of student neurons that will be close to a given teacher neuron should be distributed binomially, and one should expect a small fraction of them to violate this assumption, no?

---

### Official Review · Reviewer_KUug · 2024-11-04

**Soundness:** 3
**Presentation:** 4
**Contribution:** 3
**Rating:** 6
**Confidence:** 3

**Summary:**

The authors analyze the training dynamics of two-layer neural networks with ReLU activation in teacher-student settings, where both the teacher and student networks have multiple widths. Motivated by the analysis of (Xu and Du, 2023) for the teachers with single neurons, they provide a three-phase convergence framework, consisting of alignment, tangent growth, and local convergence,  to the training, and finally obtain the global convergence guarantee with $O(T^{-1/3})$ local convergence, where $T$ is the number of iteration.

**Strengths:**

The teacher-student setting is one of the well-studied topics in deep learning theory literature, and treating teachers with multiple neurons is still lacking investigation. This paper tackles this critical problem and obtains certain results. The writing of this paper provides a detailed explanation of theoretical outcomes and their proofs, which makes the paper more accessible for readers to follow.

**Weaknesses:**

While this paper provides novel theoretical findings to the teacher-student settings literature, I have several concerns about them.

- The first one is about the restriction of the teacher model. The authors impose several restrictions to the teacher model, such as orthogonality of each neuron and positivity of each coefficient of each neuron. Could the authors relax these assumptions? While the authors mention the orthogonality in the paper, is there any (possible) quantitative evaluation when the orthogonality does not hold? Moreover, I am curious about the accessibility to the case where both positive and negative teacher neurons exist.

- The other one is the assumption of weak recovery, which the authors refer to in the conclusion. Although how the student neurons align to one of the teacher neurons is of interest, this assumption seems to impose this at initialization while the alignment phase still exists. Moreover, I could not find how $\zeta$ in Assumption 1 can be small in the statements. Please correct me if there is anything I may have missed.

**Questions:**

Besides the questions listed in the above, I am curious about the connection to [1], which also treats the three-stage convergence for regularized two-layer neural networks in the teacher-student settings.

[1] Zhou, Mo, and Rong Ge. "How Does Gradient Descent Learn Features--A Local Analysis for Regularized Two-Layer Neural Networks." arXiv preprint arXiv:2406.01766 (2024).

---

### Comment · Area_Chair_KRws · 2024-11-23
**Discussion period ending soon**

Thank you, Reviewer FxhA, for acknowledging the authors' reply.

To the other reviewers: Please review the authors' replies and the feedback from your peers. If any concerns remain, feel free to ask for clarifications. This is your final opportunity to engage.

Thank you for your efforts.

Best regards,
Area Chair

---

### Meta-Review · Area_Chair_KRws · 2024-12-16

**Metareview:**

### Summary

The paper investigates the dynamics of a two-layer ReLU teacher-student network with multiple neurons. It presents a three-phase convergence framework: alignment, tangential growth, and final convergence, leading to global convergence guarantees. This extends previous work by addressing multi-neuron teacher-student networks. The paper also provides insights into the implicit minimum-norm bias of gradient descent (GD) in this setup.

### Strengths

The paper provides new theoretical insights into the convergence dynamics of multi-layer neural networks, particularly by extending the teacher-student framework to networks with multiple neurons. Despite its technical complexity, the paper is well-written and accessible, with improvements in clarity made in response to reviewer feedback.

### Weaknesses

The paper relies on strong assumptions, such as orthogonality, positivity, and balancing, which, while common in such analyses, may limit the generality of the results. These assumptions raise concerns about the broader applicability of the findings.

### Reasons for acceptance

The paper makes a valuable theoretical contribution by extending the teacher-student model to multi-neuron networks. Despite concerns about assumptions, initialization, and notation, the authors have effectively addressed many reviewer criticisms. Given the significance of the theoretical findings and their potential impact on the community, I recommend accepting the paper.

**Additional Comments On Reviewer Discussion:**

The main critiques from the reviewers focused on **clarity** and the **strong assumptions** used by the authors. During the discussion period, the authors addressed most of these concerns effectively, providing clarifications that showed their assumptions are in line with previous work. Additionally, they made significant improvements to the presentation, enhancing the accessibility of the key concepts and providing summary presentations that facilitate the reader.

In conclusion, the reviewers generally agree that the paper offers a valuable theoretical contribution and represents an important step forward in understanding the dynamics of multi-layer neural networks within the teacher-student framework. With the revisions made, the paper is on track to be a significant contribution to the field.

---

### Decision · Program_Chairs · 2025-01-22

Accept (Poster)